# Tudor staphylococcal nuclease is a docking platform for stress granule components and is essential for SnRK1 activation in *Arabidopsis*

Emilio Gutierrez-Beltran[1,2,*] (iD), Pernilla H Elander[3], Kerstin Dalman[3] (iD), Guy W Dayhoff II[4],
Panagiotis N Moschou[5,6,7] (iD), Vladimir N Uversky[8,9] (iD), Jose L Crespo[1] (iD) & Peter V Bozhkov[3,**] (iD)

## Abstract

Tudor staphylococcal nuclease (TSN; also known as Tudor-SN, p100, or SND1) is a multifunctional, evolutionarily conserved regulator of gene expression, exhibiting cytoprotective activity in animals and plants and oncogenic activity in mammals. During stress, TSN stably associates with stress granules (SGs), in a poorly understood process. Here, we show that in the model plant *Arabidopsis thaliana*, TSN is an intrinsically disordered protein (IDP) acting as a scaffold for a large pool of other IDPs, enriched for conserved stress granule components as well as novel or plant-specific SG-localized proteins. While approximately 30% of TSN interactors are recruited to stress granules de novo upon stress perception, 70% form a protein–protein interaction network present before the onset of stress. Finally, we demonstrate that TSN and stress granule formation promote heat-induced activation of the evolutionarily conserved energy-sensing SNF1-related protein kinase 1 (SnRK1), the plant orthologue of mammalian AMP-activated protein kinase (AMPK). Our results establish TSN as a docking platform for stress granule proteins, with an important role in stress signalling.

**Keywords** heat stress; intrinsically disordered regions; SnRK1/SNF1/AMPK; stress granules; tudor staphylococcal nuclease

**Subject Categories** Plant Biology; Proteomics; RNA Biology

**The EMBO Journal (2021) 40: e105043**

## Introduction

Upon stress perception, eukaryotic cells compartmentalize specific mRNA molecules stalled in translation initiation into a type of evolutionary conserved membrane-less organelles called stress granules (SGs) (Thomas *et al*, 2011; Protter & Parker, 2016). In these biomolecular condensates, mRNA molecules are stored, degraded, or kept silent to prevent energy expenditure on the production of useless, surplus, or even harmful proteins under stress conditions. Recent research in yeast *Saccharomyces cerevisiae* and animal models established the molecular composition of SGs. SGs typically contain translationally arrested mRNAs, small ribosomal subunits, various translation initiation factors (eIF), poly(A)-binding protein (PAB), and a variety of RNA-binding proteins (RBPs) and non-RNA-binding proteins (Buchan & Parker, 2009). SGs play a major role in translational repression by sequestering, stabilizing and storing mRNA molecules, as well as by indirectly modulating signalling pathways (Protter & Parker, 2016; Mahboubi & Stochaj, 2017). Accordingly, SGs have a pro-survival function during stress and relate to cancer and human disease (Wolozin, 2012; Anderson *et al*, 2015; Wolozin & Ivanov, 2019).

Apart from components of SGs, proteomic and genetic screens in yeast and animal models have identified proteins modulating SG assembly, which is a highly coordinated process driven by the collective interactions of a core protein–RNA network (Ohn *et al*, 2008; Buchan *et al*, 2011; Martinez *et al*, 2013; Jain *et al*, 2016; Yang *et al*, 2020). A recent model for the assembly of mammalian and yeast SGs encompasses two major steps: first the formation of a dense stable SG core by a liquid–liquid phase separation (LLPS) followed by accumulation of proteins into a peripheral shell (Jain *et al*, 2016; Markmiller *et al*, 2018). Although the molecular

---

1  Instituto de Bioquímica Vegetal y Fotosíntesis, Consejo Superior de Investigaciones Científicas (CSIC)-Universidad de Sevilla, Sevilla, Spain
2  Departamento de Bioquímica Vegetal y Biología Molecular, Facultad de Biología, Universidad de Sevilla, Sevilla, Spain
3  Department of Molecular Sciences, Uppsala BioCenter, Swedish University of Agricultural Sciences and Linnean Center for Plant Biology, Uppsala, Sweden
4  Department of Chemistry, College of Art and Sciences, University of South Florida, Tampa, FL, USA
5  Institute of Molecular Biology and Biotechnology, Foundation for Research and Technology - Hellas, Heraklion, Greece
6  Department of Plant Biology, Uppsala BioCenter, Swedish University of Agricultural Sciences and Linnean Center for Plant Biology, Uppsala, Sweden
7  Department of Biology, University of Crete, Heraklion, Greece
8  Department of Molecular Medicine and USF Health Byrd Alzheimer's Research Institute, Morsani College of Medicine, University of South Florida, Tampa, FL, USA
9  Institute for Biological Instrumentation of the Russian Academy of Sciences, Federal Research Center "Pushchino Scientific Center for Biological Research of the Russian Academy of Sciences", Pushchino, Russia
   *Corresponding author. Tel: +34 954489595; E-mail: egutierrez@us.es
   **Corresponding author. Tel: +46 18673228; E-mail: peter.bozhkov@slu.se

mechanisms underlying SG assembly remain currently unclear, all of the proposed models converge on the idea that SG assembly is driven by a combination of homotypic and heterotypic interactions involving intrinsically disordered regions (IDRs) (Guillen-Boixet et al, 2020; Schmit et al, 2021). Several lines of evidence also suggest that the dynamics of SGs are controlled, at least in part, by SG remodellers, that include ATP-dependent complexes or ubiquitin-related proteins (Seguin et al, 2014; Jain et al, 2016; Marmor-Kollet et al, 2020).

In plants, little is known about the molecular composition and function of SGs as well as their assembly and cross-talk with other signalling pathways. Previous studies in Arabidopsis thaliana (Arabidopsis) revealed formation of SGs under heat, hypoxia and salt stress (Sorenson & Bailey-Serres, 2014; Yan et al, 2014; Gutierrez-Beltran et al, 2015b) and both proteome and metabolome compositions of heat-induced SGs (Kosmacz et al, 2019). To date, a few protein components of plant SGs have been functionally characterized and most of them have direct homologues in yeast and/or mammalian SG proteomes. These include RNA-binding protein 47 (RBP47), Oligouridylate Binding Protein 1 (UBP1) and Tudor Staphylococcal Nuclease (TSN) (Sorenson & Bailey-Serres, 2014; Gutierrez-Beltran et al, 2015b; Kosmacz et al, 2018). Two recent lines of evidence suggest that TSN might be important for assembly and/or function of SGs. First, TSN localizes in SGs in such distant lineages as protozoa, animals and plants (Zhu et al, 2013; Yan et al, 2014; Gao et al, 2015; Gutierrez-Beltran et al, 2015b; Cazares-Apatiga et al, 2017). Second, TSN interacts with proteins constituting the core of SGs, such as PAB1, eIF4E and eIF5A in different organisms including mammalian and Bombyx mori cells (Weissbach & Scadden, 2012; Zhu et al, 2013; Gao et al, 2014).

The domain architecture of TSN is conserved in all studied organisms and includes a tandem repeat of four Staphylococcal Nuclease (SN) domains at the N terminus followed by a Tudor domain and a partial SN domain at the C terminus (Abe et al, 2003; Gutierrez-Beltran et al, 2016). TSN is known to be critically involved in the regulation of virtually all pathways of gene expression, ranging from transcription to RNA silencing (Gutierrez-Beltran et al, 2016; Chou et al, 2017). For example, Arabidopsis TSN1 and TSN2, two functionally redundant TSN homologues, have been described to play two antagonistic roles in RNA metabolism during stress (Gutierrez-Beltran et al, 2015a). Yet, the cellular level of TSN protein itself should be carefully regulated. While its depletion triggers cell death (Sundström et al, 2009; Gutierrez-Beltran et al, 2016; Cui et al, 2018), increased expression of TSN is closely associated with various types of cancer, adding it to a shortlist of most potent oncogenes and attractive targets for anti-cancer therapy (Jariwala et al, 2017; Yu et al, 2017).

In yeast and mammals, the universal molecular components of SGs co-exist with other, cell type- and stress stimuli-specific proteins, suggesting that SGs might play additional, yet unexplored, roles during stress. For example, SG formation in both yeast and human cells mediates target of rapamycin (TOR) signalling under stress by sequestering both TOR complex (TORC1) and downstream kinases (Takahara & Maeda, 2012; Wippich et al, 2013). By contrast, sequestration of the pleiotropic adaptor protein Receptor For Activated C Kinase 1 (RACK1) in SGs inhibits the stress-induced activation of the c-Jun N-terminal kinases (JNK) cascade that triggers apoptotic death (Arimoto et al, 2008). In yet another scenario, sequestration of the

coiled-coil containing Rho-associated protein kinase 1 (ROCK1) into SGs promotes cell survival by abolishing JNK-mediated cell death (Tsai & Wei, 2010). In summary, SG formation can alter signalling pathways by protein sequestration during stress conditions, but whether such a mode of regulation exists in plants remains elusive.

Here, we isolated TSN-interacting proteins from Arabidopsis plants subjected to heat and salt stress, and further combined microscopy, reverse genetics and bioinformatics to advance our understanding of the regulation and molecular function of SGs in plants. We show that TSN engages its highly disordered N-terminal region in providing a platform for docking homologues of key components of yeast and mammalian SGs, as well as novel or plant-specific SG-localized proteins. TSN forms a large disorder-enriched protein–protein interaction network under non-stress conditions, that is poised to enable rapid SG assembly in response to stress. Finally, our data demonstrate that TSN and formation of SGs confer heat-induced activation of the catalytic α-subunit (SnRK1α) of SnRK1 heterotrimeric complex, thus linking TSN and SGs to the energy status of the plant cells.

## Results

### Generation and characterization of Arabidopsis TAPa-expressing lines

As a first step to investigate the role of TSN in SGs, we used TSN2, one of the two Arabidopsis TSN isoforms, as bait for alternative tandem affinity purification (TAPa; Fig EV1A) (Rubio et al, 2005). TSN2 and green fluorescent protein (GFP; negative control) were tagged at their C-termini with TAPa epitope containing two copies of the immunoglobulin-binding domain of protein A from Staphylococcus aureus, a human rhinovirus 3 protease cleavage site, a 6-histidine repeat and 9-Myc epitopes (Figs 1A and EV1A). The resulting TSN2-TAPa and GFP-TAPa vectors were introduced into Arabidopsis Columbia (Col) background. Two lines per construct showing readily detectable expression by immunoblot were selected for further studies (Fig EV1B).

To verify whether the presence of TAPa epitope could affect intracellular localization and functionality of TSN protein, we performed two additional experiments. First, the immunostaining of root cells from 5-day-old seedlings using α-Myc revealed that similar to native TSN (Yan et al, 2014; Gutierrez-Beltran et al, 2015b), TSN2-TAPa displayed diffuse cytoplasmic localization under no stress (NS) conditions but redistributed to cytoplasmic foci following heat stress (HS) (Fig 1B). In contrast, GFP-TAPa remained cytoplasmic regardless of conditions (Fig 1B). Co-localization analysis of the TSN2-TAPa and SG marker eIF4E confirmed that the HS-induced TSN2-TAPa foci are SGs (Fig EV1C). Second, expression of TSN2-TAPa in tsn1 tsn2 seedlings complemented previously reported root cell death phenotype under HS caused by TSN deficiency (Fig EV1D) (Gutierrez-Beltran et al, 2015b). Heat stress induction in these experiments was confirmed by expression analysis of HSP101 and HSF, two HS marker genes (Pecinka et al, 2010) (Appendix Fig S1). Collectively, these data demonstrate that C-terminally TAPa-tagged TSN retains localization and function of its native counterpart when expressed in Arabidopsis and therefore can be used as a bait for the isolation of TSN-interacting proteins.

## Isolation of TSN2-interacting proteins

Since previously we identified TSN2 as a robust marker of SGs induced by HS (Gutierrez-Beltran *et al*, 2015b), we initially analysed TSN2 interactomes isolated from fully expanded leaves of 18-day-old plants growing under NS conditions (23°C) or subjected to HS (39°C for 60 min). To examine the efficiency of purifying TSN2-TAPa and GFP-TAPa proteins from the corresponding transgenic

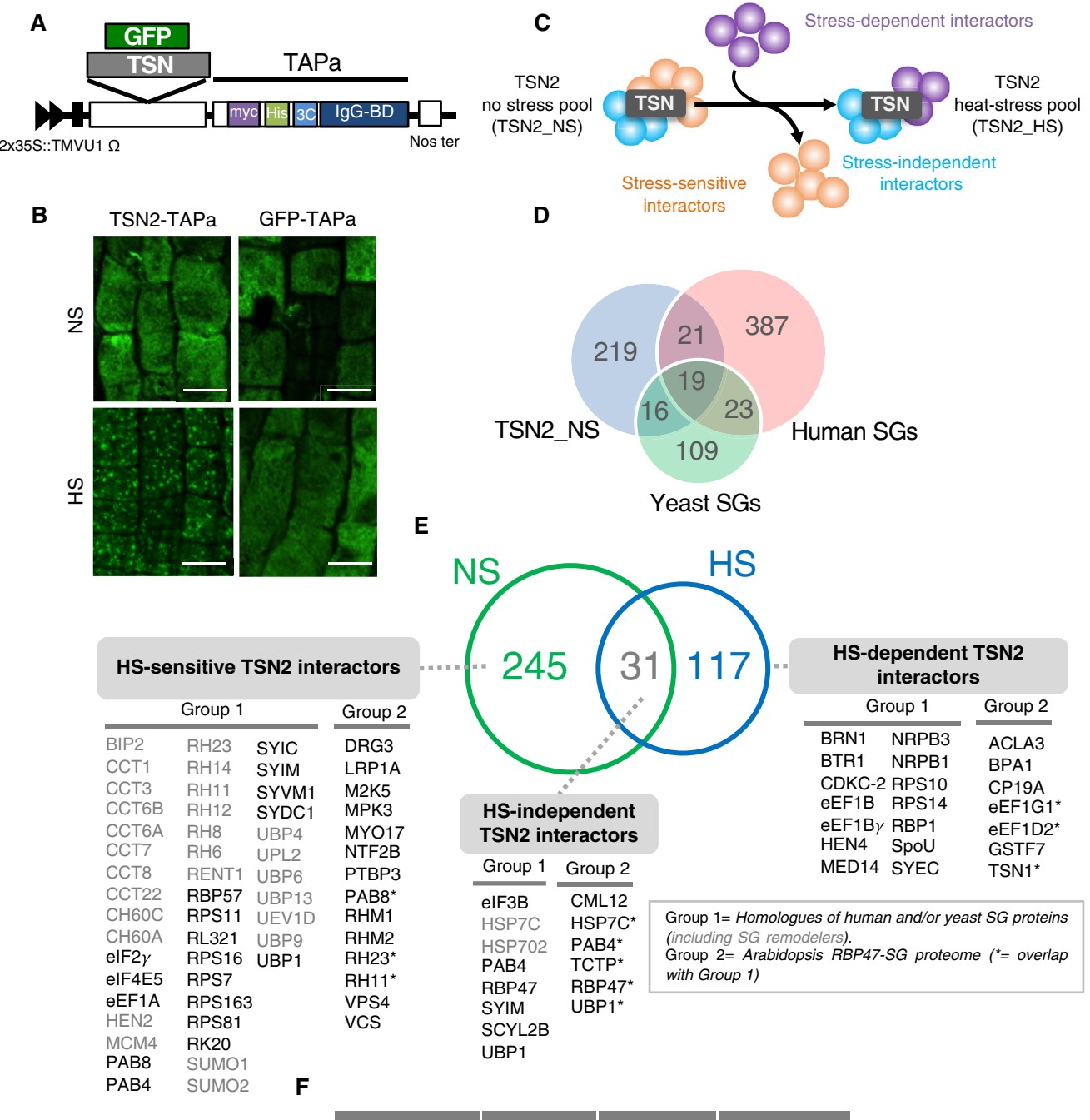

**Figure 1.**

◀

**Figure 1. Tandem affinity purification and characterization of the *Arabidopsis* TSN2-interacting proteins.**

A   Schematic illustration of the expression cassette in TAPa vector. The vector allows translational fusion of TSN or GFP at their C termini to the TAPa tag. The expression is driven by two copies of the cauliflower mosaic virus 35S promoter (2 × 35S) and a tobacco mosaic virus (TMV) U1 X translational enhancer. The TAPa tag consists of two copies of the protein A IgG-binding domain (IgG-BD), an eight amino acid sequence corresponding to the 3C protease cleavage site (3C), a 6-histidine stretch (His), and nine repeats of the Myc epitope (myc). A Nos terminator (Nos ter) sequence is located downstream of each expression cassette.

B   Immunolocalization of TSN2-TAPa and GFP-TAPa fusion proteins in root cells of 5-day-old seedlings. The seedlings were grown under no stress (NS) conditions (23°C) or incubated for 40 min at 39°C (HS) and then immunostained with α-Myc. Scale bars = 10 μm.

C   Schematic representation of three classes of TSN-interacting proteins, i.e. stress-dependent, stress-sensitive and stress-independent interactors, identified upon the comparison of TSN2_NS and TSN2_HS protein pools.

D   Venn diagram showing the comparison of TSN2_NS interactome with human and yeast SG proteomes (Jain *et al*, 2016).

E   Venn diagram showing the comparison between TSN2_NS and TSN2_HS protein pools. TSN2-interacting proteins are divided into three classes: HS-sensitive, HS-independent and HS-dependent. Within each class, the proteins are further classified into two groups. Group 1 contains known human or yeast SG proteins (Jain *et al*, 2016), including SG remodellers (marked in grey colour), whereas group 2 represents components of recently isolated *Arabidopsis* RBP47-SG proteome (Kosmacz *et al*, 2019). The full lists of TSN2-interacting proteins, including previously uncharacterized and potentially novel SG components not belonging to either group 1 or group 2, are provided in Dataset EV1.

F   Frequency of RBPs and proteins with prion-like domains or ATPase activity found in TSN2_NS and TSN2_HS protein pools in comparison with yeast and human SG proteomes (Jain *et al*, 2016).

plants, we performed TAPa approach (Fig EV1A) on a small scale using 10-day-old seedlings. Immunoblot analysis with α-Myc confirmed that both TAPa-tagged proteins could be efficiently purified (Fig EV1E).

Mass spectrometry-based label-free quantitative proteomics analysis yielded 2,535 hits across all samples. The relative abundance of proteins was determined using MaxQuant intensity-based absolute quantification (iBAQ), which reports summed intensity values of the identified peptides divided by the number of theoretical peptides (Tyanova *et al*, 2016; Esgleas *et al*, 2020). In order to identify specific interactors of TSN2, we filtered the results using a two-step procedure. First, we selected proteins specifically enriched in either TSN2_NS or TSN2_HS pools compared with the GFP pool. Thereafter, proteins were filtered based on subcellular localization according to the *Arabidopsis* subcellular database SUBA version 4 (Hooper *et al*, 2017), excluding proteins localized to chloroplasts. As a result, we obtained 277 and 149 proteins representing presumptively physiologically relevant interactomes of TSN2 under NS and HS settings, respectively (TSN2_NS and TSN2_HS pools; Dataset EV1).

## TSN forms a network of SG protein–protein interactions before the onset of stress

The comparison of TSN2_NS and TSN2_HS protein pools enabled classification of TSN interactors into one of the three classes (Fig 1C): (i) stress-independent interactors, which always associate with TSN regardless of conditions; (ii) stress-dependent interactors, which associate with TSN only under HS; and (iii) stress-sensitive interactors, whose association with TSN is lost during HS.

Although SGs are microscopically visible only under stress conditions (Jain *et al*, 2016), analysis of eggNOG orthologs (Huerta-Cepas *et al*, 2019) revealed that ˜20% of proteins from both TSN2_NS and TSN2_HS pools are known components of human or yeast SGs [Fig 1D and E (group 1) and Fig EV2A; Dataset EV2] (Jain *et al*, 2016). Furthermore, the *in silico* analysis showed a significant degree of similarity in the functional distribution of composite proteins between TSN2_NS and TSN2_HS pools and both yeast and human SG proteomes. All of them were enriched in RNA-binding proteins (RBPs), proteins with predicted prion-like domains and proteins with ATPase activity (Fig 1F, Dataset EV2). Apart from the overlaps between TSN2_NS and TSN2_HS pools and yeast and human SG

proteomes, 7.5% (21 hits) and 10.7% (16 hits) proteins from TSN2_NS or TSN2_HS pools, respectively, were shared with the recently published *Arabidopsis* SG proteome [RBP47-SG proteome; Fig 1E (group 2) and Fig EV2B; Dataset EV2] isolated from heat-stressed (30 min at 42°C) seedlings expressing GFP-RBP47 (Kosmacz *et al*, 2019). However, larger parts, i.e. 77% (214 proteins) and 79% (118 proteins) of TSN2_NS and TSN_HS protein pools, respectively, were not shared with either yeast, human or *Arabidopsis* RBP47-SG proteomes (Dataset EV2), representing ample resource for finding novel or plant-specific SG components.

Interestingly, 89% (245/277) of hits from the TSN2_NS pool were absent in the TSN2_HS pool, thus constituting the HS-sensitive part of the TSN2 interactome (Fig 1C and E). A significant part of the HS-sensitive pool was represented by the homologues of yeast or human SG remodellers (Fig 1E, proteins marked in grey colour), including protein chaperones [e.g. cpn60 chaperonin proteins (CCTs) or heat shock proteins, such as CH60s and BIP2], multiple RNA and DNA helicases (e.g. RH, MCM and RENT1) or ubiquitin-related proteins (e.g. SUMO1, SUMO2, UPLs or UBPs). The remaining, smaller part of the TSN2_NS pool (11%, 31/277 proteins) was shared with the TSN2_HS pool and represented HS-independent TSN2 interactors (Fig 1C and E). The latter class of proteins included UBP1, RBP47, PAB4 and TCTP, among others. Lastly, 78% (117/149) proteins from the TSN2_HS pool, including several RBPs (HEN4 or BRN1), individual subunits of eEF1 elongation factor (eEF1B and eEF1Bγ), or DNA-directed RNA polymerase II subunits (NRPB1 and NRPB3), were absent from the TSN2_NS pool, representing HS-dependent TSN2 interactors (Fig 1C and E).

We additionally retrieved publicly available direct protein–protein interaction (PPI) data for all proteins found in our proteomic studies. Both TSN2_NS and TSN2_HS protein pools formed a dense network of protein–protein interactions, comprising 239 and 120 nodes and 1,059 and 177 edges, respectively (Appendix Fig S2). In this context, the average number of interactions per protein for these two pools was 8.86 ($P < 1 \times 10^{-16}$) and 2.95 ($P = 7.5 \times 10^{-07}$), respectively. Together with our findings that known SG remodellers interact with TSN in *Arabidopsis* cells in the absence of stress (Fig 1E), these new results pointed to a pre-existing steady-state network of protein–protein interactions as a basal mechanism during SG formation, where TSN could act as a protein assembly platform.

## TSN is a scaffold protein for SG components

Stress granules are constituted by a dynamic shell and a more stable core (Jain *et al*, 2016). Core proteins have been suggested to act as a scaffold for other SG components (Guillen-Boixet *et al*, 2020; Schmit *et al*, 2021). In a previous study, we observed that TSN did not exchange between the cytoplasm and SG foci upon a fluorescence recovery after photobleaching (FRAP) analysis, suggesting its role as a scaffold protein (Gutierrez-Beltran *et al*, 2015b).

Deletion of scaffold-like molecules is known to have a strong effect on the composition of membrane-less organelles (Espinosa *et al*, 2020; Xing *et al*, 2020). With this in mind and in order to gain a better insight into presumably scaffolding role of TSN in SGs, we investigated the effect of TSN deficiency on the interactome of another plant SG marker protein, RBP47 (Weber *et al*, 2008; Kosmacz *et al*, 2019). For this, we immunoprecipitated GFP-RBP47-bound protein complexes from fully expanded leaves of 18-day-old WT and double *tsn1 tsn2* knockout plants growing under NS conditions (23°C) or subjected to HS (39°C for 60 min) (Appendix Fig S3). In these experiments, we used free GFP-expressing plants as control and followed the same label-free quantitative proteomics procedure as described above for the TSN2-TAPa experiments. Notably, TSN2 was identified in both RBP47_NS and RBP47_HS protein pools (Dataset EV3).

We discovered that in the absence of stress, deletion of TSN resulted in more than 10-fold increase in the RBP47 interactome size accompanied by complete renewal of its protein composition (Fig 2A and B; Dataset EV3). Although TSN deficiency did not significantly affect the size of the RBP47 interactome under HS conditions, it induced almost complete renewal of the protein pool (Fig 2A and B; Dataset EV3). Apart from that, comparison of RBP47 interactomes and TSN2 interactomes revealed ˜11% (31 proteins) overlap in the protein composition between the TSN2_NS pool and RBP47_NS pool isolated from *tsn1 tsn2* plants (Fig 2B). Furthermore, 11 out of 31 shared proteins are homologous to the yeast ATP-driven remodelling complexes [Sheet RBP47_NS (*tsn1 tsn2*), Dataset EV3]. Taken together, these data demonstrate massive reorganization of the RBP47 interactome induced by TSN deficiency, providing evidence for the role of TSN as a scaffold during SG formation.

## TSN-interacting proteins co-localize with TSN2 in cytoplasmic foci

To ascertain the SG localization of TSN2-interacting proteins identified by mass spectroscopy, we selected 16 of the most interesting proteins (Fig 3A). These included homologues of well-known yeast and animal SG-associated proteins (eIF4E5, PAB4 and the ribosomal subunit RPS11) and hypothetical plant-specific SG components with a role in fundamental eukaryotic pathways (e.g. SKP1, MCA-Ia, TCTP and both SnRK1α1 and SnRK1α2 isoforms). First, we performed co-localization studies to investigate whether selected TSN-interacting proteins were translocated to TSN2 foci under stress. To this end, protoplasts were isolated from *Nicotiana benthamiana* (*N. benthamiana)* leaves co-transformed with RFP-TSN2 and individual GFP-TSN-interacting proteins. Co-transformation of the cytoplasmic protein GFP-ADH2 or the SG marker GFP-UBP1 with RFP-TSN2 was used as a negative and

**A**

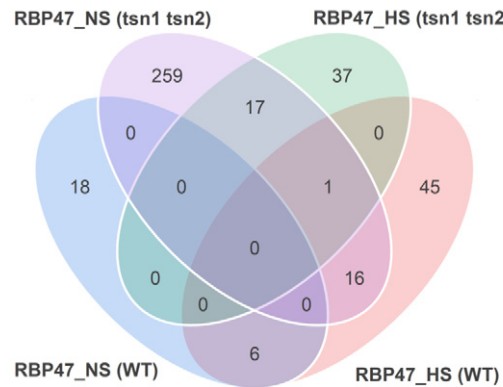

**B**

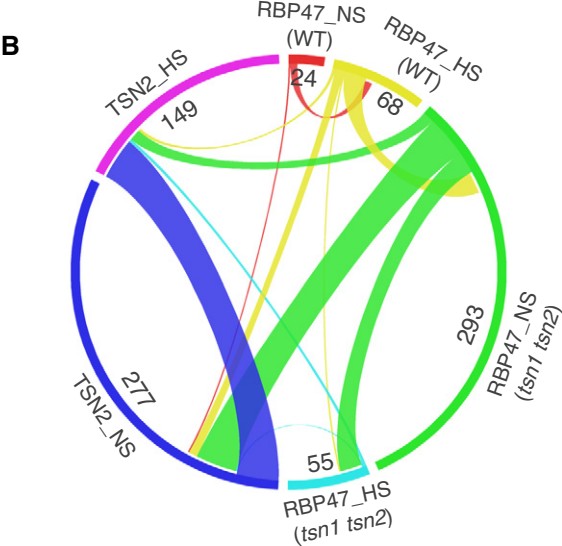

**Figure 2. TSN deficiency promotes a massive reorganization of the RBP47 interactome.**

A Venn diagram showing the comparison between RBP47_NS and RBP47_HS protein pools isolated from WT and *tsn1 tsn2* plants.

B Circos plot showing the comparison between four RBP47 interactomes and two TSN2 interactomes.

positive control, respectively (Fig 3B and C). The degree of co-localization was calculated using pixel correlation analysis (Fig 3C) (French *et al*, 2008). As shown in Figs 3B and C, and EV3; Appendix Fig S4, all selected proteins co-localized with TSN2 in punctate foci upon HS.

Next, to elucidate whether these proteins are associated with TSN2 in the heat-induced SGs, we performed bimolecular fluorescence complementation (BiFC) analyses in *N. benthamiana* leaf cells or protoplasts co-transformed with cYFP-TSN2 and individual nYFP-TSN-interacting proteins. Fluorescence complementation was observed in 10 out of 16 shortlisted TSN-interacting proteins. The YFP signal exhibited diffuse cytoplasmic localization under control conditions (23°C; Appendix Fig S5) and redistributed to punctate foci upon HS (Fig 3D).

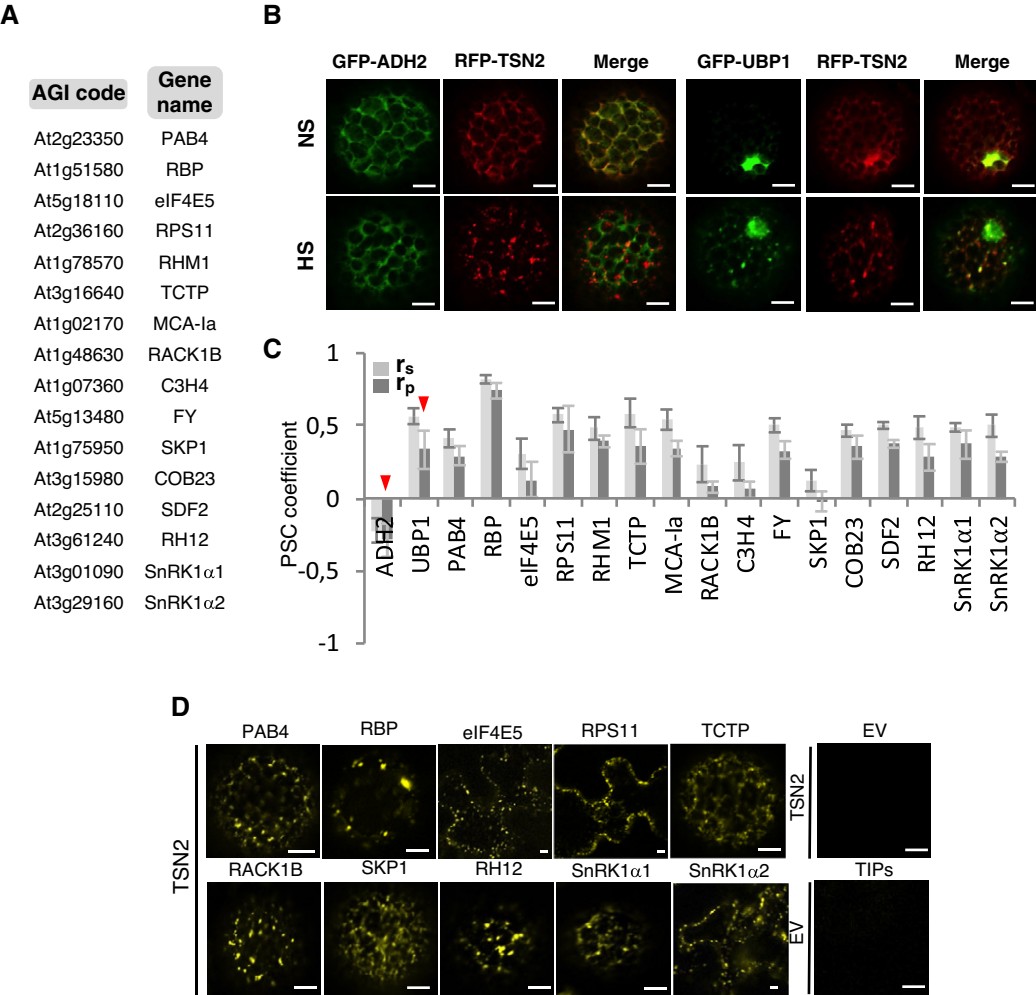

**Figure 3.   TSN2 and its interactors are localized in heat-induced SGs.**

A   A list of TSN-interacting proteins selected for the co-localization analysis.

B   Co-localization of RFP-TSN2 (red) with GFP-ADH2 (negative control) and GFP-UBP1 (positive control) in *N. benthamiana* protoplasts incubated under 23°C (NS) or at 39°C for 40 min (HS). Scale bars = 5 μm.

C   Pearson and Spearman coefficients ($r_p$ and $r_s$, respectively) of co-localization (PSC) of RFP-TSN2 with individual GFP-tagged TSN-interacting proteins listed in *A* and with both negative and positive control proteins (denoted by red arrowheads) under HS. Data represent means ± SD of at least five replicate measurements from three independent experiments.

D   BiFC between cYFP-TSN2 and nYFP-TSN-interacting proteins in *N. benthamiana* leaf cells or protoplasts after HS (39°C for 40 min). BiFC analysis of cYFP-TSN2 and nYFP-TSN-interacting proteins (TIPs) with empty vectors (EV) encoding nYFP and cYFP, respectively, was used as a negative control. Only one representative example of BiFC between cYFP and nYFP-TIP is shown. Scale bars = 5 μm.

To further corroborate the association of TSN with novel plant SG components *in planta*, translationally controlled tumour protein (TCTP) and an uncharacterized RNA-binding protein (RBP) were selected. TCTP was previously observed in both nuclei and cytoplasm (Betsch *et al*, 2017). GFP-tagged RBP and TCTP proteins re-localized to cytoplasmic foci under HS in *Arabidopsis* root tip cells (Fig EV4A). The SG identity of the RBP and TCTP foci was validated by co-localization analysis with the SG marker eIF4E (Fig EV4B) (Gutierrez-Beltran *et al*, 2015b). Subsequently, a Förster resonance energy transfer (FRET) assay in heat-stressed *N. benthamiana* leaves confirmed that TSN2 interacted with TCTP and RBP (Fig EV4C). Finally, TSN co-immunoprecipitated with both TCTP and RBP but not with GFP (negative control) in *Arabidopsis* protein

extracts, confirming the *in vivo* protein–protein interaction (Fig EV4D). Taken together, these findings further reinforce the view that TSN plays a scaffolding role in recruiting a wide range of proteins to SGs.

### TSN associates with SG proteins via the highly disordered N-terminal region

Studies in mammalian and yeast cells have suggested that SGs are multicomponent viscous liquid droplets formed in the cytoplasm by LLPS (Kroschwald *et al*, 2015; Protter & Parker, 2016). Although the molecular details underlying intracellular LLPS are largely obscure, recent evidence suggests that IDRs mediate this process (Posey *et al*,

2018; Alberti *et al*, 2019; Yang *et al*, 2020). In this context, we estimated the predicted enrichment of IDRs and propensity of proteins for LLPS for both TSN2_NS and TSN2_HS interactomes as compared to GFP-TAPa control protein pools using IUPred and PSPredictor algorithms, respectively (preprint: Sun *et al*, 2019; Erdos & Dosztanyi, 2020). The analysis revealed significant enhancement of IDR frequency (Fig 4A) and propensity for LLPS (Fig 4B) in both TSN interactomes in agreement with the scaffolding role of TSN in the formation of phase-separated granules.

In mammalian cells, IDRs of G3BP or hnRNPA1 regulate SG assembly via LLPS (Molliex *et al*, 2015; Guillen-Boixet *et al*, 2020; Yang *et al*, 2020). Considering this fact as well as that TSN was shown to modulate the integrity of SGs in *Arabidopsis* (Gutierrez-Beltran *et al*, 2015b), we evaluated the per-residue intrinsically disordered propensities of TSN2 itself by six commonly used predictors, including PONDR-VLXT, PONDR-VL3, PONDR-VSL2, IUpred_short, IUpred_long and PONDR-FIT (Meng *et al*, 2015). Figure 4C shows that TSN2 is expected to have 11 disordered regions if averaged for six predictors (score above 0.5). Thus, the SN region (tandem repeat of four N-terminally located SN domains) of TNS2 is predicted to be highly disordered, whereas the Tudor domain is predicted to be one of the most ordered parts of the protein. This observation was confirmed using the $D^2P^2$ database providing information about the predicted disorder and selected disorder-related functions (Appendix Fig S6A) (Oates *et al*, 2013). Notably, similar results were obtained for the TSN1 protein isoform which is functionally redundant with TSN2 (Appendix Fig S6A and B) (dit Frey *et al*, 2010).

To investigate whether it is the highly disordered part of TSN which is required for interaction with SG components, we compared the association of the SN region, the Tudor region (composed of the Tudor domain and the fifth, partial SN domain; Fig 4C) and full-length TSN (as a control) with four different TSN-interacting proteins in heat-stressed *N. benthamiana* leaves using BiFC (Fig 4D). The experiment revealed reconstitution of fluorescent signal with all four TSN interactors in case of both full-length TSN2 and SN region, whereas none of the interactors could form a complex with Tudor region (Fig 4D). Furthermore, expression of either full-length TSN2 or SN region yielded identical, punctate BiFC localization pattern. Taken together, these results prompted us to hypothesize that TSN protein could recruit SG components via IDRs, promoting rapid coalescence of microscopically visible SGs upon stress exposure.

### *Arabidopsis* SG-associated proteins are common targets of both TSN1 and TSN2 isoforms

TSN1 and TSN2 proteins were suggested to be redundant in conferring *Arabidopsis* stress tolerance (dit Frey *et al*, 2010; Gutierrez-Beltran *et al*, 2015b). To investigate whether this redundancy is conserved at the SG level, we isolated the TSN1 interactome from unstressed plants using the same TAPa procedure as described above for TSN2 (Fig EV1A, B and E). As a result, we obtained the TSN1_NS pool enriched in 215 proteins (Dataset EV1). Out of these, 110 (51%) were TSN1-specific, whereas the remaining fraction (105 proteins, 49%) represented common interactors of TSN1 and TSN2, reflecting their functional redundancy (Fig 5A). Notably, the pool of common interactors of TSN1 and TSN2 was enriched in homologues

of human and/or yeast SG proteins, such as PAB4, small ribosomal subunits, RNA or DNA helicases or CCT proteins (group 1, Fig 5A). In addition, the common TSN1 and TSN2 interactors included many recently identified members of *Arabidopsis* RBP47-SG proteome (group 2, Fig 5A) (Kosmacz *et al*, 2019), as well as novel plant SG components (group 3, Fig 5A) verified in the current study through either BiFC analysis or co-localization or by using both methods (Figs 3, EV3 and EV4).

To corroborate the proteomics results, we chose DEAD-box ATP-dependent RNA helicase 12 (RH12), as a common interactor of TSN1 and TSN2. RH12 is a nucleocytoplasmic protein associated with SGs under stress (Chantarachot *et al*, 2020). First, we confirmed the molecular interaction between two isoforms of TSN and RH12 by co-immunoprecipitation in cell extracts from *Agrobacterium*-infiltrated *N. benthamiana* leaves. RH12 co-immunoprecipitated with both TSN1 and TSN2 but not with GFP (Fig 5B). Second, we produced *Arabidopsis* lines stably expressing GFP-RH12 under its native promoter and observed re-localization of the fusion protein to heat-induced SGs in root tip cells (Fig 5C and Appendix Fig S7). Taken together, these data are consistent with TSN1 and TSN2 as functionally redundant in providing a scaffold platform for the recruitment of a wide range of plant SG components.

### Identification of a salt stress-induced TSN2 interactome

TSN2 localizes to SGs under salt stress (Yan *et al*, 2014). To investigate the differences between salt stress- and HS-induced TSN interactomes as a proxy for SG proteome variability under different types of stresses, we purified TSN2-interacting proteins from salt-treated *Arabidopsis* plants using our standard TAPa purification procedure. The resulting TSN2_NaCl protein pool was much (9.3–17 times) smaller than both TSN2_NS and TSN2_HS pools, and contained only 16 protein hits (Dataset EV1), 5 and 7 of which were shared with TSN2_NS and TSN2_HS pools, respectively (Fig 6A). Apart from the presence of well-defined mammalian and/or yeast SG proteins, such as HSP70, all three protein pools contained RBP47. To corroborate this result, we performed co-immunoprecipitation of native TSN using protein extracts prepared from GFP-RBP47-expressing *Arabidopsis* seedlings exposed to heat (60 min at 39°C) or salt (60 min, 200 mM NaCl) stress. TSN co-immunoprecipitated with GFP-RBP47 under both stresses, as well as in the absence of stress (Fig 6B), suggesting that RBP47 is a constitutive interactor of TSN that might be recruited to SGs under various types of stresses.

Since we have found that TSN exhibits stress type-dependent variation in both size and composition of its interactome (Fig 6A; Dataset EV1), we then addressed SG recruitment of TSN-interacting proteins to SGs in a stress-type-specific manner using confocal microscopy. To this end, we examined the localization of RBP47 (present in all three TSN2 interactomes), as well as UBP1, TCTP and SnRK1α2 (all present in both TSN_NS and TSN_HS pools but absent in TSN2_NaCl pool) in root tip cells of 5-day-old *Arabidopsis* seedlings expressing GFP-tagged fusions of these proteins. Analysis revealed that while RBP47 and UBP1 were localized to both HS- and salt-induced cytoplasmic puncta, TCTP and SnRK1α2 showed punctate localization only under HS (Fig 6C). These data point to heterogeneity of SG composition in plants and additionally demonstrate that some SG resident proteins might not associate with TSN in SGs (e.g. UBP1 absent in the TSN2_NaCl protein pool).

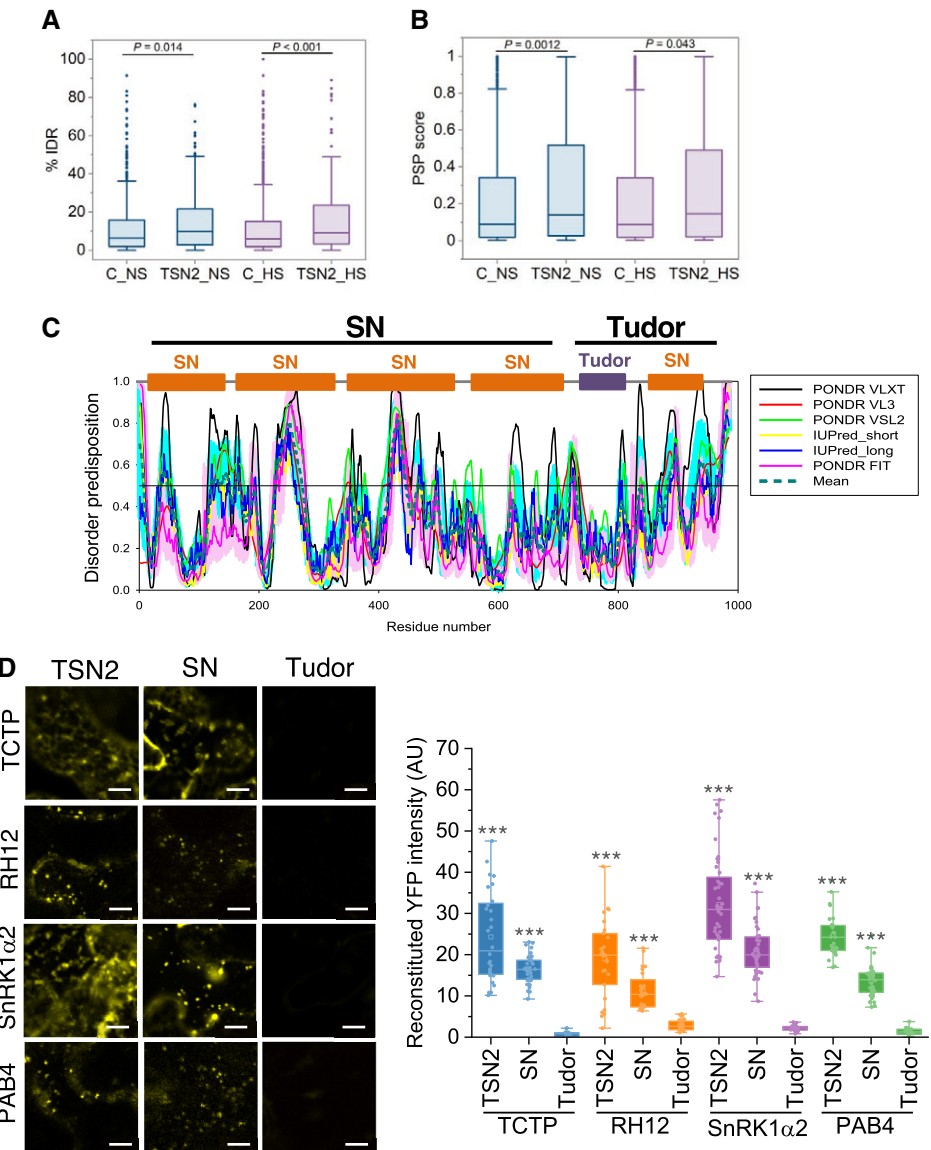

**Figure 4. The highly disordered region of TSN2 is required for interaction with SG proteins.**

A, B  % IDR (A) and propensity for LLPS (B) in TSN2_NS and TSN2_HS interactomes versus corresponding GFP-TAPa controls (C_NS and C-HS) using IUPred and PSPredictor algorithms, respectively. Upper and lower quartiles, medians and extreme points are shown. The number of protein sequences included to the analyses was 566, 277, 995 and 149 for C_NS, TSN2_NS, C_HS and TSN2_HS, respectively. *P* values denote statistically significant differences for comparisons to controls (two-tailed *t*-test).

C  Disorder profiles of TSN2 generated by PONDR-VLXT, PONDR-VL3, PONDR-VSL2, IUPred-short, IUPred-long and PONDR-FIT and a consensus disorder profile (based on mean values of six predictors). SN, staphylococcal nuclease region composed of four N-terminally situated SN domains. C-terminally situated Tudor region is composed of the domain of the same name and a partial SN domain.

D  BiFC between cYFP-TSN2 (full-length), cYFP-SN or cYFP-Tudor and nYFP-TSN-interacting proteins in *N. benthamiana* protoplasts after HS (39°C for 40 min). Scale bars = 10 μm. Boxplots show quantification of the reconstituted YFP signal. AU, arbitrary units. Upper and lower box boundaries represent the first and third quantiles, respectively, horizontal lines mark the median, and whiskers mark the highest and lowest values. Three independent experiments, each containing seven individual measurements, were performed. ***P* < 0.001 versus Tudor (one-way ANOVA).

## TSN interacts with and mediates assembly of SnRK1α in heat SGs

The evolutionary conserved subfamily of yeast sucrose nonfermenting-1 protein kinase (SNF1)/mammalian AMP-activated protein kinase (AMPK)/plant SNF1-related kinase 1 (SnRK1) plays a central role in metabolic responses to declined energy levels in response to nutritional and environmental stresses (Broeckx *et al*, 2016). These kinases typically function as a heterotrimeric complex composed of two regulatory subunits, β and γ, and an α-catalytic subunit. In *Arabidopsis*, the α-catalytic subunit of SnRK1 is encoded

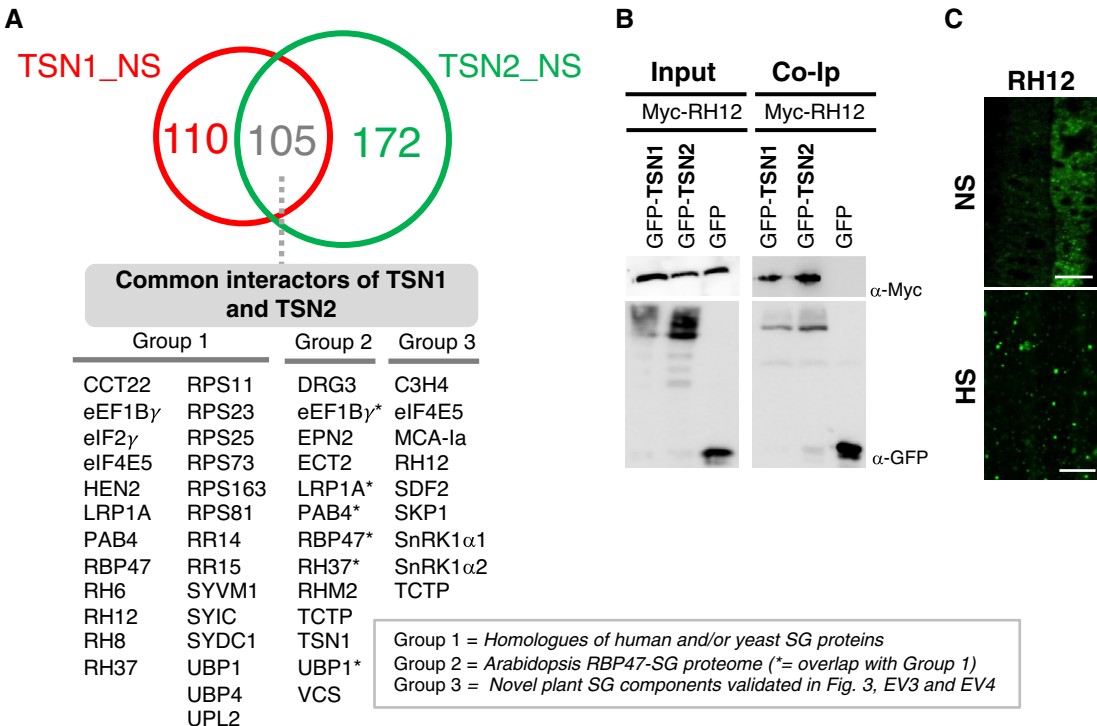

**Figure 5.  Interactomes of *Arabidopsis* TSN1 and TSN2 largely overlap.**

A   Venn diagram showing the overlap between TSN1 and TSN2 interactomes isolated by TAPa from *Arabidopsis* plants grown under NS conditions. Common interactors of TSN1 and TSN2 are classified into three groups: (i) homologues of human and/or yeast SG proteins, (ii) proteins constituting recently isolated *Arabidopsis* RBP47-SG proteome (Kosmacz *et al*, 2019) and (iii) novel plant SG components validated in Figs 3, EV3 and EV4. The full lists of TSN1- and TSN2-interacting proteins, including as yet uncharacterized and potentially novel SG components not belonging to any of the three groups, are provided in Dataset EV1.

B   Co-immunoprecipitation (Co-Ip) of the two TSN isoforms and RH12 in protein extracts prepared from *N. benthamiana* leaves agro-infiltrated with GFP-TSN1 or GFP-TSN2 and Myc-RH12. Free GFP was used as a negative control. Input and Co-Ip fractions were analysed by immunoblotting using α-Myc and α-GFP.

C   Localization of RH12 in root cells of 5-day-old *Arabidopsis* seedlings expressing GFP-RH12 under control of the native promoter. The seedlings were grown under 23°C (NS) or incubated at 39°C for 60 min (HS). Scale bars = 10 µm.

Source data are available online for this figure.

by two functionally redundant genes, *SnRK1α1* and *SnRK1α2* (Baena-Gonzalez *et al*, 2007). We found that SnRK1α1 and SnRK1α2 (also known as KIN10 and KIN11, respectively) are TSN-interacting proteins re-localized to SGs upon HS (Figs 3, 4D and 6C). To dissect the functional relevance of TSN binding and SG localization of SnRK1α1 and SnRK1α2 proteins, we first corroborated the interaction with TSN2 using two different approaches. First, we performed co-immunoprecipitation of native TSN from protein extracts prepared from heat-stressed *Arabidopsis* plants expressing GFP-SnRK1α1 and GFP-SnRK1α2. We found that native TSN co-immunoprecipitated with both GFP-SnRK1α1 and GFP-SnRK1α2 but not with free GFP, which was used as a negative control (Fig 7A). In addition, co-immunoprecipitation analysis confirmed our proteomics data suggesting that SnRK1α is bound to TSN also in the absence of stress (Fig 7B). Second, a FRET assay demonstrated that TSN2 directly interacts with both SnRK1α1 and SnRK1α2 in *N. benthamiana* leaves under HS (Fig 7C).

To explore the molecular link between SnRK1α and TSN2, we first examined the potential localization of SnRK1 in the root tip cells of heat-stressed WT and TSN-deficient (*tsn1 tsn2*) plants. As shown in Fig 7D, TSN is dispensable for localization of either GFP-SnRK1α1 or GFP-SnRK1α2 to cytoplasmic foci. However, we

observed a significant decrease in the number of SnRK1α foci and a simultaneous increase in their size in *tsn1 tsn2* compared with WT plants (Fig 7E and F). The HS induction in WT and *tsn1 tsn2* plants was confirmed by expression analysis of *HSP101* and *HSF* (Appendix Fig S1). We conclude that TSN takes part in the assembly of SnRK1α isoforms in *Arabidopsis* heat SGs.

Stress granules are dynamic structures where many proteins move continuously (Mahboubi & Stochaj, 2017). To investigate the role of TSN in the SnRK1α dynamics, we measured SnRK1α mobility within heat-induced SGs in the root tip cells of WT and *tsn1 tsn2* plants using FRAP. While SnRK1α2 revealed a lack of any fluorescent signal recovery in a TSN-independent manner, TSN deficiency led to a significant decrease in both signal recovery rate and proportion of the initial signal recovery of SnRK1α1 (Figs 7G and H, and EV5A). Thus, we conclude that TSN is required for full mobility of the SnRK1α1 isoform.

**Catalytic and regulatory domains of SnRK1α1 exhibit differential behaviour in SGs**

To investigate the role of N-terminal catalytic and C-terminal regulatory domains of SnRK1α1 (hereafter designated as SnRK1α1$^{CD}$ and

**A**

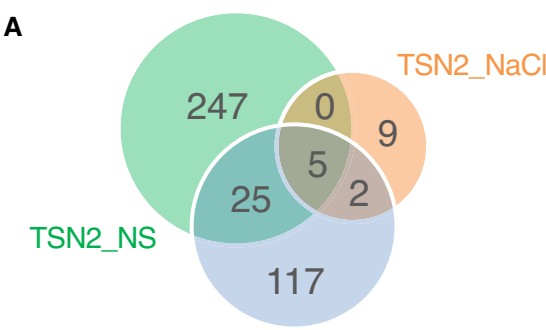

**B**

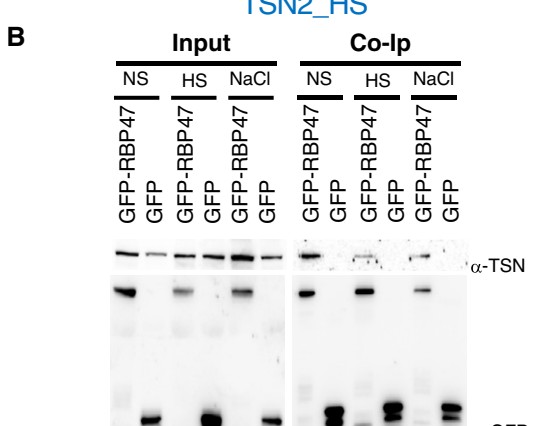

**C**

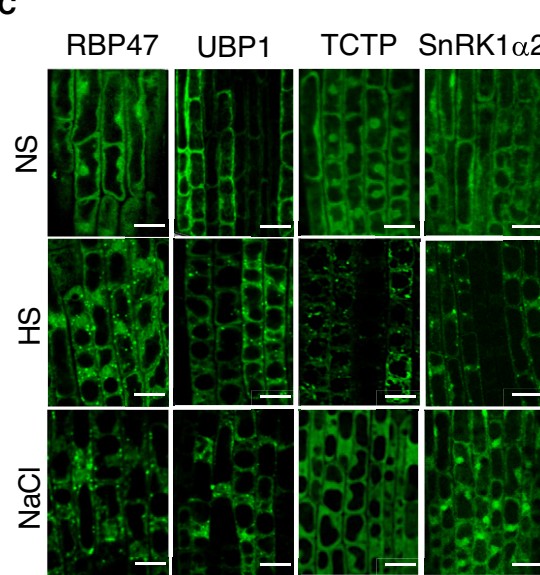

Figure 6.    Identification of *Arabidopsis* salt-induced TSN2 interactome.

A   Venn diagram showing a comparison between TSN2_NS, TSN2_HS and TSN2_NaCl protein pools.
B   Co-immunoprecipitation (Co-Ip) of TSN and RBP47 in protein extracts prepared from 10-day-old *Arabidopsis* seedlings expressing *Pro35S:GFP-RBP47* and grown under no stress (NS), HS (39°C for 60 min) or salt (NaCl) stress (150 mM NaCl for 60 min) conditions. The GFP-expressing line was used as a negative control. Endogenous TSN (107 kD) was detected in total fractions (Input) and fractions co-immunoprecipitated (Co-Ip) with RBP47 but not with free GFP in all three conditions. Input and Co-Ip fractions were analysed by immunoblotting using α-TSN and α-GFP.
C   Localization of GFP-tagged proteins in root cells of 5-day-old *Arabidopsis* seedlings expressing *Pro35S:GFP-RBP47*, *Pro35S:GFP-UBP1*, *Pro35S:GFP-TCTP* and *ProUBQ:GFP-SnRK1α2*. The seedlings were grown under 23°C (NS), incubated at 39°C for 60 min (HS) or treated with 200 mM NaCl at 23°C for 60 min (NaCl). Scale bars = 10 µm.

Source data are available online for this figure.

suppressed by the addition of cycloheximide (CHX), which is known to prevent the formation of SGs in yeast, mammalian and plant cells by reducing the pool of free RNA (Fig 7J) (Gutierrez-Beltran *et al*, 2015b; Jain *et al*, 2016; Saad *et al*, 2017). Punctate and predominantly diffused cytoplasmic localization patterns of the catalytic and the regulatory domains of SnRK1α1, respectively, were also observed in root tip cells from 5-day-old *Arabidopsis* WT seedlings expressing GFP-SnRK1α1[CD] or GFP-SnRK1α1[RD] and exposed to HS (Fig EV5C), and was confirmed by foci quantification (Fig EV5D). Furthermore, in a kinetic analysis the number of SnRK1α1[RD] foci in *N. benthamiana* protoplasts was higher at 20 min than at 40 min of HS (Fig EV5E). Collectively, these results indicate that regulatory and catalytic domains may have different roles in targeting SnRK1α1 to the heat SGs.

**TSN and SGs confer heat-induced activation of SnRK1**

To link SG localization of SnRK1α1 with its heat-dependent regulation, we initially investigated whether HS affects SnRK1 kinase activity *in vivo*. To this end, we subjected 10-day-old WT *Arabidopsis* seedlings to 39°C for 0, 20, 40 and 60 min and then assessed SnRK1α T175 phosphorylation by immunoblotting using α-phospho-AMPK Thr175 (α-pT175), which recognizes phosphorylated forms of both SnRK1α1 (upper band 61.2 kD) and SnRK1α2 (lower band 58.7 kD) (Rodrigues *et al*, 2013; Nukarinen *et al*, 2016). In a control test, we confirmed the α-pT175 affinity efficiency using ABA treatment which is known to induce SnRK1α T175 phosphorylation (Appendix Fig S8) (Jossier *et al*, 2009). Time-course analysis of the level of SnRK1α T175 phosphorylation under HS demonstrated that the two SnRK1α isoforms were rapidly activated by stress (Fig 8A). Yet, the levels of unphosphorylated SnRK1α and TSN remained constant during HS (Fig 8A). To verify whether heat-induced activation of SnRK1α depends on the formation of SGs, the seedlings were treated with CHX and then subjected to HS. CHX treatment abrogated heat-induced phosphorylation of SnRK1α T175 (Fig 8B). To correlate heat-induced activation of the SnRK1α isoforms with their targeting to SGs, we carried out a time-course analysis of SnRK1α localization in root tip cells of 5-day-old seedlings expressing GFP-SnRK1α1 or GFP-SnRK1α2. This analysis revealed that both SnRK1α isoforms become visibly associated with SGs after 40 min of HS and that the number of GFP-SnRK1α foci further increases by 60 min

SnRK1α1[RD], respectively; Fig 7I) in SGs, we monitored the localization of their GFP-tagged variants and SG marker RFP-RBP47 in *N. benthamiana* protoplasts. Under control conditions (NS), both SnRK1α1[CD] and SnRK1α1[RD] domains were localized in the cytoplasm and nucleus, similar to the full-length SnRK1α1 (Fig EV5B). After exposure to HS (40 min at 39°C), SnRK1α1 and SnRK1α1[CD] became associated with RBP47 foci, whereas SnRK1α1[RD] remained mostly in the cytoplasm (Fig 7J). Notably, re-localization of both SnRK1α1 and RBP47 to cytoplasmic puncta during HS was strongly

(Fig 8C and D), perfectly matching the kinetics of SnRK1α T175 phosphorylation (Fig 8A). These results establish a link between the formation of heat SGs and activation of SnRK1α.

To investigate whether TSN is involved in the regulation of the SnRK1α kinase activity, we evaluated the level of SnRK1α T175 phosphorylation in *tsn1 tsn2* seedlings under HS. Similar to the CHX treatment, TSN deficiency prevented heat-induced phosphorylation of SnRK1α T175 (Fig 8E). This effect was reverted by complementation of the *tsn1 tsn2* mutant with *TSN2* (Fig 8F). Thus, we

hypothesized that TSN might be a positive upstream regulator of the SnRK1-dependent stress signalling pathway. Next, we performed RT–qPCR analysis of the *DARK INDUCIBLE 2* (*DIN2; At3g60140*) and *DIN6* (*At3g47340*), two target genes of the SnRK1-dependent signalling pathway (Baena-Gonzalez *et al*, 2007; Rodrigues *et al*, 2013; Belda-Palazon *et al*, 2020), in 10-day-old WT and *tsn1 tsn2* seedlings. Given the lethality of the double *snrk1α1 snrk1α2* knock-out, we employed a partial loss-of-function mutant *snrk1α1$^{-/-}$ snrk1α2$^{-/+}$* as a control (Belda-Palazon *et al*, 2020). Heat stress

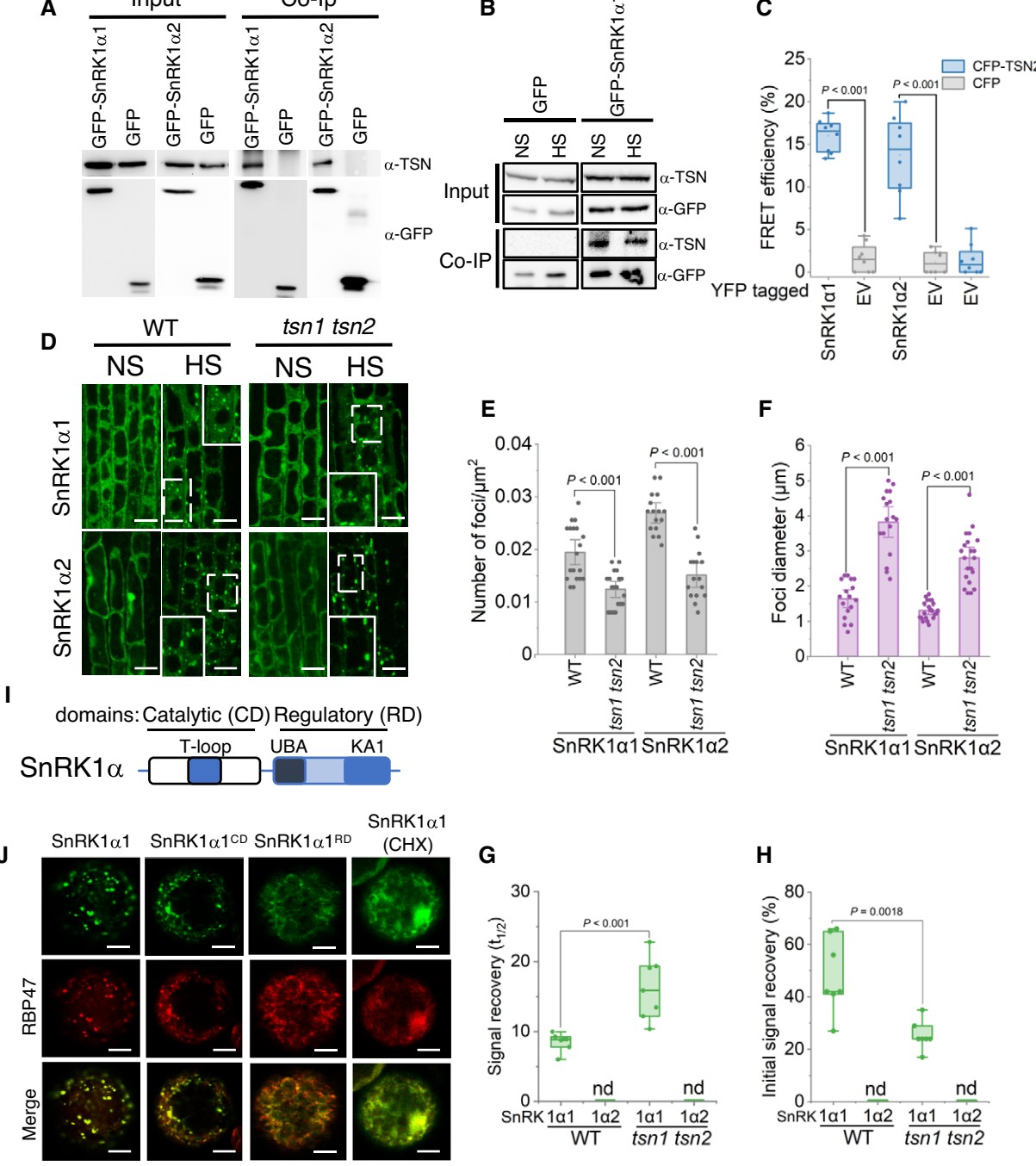

**Figure 7.**

**Figure 7.   TSN interacts with and mediates the assembly of SnRK1α in heat SGs.**

A   Co-immunoprecipitation of TSN and SnRK1α1 and SnRK1α2 in protein extracts prepared from 10-day-old *Arabidopsis* seedlings expressing *ProUBQ:GFP-SnRK1α1* or *ProUBQ:GFP-SnRK1α2* and exposed to HS (39°C for 60 min). The GFP-expressing line was used as a negative control. Endogenous TSN was detected in the total fractions (Input) and in the fractions co-immunoprecipitated (Co-Ip) with SnRK1α1 or SnRK1α2 but not with free GFP. Input and Co-Ip fractions were analysed by immunoblotting using α-TSN and α-GFP.

B   Co-immunoprecipitation of TSN and SnRK1α1 in protein extracts prepared from 10-day-old *Arabidopsis* seedlings expressing *ProUBQ:GFP-SnRK1α1* and grown under NS (23°C) conditions or subjected to HS (39°C for 60 min). Endogenous TSN was detected in the total fractions (Input) and in the fractions co-immunoprecipitated (Co-Ip) with SnRK1α1 under both NS and HS conditions. Input and Co-Ip fractions were analysed by immunoblotting using α-TSN and α-GFP.

C   FRET assay of the indicated protein combinations using CFP-YFP pair in *N. benthamiana* leaves under HS (39°C for 40 min). EV, empty vector (negative control). Upper and lower box boundaries represent the first and third quantiles, respectively; horizontal lines mark the median of at least eight replicate measurements, and whiskers mark the highest and lowest values. The experiment was repeated three times with similar results. *P* values denote statistically significant differences for comparisons to plants expressing EV (two-tailed *t*-test).

D   Localization of GFP-SnRK1α1 and GFP-SnRK1α2 in root cells of 5-day-old *Arabidopsis* WT and *tsn1 tsn2* seedlings grown under 23°C (NS) or incubated at 39°C for 60 min (HS). Insets show enlarged areas inside dashed rectangles. Scale bars = 10 μm.

E, F   Number (E) and size (F) of SnRK1α1- and SnRK1α2-foci in root tip cells of WT and *tsn1 tsn2* seedling expressing *ProUBQ:GFP-SnRK1α1* or *ProUBQ:GFP-SnRK1α2*, respectively, after HS (60 min at 39°C). Data represent means ± SD of at least 16 replicate measurements from three independent experiments. *P* values denote statistically significant differences for comparisons to WT plants (two-tailed *t*-test).

G, H   Signal recovery rate ($t_{1/2}$; G) and proportion of the initial signal recovered (%; H) of GFP-tagged isoforms of SnRK1α in root tip cells of WT and *tsn1 tsn2* seedlings expressing *ProUBQ:GFP-SnRK1α1* and *ProUBQ:GFP-SnRK1α2* after HS (60 min at 39°C). nd, not detected. Upper and lower box boundaries represent the first and third quantiles, respectively; horizontal lines mark the median of at least seven replicate measurements, and whiskers mark the highest and lowest values. The experiment was repeated three times with similar results. *P* values denote statistically significant differences for comparisons to WT plants (two-tailed *t*-test).

I   Schematic diagram of SnRK1α protein structure showing catalytic (CD) and regulatory (RD) domains. The CD includes the phosphorylated T-loop region. RD includes both kinase-associated 1 (KA1) and ubiquitin-associated (UBA) subdomains.

J   Co-localization of GFP-SnRK1α1, GFP-SnRK1α1$^{CD}$ or GFP-SnRK1α1$^{RD}$ with RFP-RBP47 in *N. benthamiana* protoplasts subjected to HS (40 min at 39°C). For co-localization analysis under NS conditions see Fig EV5B. For CHX treatment, protoplasts were incubated with 200 ng/μl CHX for 30 min at 23°C before HS. GFP and RFP fusion proteins were expressed under the control of the UBQ and 35S promoter, respectively. Scale bars = 5 μm.

Source data are available online for this figure.

induction in WT and mutants was confirmed by increased expression of *HSP101* and *HSF* (Appendix Fig S1). We found that while in WT, the expression of *DIN2* and *DIN6* was markedly enhanced by HS, this effect was abrogated by CHX treatment or deficiency of either SnRK1α (*snrk1α1$^{-/-}$ snrk1α2$^{-/+}$*) or TSN (*tsn1 tsn2*) (Fig 8G). Accordingly, complementation of *tsn1 tsn2* mutant with *TSN2* WT allele partly rescued the HS-dependent increase in expression of *DIN2* and *DIN6* (Fig 8G). Our data demonstrate that TSN is essential for SnRK1-dependent signalling under HS.

## Discussion

One of the earliest, evolutionarily conserved events upon stress perception in eukaryotic cells is the assembly of cytoplasmic SGs which provide a mechanism for cell survival (Thomas *et al*, 2011; Mahboubi & Stochaj, 2017). Understanding the molecular composition and regulation of SGs is a rapidly growing field, but most of the research so far has utilized animal or yeast systems.

The scaffold-client model has been used to explain the composition heterogeneity of membrane-less organelles, such as SGs, in mammalian and yeast cells (Banani *et al*, 2016; Schmit *et al*, 2021). In this model, the assembly of granules is regulated by the valency, concentration and molar ratio of scaffold molecules. In accordance with this model, deletion of the scaffold-like molecules perturbs the molecular composition of membrane-less organelles (Espinosa *et al*, 2020; Xing *et al*, 2020). While scaffolds are defined as components essential for the structural integrity of the membrane-less organelles, clients are not necessary for the integrity but are recruited through interactions with scaffolds. Multiple lines of evidence suggest also that IDRs of scaffold proteins contribute to the assembly of the membrane-less organelles including SGs (Gilks *et al*, 2004; Yang *et al*, 2020; Fomicheva & Ross, 2021).

We have previously shown that TSN is stably associated with *Arabidopsis* SGs and that its deletion affects the structural integrity of SGs, the observations leading us to assume scaffolding role for TSN (Gutierrez-Beltran *et al*, 2015b). In agreement with the proposed role, here we present experimental evidence that TSN deficiency strongly affects the composition of the SG proteome (Fig 2). These results, together with the finding that the N-terminal tandem repeat of four SN domains is an ID-reach region recruiting TSN to SGs and participating in protein–protein interaction make it reasonable to envisage the role of the SN domains of TSN as a docking platform maintaining a pre-existing state of SGs in plant cells.

It has been recently postulated that SG assembly in yeast and mammalian cells is a highly regulated multi-step process controlled by numerous proteins collectively known as SG remodellers, and in particular by ATP-dependent remodelling complexes (Jain *et al*, 2016; Protter & Parker, 2016). Thus, ATP-dependent events mediated by ATPases, such as movement of mRNPs to sites of SG formation by motor proteins or remodelling of mRNPs to load required components, could be imperative for promoting SG assembly. In this context, the interaction of the CCT ATPase complex with SG components and activity of the DEAD-box helicase 1 (Ded1) are both crucial for the proper assembly of SGs in yeast cells (Hilliker *et al*, 2011; Jain *et al*, 2016). In addition to ATP-dependent remodellers, ubiquitin-related proteins including ubiquitin-like SUMO ligases, ubiquitin-protein ligases (UPL) and proteases (UBP) have been shown to control the assembly of mammalian and yeast SGs (Xie *et al*, 2018; Keiten-Schmitz *et al*, 2020; Marmor-Kollet *et al*, 2020). Considering that enrichment of the TSN interactome for SG remodellers, including CCT proteins, SUMO ligases, ubiquitin-related proteins and DEAD-box RNA/DNA helicases, occurs in the absence of stress stimulus (Fig 1E), we hypothesize that interaction between these proteins and TSN is necessary for the early steps of SG assembly in plants. Once stress

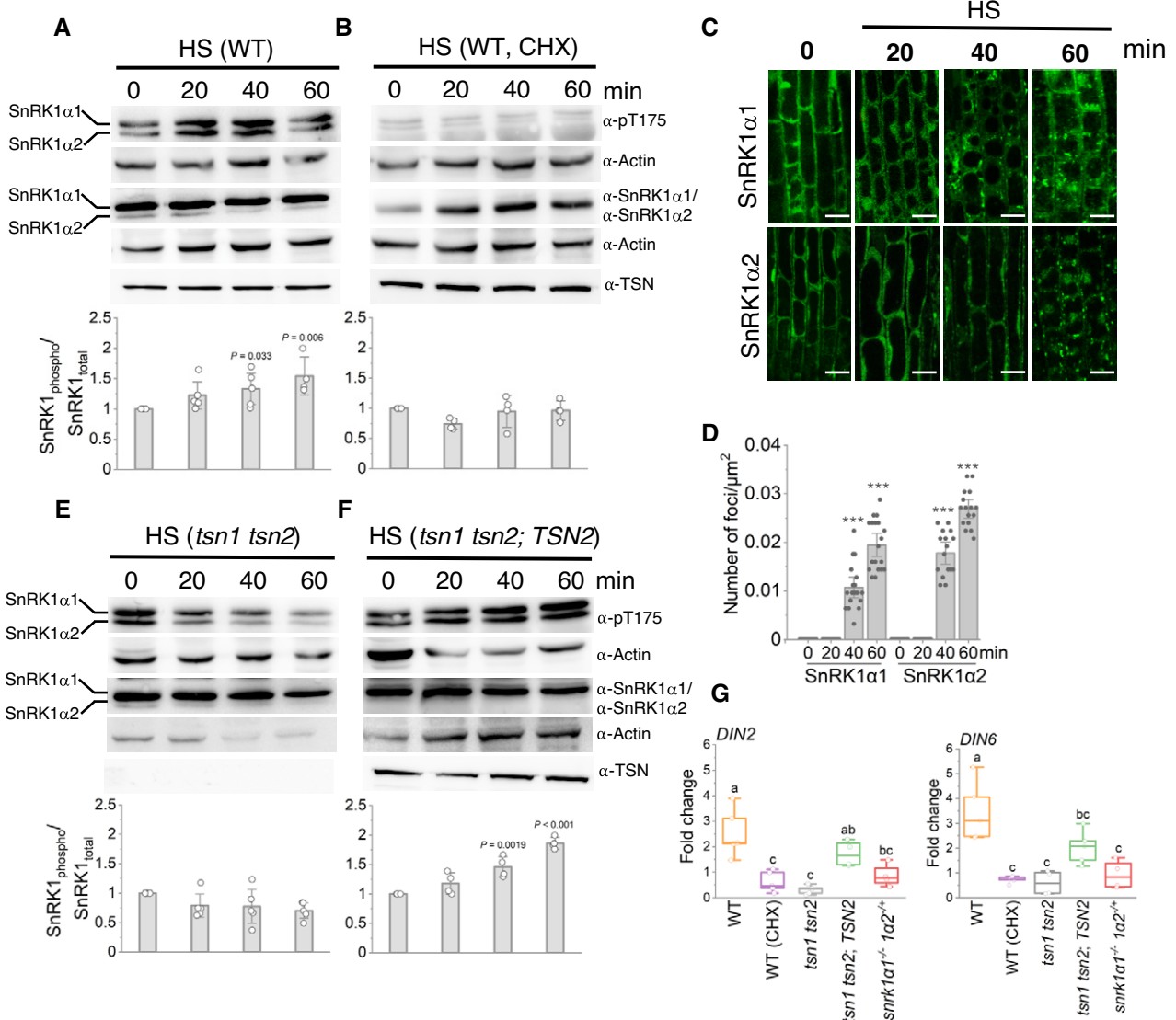

**Figure 8. TSN and SGs confer heat-induced activation of SnRK1.**

A, B Immunoblot analysis with indicated antibodies of protein extracts prepared from root tips of 10-day-old *Arabidopsis* WT heat-stressed seedlings (39°C) 0, 20, 40 and 60 min after the onset of HS. For CHX treatment in B, the seedlings were pre-treated with 200 ng/μl CHX for 30 min before HS. The charts show SnRK1 activity, expressed as the ratio of phosphorylated to total SnRK1 protein. The data represent mean ratios of integrated band intensities (for both isoforms) normalized to 0 min ± SD from at least four different experiments. *P* values denote statistically significant differences for comparisons to 0 min (two-tailed *t*-test).

C Localization of GFP-SnRK1α1 and GFP-SnRK1α2 in root cells of 5-day-old *Arabidopsis* WT seedlings incubated at 39°C and imaged at the indicated time points. Scale bars = 10 μm.

D Quantification of GFP-SnRK1α1 and GFP-SnRK1α2 foci in the experiment shown in C. Data represent means ± SD of at least 16 replicate measurements. The experiment was repeated three times with similar results. \*\*\**P* < 0.05 (two-tailed *t*-test).

E, F Immunoblot analysis with indicated antibodies of protein extracts prepared from root tips of 10-day-old *Arabidopsis tsn1 tsn2* (E) *or tsn1 tsn2* expressing *ProTSN2: GFP-TSN2* (F) heat-stressed seedlings (39°C) 0, 20, 40 and 60 min after the onset of HS. The charts show SnRK1 activity, expressed as the ratio of phosphorylated to total SnRK1 protein. The data represent mean ratios of integrated band intensities (for both isoforms) normalized to 0 min ± SD from at least four different experiments. *P* values denote statistically significant differences for comparisons to 0 min (two-tailed *t*-test).

G Expression levels of *DIN2* and *DIN6* in *Arabidopsis* WT, *tsn1 tsn2*, *tsn1 tsn2;TSN2* and *snrk1α1*[−/−] *snrk1α2*[−/+] 10-day-old heat-stressed seedlings relative to unstressed controls. For CHX treatment, the WT seedlings were pre-treated with 200 ng/μl CHX for 30 min before HS. Upper and lower box boundaries represent the first and third quantiles, respectively. Horizontal lines mark the median of five replicate measurements, and whiskers mark the highest and lowest values. Means with different letters are significantly different at *P* < 0.05 (one-way ANOVA).

Source data are available online for this figure.

stimulus is perceived, the SG remodellers might detach from TSN and aid in SG shell assembly.

Our present study has identified more than 400 TSN interactors, most of which (~77%) are previously unknown candidates for SG components. While this provides a broad resource for functional studies, one should however keep in mind a caveat of detecting non-specific binders by expressing a bait protein under strong promoter (Van Leene *et al*, 2015; Xing *et al*, 2016) and therefore the need for further validation of a particular TSN interactor.

The composition of the SG proteome in animal and yeast cells displays highly variable characteristics influenced by the type of stress or cell type (Markmiller *et al*, 2018). In agreement, we found profound variation in the repertoire of TSN-interacting proteins isolated under different types of stress (Fig 6). One of the most enriched categories of SG-associated proteins is RBPs regulating RNA transport, silencing, translation and degradation (Wolozin & Apicco, 2015). Likewise, RBPs accounted for 55% of TSN2_HS and TSN2_NS interactomes (Fig 1F), providing a further mechanistic explanation for the previously established role of TSN in mRNA stabilization and degradation (Gutierrez-Beltran *et al*, 2015a).

The current predominant model for SG assembly rests on LLPS driven by multivalent interactions through IDRs (Molliex *et al*, 2015; Rayman *et al*, 2018; Kuechler *et al*, 2020). Our data further demonstrate that TSN interactomes under NS and HS conditions are significantly enriched in IDRs (Fig 4A) and proteins with a propensity for LLPS (Fig 4B). Lastly, TSN itself is highly disordered, with the most ID found within tandem of four N-terminally situated SN domains (Fig 4C). This part of TSN confers its interaction with partner proteins, SG localization and cytoprotective property in both mammalian and plant cells (Fig 4D; Gao *et al*, 2015; Gutierrez-Beltran *et al*, 2015b). Taken together, our results demonstrate that the function of IDRs in SG condensation is conserved in plants.

It is well known that numerous stress- and nutrient-signalling pathways converge on SGs (Kedersha *et al*, 2013; Mahboubi & Stochaj, 2017). Our study has established the two-component catalytic subunit of the *Arabidopsis* SnRK1 complex as a TSN interactor. The SnRK1 complex is considered a central regulator of the plant transcriptome in response to darkness and other stress signals (Baena-Gonzalez *et al*, 2007). Recent work showed that overexpression of the catalytic domain of the SnRK1α1 kinase in *Arabidopsis* protoplasts was sufficient to promote SnRK1 signalling (Ramon *et al*, 2019). Here, we show that SG localization of SnRK1α1$^{CD}$ and full-length SnRK1α isoforms coincides with increase in SnRK1α kinase activity (Figs 8 and EV5C and D) pointing to the possibility that targeting to SGs could provide a mechanism for increasing enzyme concentration via condensation to ensure enhanced SnRK1 signalling during stress exposure (Alberti *et al*, 2019; Lyon *et al*, 2021). Furthermore, TSN appears to mediate SnRK1α condensation as its deletion decreased the number and increased the size of the cytoplasmic SnRK1α puncta in the heat-stressed cells (Fig 7E and F). Interestingly, the regulatory domain of SnRK1α1 (SnRK1α1$^{RD}$) revealed a faster association with SGs than SnRK1α1$^{CD}$ upon HS (Figs EV5E and 7J). Given that the SnRK1α1$^{RD}$ is responsible for binding the β and γ regulatory SnRK1 subunits (Kleinow *et al*, 2000) and that SnRK1β2 was shown to control SnRK1α1 localization (Ramon *et al*, 2019), it is tempting to speculate that localization of SnRK1α1 in SG is controlled by interaction with SnRK1 β and γ subunits through its regulatory domain.

SnRK1 and its yeast and mammalian orthologues SNF1 and AMPK, respectively, have been extensively studied as one of the key regulators of target of rapamycin (TOR) (Shaw, 2009; Van Leene *et al*, 2019). In plants, SnRK1 and TOR proteins play central and antagonistic roles as integrators of transcriptional networks in stress and energy signalling (Baena-Gonzalez *et al*, 2007; Belda-Palazon *et al*, 2020). Whereas SnRK1 signalling is activated during stress and energy limitation, TOR promotes growth and biosynthetic processes in response to nutrients and energy availability (Baena-Gonzalez & Hanson, 2017; Carroll & Dunlop, 2017; Van Leene *et al*, 2019). Although it has been demonstrated that the mammalian orthologue (AMPK) is a *bona fide* SG component involved in the regulation of SG biogenesis (Mahboubi *et al*, 2015), there is no evidence connecting SnRK1 activation and SGs. Here, we demonstrate that the formation of SGs and the presence of TSN are both necessary for activation of SnRK1 signalling in response to HS (Fig 8).

It has been shown that mammalian mTOR is translocated to SGs under stress, leading to its inactivation (Heberle *et al*, 2015). While there is no evidence so far that TOR is a component of plant SGs, inhibition of TOR kinase activity in plants by nutritional or cold stress has been reported (Xiong *et al*, 2013; Wang *et al*, 2017). We thus speculate that SGs and their integral constituent protein TSN might regulate the SnRK1-TOR signalling module; however, further work is required to decipher the mechanistic details and physiological roles of this regulation.

In conclusion, our study has two important implications. First, despite recent advances in understanding SGs in mammals and budding yeast, our insights into plant SGs are still very limited. Our work provides a broad resource of SG-related protein interactions and functional data that should promote plant SG research. Second, there is growing evidence linking SGs, AMPK and TSN with cancer and other human diseases. Our work suggests a new mechanism of stress-induced AMPK/SNF1/SnRK1 activation engaging both TSN and formation of SGs. It remains to be seen whether a similar mechanism is conserved in mammals and could thus be used in medical interventions.

# Materials and Methods

### Plant material and growth conditions

The *tsn1 tsn2* double mutant for *TSN1* (At5g07350) and *TSN2* (At5g61780), in the Landsberg erecta (Ler; line CSHL_ET12646) and Columbia (Col; line SALK_143497) backgrounds, respectively, was isolated as shown previously (Gutierrez-Beltran *et al*, 2015b). The mutant was back-crossed five times with Col plants to generate an isogenic pair. Finally, both *tsn1 tsn2* mutant and wild-type (WT) plants were selected from F5. The *snrk1α1$^{-/-}$ snrk1α2$^{-/+}$* mutant was previously described (Ramon *et al*, 2019). *snrk1α1$^{-/-}$ snrk1α2$^{-/+}$* plants were preselected on BASTA-containing medium. Plants were grown on soil or half-strength Murashige and Skoog (MS) medium (Sigma-Aldrich) containing 0.5% sugar and 0.8% agar under long-day conditions (16-h light/8-h dark) at 23°C (NS conditions). For visualization of SGs, 5-day-old seedlings expressing GFP fusion proteins were grown on vertical plates containing half-strength MS medium and incubated for 60 min on a thermoblock at 39°C (HS conditions) or on plates containing 200 mM NaCl (salt stress).

## Plasmid construction

All oligonucleotide primers and constructs used in this study are described in Appendix Tables S1 and S2, respectively. All plasmids and constructs were verified by sequencing using the M13 forward and reverse primers. *TSN1*, *TSN2* and *GFP* were amplified by PCR and resulting cDNA sequences were introduced into pC-TAPa (C-terminal TAPa fusion) to generate *Pro35S:TSN1-TAPa*, *Pro35S:TSN2-TAPa* and *Pro35S:GFP-TAPa,* respectively (Rubio *et al*, 2005). *RH12* cDNA and promoter (2 kb) were amplified and cloned into pGWB4 vector using HiFi DNA assembly cloning kit (NEB biolabs) to generate *ProH12:RH12-GFP*. *TCTP*, *UBP1* and *RBP47* cDNAs were amplified and cloned into pMDC43 vector to generate *Pro35S:GFP-TCTP*, *Pro35S:GFP-UBP1* and *Pro35S:GFP-RBP47*, respectively. *SnRK1α1*, *SnRK1α1CD*, *SnRK1α1RD* and *SnRK1α2* cDNAs were amplified and cloned into pUBC-GFP-Dest vector to generate *ProUBQ:SnRK1α1-GFP* (including variants) and *ProUBQ:SnRK1α2*, respectively (Grefen *et al*, 2010).

cDNA clones of TSN-interacting proteins in the Gateway compatible vector pENTR223 were obtained from the ABRC stock centre (Yamada *et al*, 2003). For expression of N-terminal GFP and RFP fusions under the control of 35S promoter, cDNAs encoding *TSN2* and TSN-interacting proteins were introduced into the destination vectors pMDC43 and pGWB655, respectively (Curtis & Grossniklaus, 2003). For the BiFC assay, cDNAs for *TSN2*, TSN-interacting proteins, and *SN* and *Tudor* regions were cloned into pSITE-BiFC destination vectors (Martin *et al*, 2009). For FRET experiments, cDNAs for TSN2 and TSN-interacting proteins were introduced into pGWB642 (YFP) and pGWB645 (CFP) destination vectors (Nakamura *et al*, 2010).

## Tandem affinity purification

Fully expanded leaves from *Arabidopsis* Col transgenic plants expressing TSN-TAPa and GFP-TAPa and grown for 18 days in 18:6 light/dark conditions at 23°C (NS), 39°C for 60 min (HS) and 200 mM NaCl for 5 h (NaCl) were harvested (15 g, fresh weight) and ground in liquid $N_2$ in 2 volumes of extraction buffer (50 mM Tris–HCl pH 7.5, 150 mM NaCl, 10% glycerol, 0.1% Nonidet P-40 and 1× protease inhibitor cocktail; Sigma-Aldrich) and centrifuged for 12,000 g for 10 min at 4°C. Supernatants were collected and filtered through two layers of Miracloth (Calbiochem). Plant extracts were incubated with 700 µl IgG beads (Amersham Biosciences) for 4–5 h at 4°C with gentle rotation. After centrifugation at 250 g for 3 min at 4°C, the IgG beads were recovered and washed three times with 10 ml of washing buffer (50 mM Tris–HCl, pH 7.5, 150 mM NaCl, 10% glycerol and 0.1% Nonidet P-40) and once with 5 ml of cleavage buffer (50 mM Tris–HCl, pH 7.5, 150 mM NaCl, 10% glycerol, Nonidet P-40 and 1 mM DTT). Elution from the IgG beads was performed by incubation with 15 µl (40 units) of PreScission protease (Amersham Biosciences) in 5 ml of cleavage buffer at 4°C with gentle rotation. Supernatants were recovered after centrifugation at 250 g for 3 min at 4°C and stored at 4°C. The IgG beads were washed with 5 ml of washing buffer, centrifuged again and the eluates pooled. The pooled eluates were transferred together with 1.2 ml of Ni-NTA resin (Qiagen, Valencia, CA, USA) into a 15-ml Falcon tube and incubated for 2 h at 4°C with gentle rotation. After centrifugation at 250 g for 3 min at 4°C, the Ni-NTA resin was washed three times with 10 ml washing buffer. Finally, elution was performed using 4 ml of imidazole-containing buffer (50 mM Tris–HCl pH 7.5, 150 mM NaCl, 10% glycerol, 0.1% Nonidet P-40, 200 mM imidazole). All the steps in the purification procedure were carried out at 4°C. For each large-scale TAPa purification, three TAPa plant samples (15 g, fresh weight each) were processed in parallel as described above. Final eluates were pooled together, proteins were precipitated using TCA/acetone extraction, and 100 µg of protein was digested according to the FASP method (Wisniewski *et al*, 2009). Two biological replicates were performed for isolating TSN interactomes from unstressed and stressed plants, respectively.

## Liquid chromatography and mass spectrometry analysis for TAPa procedure

Peptides were analysed using EASYnano-LC 1000 on a Q Exactive Plus Orbitrap mass spectrometer (Thermo Scientific). Peptides were separated on a pre-column 75 µm × 2 cm, nanoViper, C18, 3 µm, 100 Å (Acclaim PepMap 100) and analytical column 50 µm × 15 cm, nanoViper, C18, 2 µm, 100 Å (Acclaim PepMap RSLC) at a flow rate of 200 nl/min. Water and ACN, both containing 0.1% formic acid, were used as solvents A and B, respectively. The gradient was started and kept at 0–35% B for 0–220 min, ramped to 35–45% B over 10 min and kept at 45–90% B for another 10 min. The mass spectrometer was operated in the data-dependent mode (DDA) to automatically switch between full-scan MS and MS/MS acquisition. We acquired survey full-scan MS spectra from 200 to 1,800 *m/z* in the Orbitrap with a resolution of *R* = 70,000 at *m/z* 100. For data-dependent analysis, the top 10 most abundant ions were analysed by MS/MS, while +1 ions were excluded, with a normalized collision energy of 32%.

## RBP47 immunoprecipitation

Fully expanded leaves from *Arabidopsis* Col and *tsn1 tsn2* transgenic plants (1 g) expressing GFP-RBP47 and GFP and grown for 18 days in 18:6 light/dark conditions at 23°C (NS) and 39°C for 60 min (HS) were harvested (15 g, fresh weight) and ground in liquid $N_2$ in 2 volumes of extraction buffer (50 mM Tris–HCl pH 7.5, 150 mM NaCl, 0.1% Nonidet P-40 and 1× protease inhibitor cocktail; Sigma-Aldrich) and centrifuged for 10,000 g for 15 min at 4°C. Immunoprecipitation was performed with mMACS Epitope Tag Protein Isolation Kits (Miltenyi Biotec). The supernatants were mixed with magnetic beads conjugated to α-GFP (Miltenyi Biotec) and then incubated for 60 min at 4°C. The mixtures were applied to m-Columns (Miltenyi Biotec) in a magnetic field to capture the magnetic antigen–antibody complex. After extensive washing with extraction buffer (four times, 500 µl each) and 50 mM $NH_4HCO_3$ (four times, 500 µl each), immunoaffinity complexes were eluted by removing the column from the magnet and adding 200 µl of $NH_4HCO_3$. Two biological replicates were performed for isolating RBP47 interactomes.

## Liquid chromatography and mass spectrometry analysis for GFP-RBP47 immunoprecipitation

After immunoprecipitation, the peptides were digested on-beads. To this end, 0.2 µg trypsin was added to each sample before overnight incubation at 37°C. The samples were then desalted with stage tip,

dried under vacuum and analysed by LC-MS using Nano LC-MS/MS (Dionex Ultimate 3000 RLSC nano System, Thermo Fisher) interfaced with Eclipse (Thermo Fisher). Samples were loaded onto a fused silica trap column Acclaim PepMap 100, 75 μm × 2 cm (Thermo Fisher). After washing for 5 min at 5 μl/min with 0.1% TFA, the trap column was brought in-line with an analytical column (Nanoease MZ peptide BEH C18, 130A, 1.7 μm, 75 μm × 250 mm; Waters) for LC-MS/MS. Peptides were fractionated at 300 nL/min using a segmented linear-gradient 4–15% of buffer B in buffer A over 30 min (A: 0.2% formic acid and B: 0.16% formic acid, 80% acetonitrile), and then 15%-25%, 25%-50% and 50–90% over 40, 44 and 44 min, respectively. Buffer B then returned at 4% for 5 min for the next run.

The scan sequence began with an MS1 spectrum [Orbitrap analysis, resolution 120,000, scan range from $m/z$ 375–1,500, automatic gain control (AGC) target 1E6, maximum injection time 100 ms]. The top duty cycle (3 s) scheme and dynamic exclusion of 60 s were used for the selection of parent ions of 2–7 charges for MS/MS. Parent masses were isolated in the quadrupole with an isolation window of 1.2 $m/z$, AGC target 1E5, and fragmented with higher-energy collisional dissociation with a normalized collision energy of 30%. The fragments were scanned in Orbitrap with 15,000 resolution. The MS/MS scan range was determined by the charge state of the parent ion, with a lower limit set at 110 amu.

## Mass spectrometry data analysis

The raw data from TAPa and on-bead-digestion were processed using MaxQuant software (version 1.6.10.43) (Tyanova et al, 2016) and searched against an TAIR11 protein database. The following modifications were selected for the search: carbamidomethyl (C; fixed), acetyl (N-term; variable) and oxidation (M; variable). For both the full-scan MS spectra (MS1) and the MS/MS spectra (MS2), the mass error tolerances were set to 20 ppm. Trypsin was selected as a protease with a maximum of two miscleavages. For protein identification, a minimum of one unique peptide with a peptide length of at least seven amino acids and a false discovery rate below 0.01 was required. The match between runs function was enabled, and a time window of one min was set. Label-free quantification was selected using iBAQ (calculated as the sum of the intensities of the identified peptides divided by the number of observable peptides of a protein) (Schwanhausser et al, 2011).

The proteinGroups.txt file, an output of MaxQuant, was further analysed using Perseus 1.16.10.43 (Tyanova et al, 2016). The iBAQ values were normalized to summed total iBAQ value of all proteins of that sample and $\log_2$ transformed. After filtering out the protein groups with no valid Quan value, the missing values were replaced with a random normal distribution of small values. The non-paired two-tailed $t$-test (Tusher et al, 2001) was used to calculate significant differences between the two samples. Identified proteins were considered as interaction partners if their MaxQuant iBAQ values displayed a > 1.5- or 2-fold change enrichment and $P < 0.05$ ($t$-test) when compared to the control. Furthermore, at least two unique peptides were required per protein group.

## Protoplast and plant transformation

Protoplasts were isolated from leaves of 15- to 20-day-old *N. benthamiana* transiently expressing the corresponding fluorescent proteins, as described previously (Wu et al, 2009). The cell walls were digested by incubation in enzymatic solution containing 1% (w/v) Cellulose R-10, 0.25% (w/v) Macerozyme R-10, 20 mM MES-HOK pH 5.7, 400 mM Mannitol, 10 mM $CaCl_2$, 20 mM KCl, 0.1% (w/v) Bovine serum albumin (BSA) for 60 min. Protoplasts were separated from debris by centrifugation (100 $g$, 3 min, 4°C), washed two times with ice-cold W5 buffer (154 mM NaCl, 125 mM $CaCl_2$, 5 mM KCl and 2 mM MES-KOH pH 5.7) and resuspended in ice-cold W5 buffer at a density of $2.5 \times 10^5$ protoplasts/ml. The protoplast suspension was incubated for 15 min on ice before HS at 39°C.

*Arabidopsis* Col plants were transformed as described previously (Clough & Bent, 1998) using *Agrobacterium tumefaciens* (Agrobacterium) strain GV3101. In Figs 5–8, plants from the T2 and T3 generations were used. Transgenic plants were confirmed by genotyping. For transient expression in *N. benthamiana* mesophyll cells, *Agrobacterium* strain GV3101 was transformed with the appropriate binary vectors by electroporation as described previously (Gutierrez-Beltran et al, 2017). Positive clones were grown in Luria-Bertani until reaching $OD_{600} = 0.4$ and were pelleted after centrifugation at 3,000 $g$ for 10 min. Cells were resuspended in MM (10 mM MES, pH 5.7, 10 mM $MgCl_2$ supplemented with 0.2 mM acetosyringone) until $OD_{600} = 0.4$, incubated at room temperature for 2 h and infiltrated in *N. benthamiana* leaves using a 1 ml hypodermic syringe. Leaves were analysed after 48 h using a Zeiss 780 confocal microscope with the 40× water-immersion objective. The excitation/emission wavelength was 480/508 nm for GFP and 561/610 nm for RFP.

## Bimolecular fluorescence complementation (BiFC)

For BiFC assays, *Agrobacterium* strains GV3101 carrying *cYFP-TSN2* *cYFP-SN* or *cYFP-Tudor* and the corresponding *nYFP-TSN-interacting proteins* were co-infiltrated into *N. benthamiana* leaves ($OD_{600} = 0.3$). Fluorescence images were obtained 48 h after infiltration using a Leica TCS Sp2/DMRE confocal microscope, with an excitation wavelength of 514 nm. Transient expression of proteins in *N. benthamiana* leaves via agroinfiltration was performed as previously described (Gutierrez-Beltran et al, 2017).

## Immunocytochemistry and imaging

For immunocytochemistry, roots of 5-day-old *Arabidopsis* seedlings were fixed for 60 min at room temperature with 4% (w/v) paraformaldehyde in 50 mM PIPES, pH 6.8, 5 mM EGTA, 2 mM $MgCl_2$ and 0.4% Triton X-100. The fixative was washed away with phosphate-buffered saline buffer supplemented with Tween-20 (PBST), and cells were treated for 8 min at room temperature with a solution of 2% (w/v), driselase (Sigma-Aldrich) in 0.4 M mannitol, 5 mM EGTA, 15 mM MES pH 5.0, 1 mM PMSF, 10 μg/ml leupeptin and 10 μg/ml pepstatin A. Thereafter, roots were washed twice, 10 min each, in PBST and then in 1% (w/v) BSA in PBST for 30 min before overnight incubation with a primary antibody (rabbit α-eIF4E diluted 1:500 or rabbit/mouse α-Myc diluted 1:500). The specimens were then washed three times for 90 min in PBST and incubated overnight with the corresponding secondary antibody [goat α-rabbit/mouse fluorescein isothiocyanate (FITC)/rhodamine conjugate] diluted 1:200. After washing in PBST, the specimens were mounted in Vectashield mounting medium (Vector Laboratories).

Staining with FDA and SYTOX Orange (both from Molecular Probes, Invitrogen) was performed on 5-day-old *Arabidopsis* seedlings. FDA and SYTOX Orange were added to final concentrations of 250 nM and 2 mg/ml, respectively, in water. After 10 min of incubation in the dark, the samples were washed twice with half-strength liquid MS medium supplemented with 1% (w/v) sucrose, pH 5.7, and observed immediately. For the CHX treatment, the protoplast suspension or seedling roots were incubated with 200 ng/µl drug for 30 min and then heat-stressed at 39°C.

## Förster resonance energy transfer (FRET)

The assay was performed as described previously (Moschou *et al*, 2013). FRET was performed using Zeiss 780 laser scanning confocal microscope and a plan-apochromat 20×/0.8 M27 objective. FRET acceptor photobleaching mode of Zeiss 780 ZEN software was used, with the following parameters: acquisition of 10 pre-bleach images, one bleach scan and 80 post-bleach scans. Bleaching was performed using 488, 514 and 561-nm laser lines at 100% transmittance and 40 iterations. Pre- and post-bleach scans were at minimum possible laser power (0.8% transmittance) for the 458 nm or 514 nm (4.7%) and 5% for 561 nm; 512 × 512 8-bit pixel format; pinhole of 181 µm and zoom factor of 2.0. Fluorescence intensity was measured in the ROIs corresponding to the bleached region. One ROI was measured outside the bleached region to serve as the background. The background values were subtracted from the fluorescence recovery values, and the resulting values were normalized by the first post-bleach time point. Three pre-bleach and three post-bleach intensities were averaged and used for calculations using the formula $FRET_{eff} = (D_{post}\text{-}D_{pre})/D_{post}$, where D is intensity in arbitrary units.

## Fluorescence recovery after photobleaching (FRAP)

The assay was performed as described previously (Moschou *et al*, 2013). Five-day-old seedlings were grown on sterile plates containing half-strength MS with 1% (w/v) sucrose. For HS treatment, plates were incubated for 60 min on a thermoblock at 39°C. GFP fluorescence was detected using a water-corrected 403 objective. During analyses, the FRAP mode of Zeiss 780 ZEN software was set up for the acquisition of one pre-bleach image, one bleach scan and 40 post-bleach scans. In FRAP of SGs, the width of the bleached region was 2 mm. The following settings were used for f photobleaching: 10–20 iterations, 10–60 s per frame and 75% transmittance with the 458- to 561-nm laser lines of the argon laser. Prebleach and post-bleach scans were at the minimum possible laser power (1.4 to 20% transmittance) for 488 or 561 nm and at 0% for all other laser lines, 512 × 512 pixel format and zoom factor of 5.1. Analyses of fluorescence intensities during FRAP were performed in regions of interest corresponding to the size of the bleached region. One region of interest was measured outside the bleached region to serve as the background. The background values were subtracted from the fluorescence recovery values, and the resulting values were normalized by the first post-bleach time point. Initial signal recovery $(\%) = 100 \times (I_{\text{final,post-bleach}} - I_{\text{initial,post-bleach}})/(I_{\text{prebleach}} - I_{\text{initial,post-bleach}})$, where $I$ is the normalized signal intensity (relative to the background intensity). Values were corrected for the artificial loss of fluorescence using values from the neighbouring cells. At

least ten cells from different roots were analysed for each FRAP experiment.

## Protein extraction and immunoblotting

Two hundred milligrams of leaf material were mixed with 350 µl of extraction buffer (100 mM Tris–HCl, pH 7.5, 150 mM NaCl, 0.1% Nonidet P-40 and 1× Protease inhibitor cocktail (Sigma, P599) and centrifuged for 15 min at 14,000 g. 4× Laemmli sample buffer was added to 100 µl supernatant and boiled for 5 min. Equal amounts of supernatant were loaded on 10% poly-acrylamide gels and blotted on a polyvinylidene difluoride (PVDF) membrane. α-Myc and α-rabbit horseradish peroxidase conjugates (Amersham, GE Healthcare) were used at dilutions 1:1,000 and 1:5,000, respectively. The reaction was developed for 1 min using a Luminata Crescendo Millipore immunoblotting detection system (Millipore, WBLUR0500).

For detection of the phosphorylated forms of SnRK1α proteins, 10-day-old seedlings were collected and ground in liquid nitrogen and the proteins were extracted using the following extraction buffer: 25 mM Tris–HCl pH 7.8, 75 mM NaCl, 15 mM EGTA, 10 mM MgCl₂, 10 mM B-glycerophosphate, 15 mM 4-Nitrophenylphosphate bis, 1 mM DTT, 1 mM NaF, 0.5 mM Na₃VO₄, 0.5 mM PMSF, 1% Protease inhibitor cocktail (Sigma, P599) and 0.1% Tween-20. The protein extracts were centrifuged at 14,000 g and 4°C for 10 min and supernatants transferred to a new tube. The protein concentration was measured using Bradford Dye Reagent (Bio-Rad); equal amounts (15 µg) of total protein for each sample were separated by SDS–PAGE (10% acrylamide gel) and transferred to a PVDF membrane (Bio-Rad). The membrane was blocked in TBST buffer containing 5% (w/v) BSA and incubated with primary antibody and secondary antibody. Antibodies used for immunoblotting were as follows: α–Phospho-AMPKα (Thr175) (α-pT175) (dilution 1:1,000, Cell Signaling Technology), α-Kin10 (dilution 1:1,000, Agrisera), α-Kin11 (dilution 1:1,000, Agrisera), α-TSN [dilution 1:1,000, (Sundström *et al*, 2009)] and α-Actin (dilution 1:10,000, Agrisera).

## Co-immunoprecipitation (Co-Ip)

Total proteins were extracted from 10-day-old seedlings with no-salt lysis buffer [50 mM Tris, pH 8.0, 0.1% Nonidet P-40 and 1% Protease inhibitor cocktail (Sigma)] at a fresh weight:buffer volume ratio of 1 g:2 ml. After centrifugation at 6,000 g and 4°C for 5 min, 20 µl of α-GFP microbeads (Miltenyi Biotec) were added to the resultant supernatant and incubated for 1 h at 4°C on a rotating wheel. Subsequent washing and elution steps were performed according to the manufacturer (µMACS GFP Isolation Kit; Miltenyi Biotec). Immunoblot analysis was done essentially as described above, and immunoprecipitates from transgenic lines expressing free GFP were used as controls. GFP-TSN-interacting proteins and native TSN were detected by mouse α-GFP (monoclonal antibody JL-8; Clontech) and mouse α-TSN (Sundström *et al*, 2009) at final dilutions of 1:1,000 and 1:5,000, respectively.

## Image analysis

The image analysis was done using ImageJ v1.41 (NIH) software (http://rsb.info.nih.gov/ij/index.html). For co-localization analyses, we calculated the linear Pearson (rp) and nonlinear Spearman's

rank (rs) correlation coefficient (PSC) for the pixels representing the fluorescence signals in both channels (French *et al*, 2008). Levels of co-localization can range from +1 to −1 for positive and negative correlations, respectively.

### Quantitative RT–PCR

Total RNA was isolated with RNA plant kit (Bioline) from 10-day-old *Arabidopsis* seedlings grown on liquid MS medium with or without HS (60 min at 39°C). First-strand cDNA was generated using the iScript cDNA Synthesis kit (Bio-Rad) in a 20-µl reaction mixture containing 1 µg of total RNA. The PCR mixtures were performed in a final volume of 18 µl using the SsoAdvanced Universal SYBR Green Supermix (Bio-Rad). The data were normalized to *UBQ10* expression, a constitutively expressed gene that is used as an internal control in *Arabidopsis* (Ramon *et al*, 2019; Belda-Palazon *et al*, 2020; Chantarachot *et al*, 2020). Relative expression levels were determined as described previously (de la Torre *et al*, 2013).

### Bioinformatics

*In silico* analysis of subcellular protein localization was performed using SUBA4 (Hooper *et al*, 2017). Prion-like domains were identified using PLAAC (Lancaster *et al*, 2014), with the minimum length for prion domains Lcore = 60, organism background *Arabidopsis*, and the parameter $\alpha = 1$. The RNA-binding proteins were predicted by the RNApred tool (Kumar *et al*, 2011). The prediction approach was based on amino acid composition, and the threshold for the support vector machine (SVM) was 0.5. To retrieve protein–protein interaction data, we used STRING database (V10) (Szklarczyk *et al*, 2015). Only physical protein–protein interactions were considered. Per-residue disorder content was evaluated by PONDR predictors, including PONDR-FIT (Xue *et al*, 2010) and PONDR-VSL2 (Peng *et al*, 2005). The intrinsic disorder propensities of TSN were evaluated according to the previously described method (Santamaria *et al*, 2017; Uversky, 2017). Disorder evaluations together with disorder-related functional information were retrieved from the $D_2P_2$ database (http://d2p2.pro/) (Oates *et al*, 2013). Intrinsically disordered regions were predicted using Iupred2A (Erdos & Dosztanyi, 2020). LLPS predisposition was evaluated using the PSPredictor tool (preprint: Sun *et al*, 2019). The image analysis was done using ImageJ version 1.52 software (http://rsb.info.nih.gov/ij/index.html). SGs were scored as positive when they had a minimum size of 0.5 µm. SG counting was performed manually with the Cell Counter plugin of ImageJ (http://rsbweb.nih.gov/ij/ plugins/cell-counter.html).

## Data availability

The mass spectrometry data from this publication have been deposited to the JPOST repository. For TSN and RBP47 interactomes, the dataset identifiers are JPST000766 (https://repository.jpostdb.org/entry/JPST000766) and JPST001103 (https://repository.jpostdb.org/entry/JPST001103), respectively.

**Expanded View** for this article is available online.

## Acknowledgements

We thank Filip Rolland (Katholieke Universiteit Leuven) for providing *snrk1α1*$^{-/-}$ *snrk1α2*$^{-/+}$ mutant, Tsuyoshi Nakagawa (Shimane University) for providing Gateway binary vectors, Elena Baena-Gonzalez (Instituto Gulbenkian de Ciência, Portugal) for providing *SnRK1α1* cDNA and critical discussion and Glenn Hicks (University of California, Riverside) for critical reading of the manuscript. This work was supported by the European Commission (MSCA-IF-ReSGulating- 702473), the Ministerio de Economía y Competitividad (Juan de la Cierva-Incorporacion, IJCI-2016-30763) and University of Seville (VIPPIT-2020-IV.4 and -I.5) to E.G.-B, by the Ministerio de Economía y Competitividad (PGC2018-099048-B-100) to J.L.C., by grants from Knut and Alice Wallenberg Foundation (2018.0026), the Swedish Research Council VR (2019-04250_VR), the Swedish Foundation for Strategic Research (RBP14-0037), and by Crops for the Future Research Programme to P.V.B.

## Author contributions

Conceptualization, EG-B and PVB; Methodology, EG-B, PNM, VNU, JLC, and PVB; Investigation, EG-B, PHE, GWDII, and VNU; Writing—original and revised manuscripts, EG-B and PVB; Writing—review and editing, EG-B, KD, PNM, VNU, JLC, and PVB; Funding Acquisition, EG-B, JLC, and PVB.

## Conflict of interest

The authors declare that they have no conflict of interest.

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
