## [Review Process File · The EMBO Journal]

Tudor Staphylococcal Nuclease is a docking platform for stress granule components and is essential for SnRK1 activation in Arabidopsis

Emilio-Gutierrez Beltran, Pernilla H. Elander, Kerstin Dalman, Guy W. Dayhoff II, Panagiotis N. Moschou, Vladimir N. Uversky, Jose L. Crespo, and Peter V. Bozhkov
DOI: [10.15252/embj.2020105043](https://doi.org/10.15252/embj.2020105043)

Corresponding authors: Emilio Gutierrez Beltran (egutierrez@us.es) , Peter Bozhkov (peter.bozhkov@slu.se)

Review Timeline:

Submission Date:	26th Mar 20
Editorial Decision:	7th May 20
Revision Received:	2nd Jun 20
Editorial Decision:	23rd Jun 20
Revision Received:	30th Mar 21
Editorial Decision:	9th Jun 21
Revision Received:	23rd Jun 21
Accepted:	1st Jul 21

Editors: Ieva Gailite / Stefanie Boehm

Transaction Report:

(Note: Depending on transfer agreements, referee reports obtained elsewhere may or may not be included in this compilation. Referee reports are anonymous unless the Referee chooses to sign their reports.)

Dear Dr. Bozhkov,

Thank you for submitting your manuscript for consideration by The EMBO Journal. We have now received a full set of reviewer reports on your manuscript, which are included below for your information. Based on the reviewer comments, we unfortunately had to conclude that the study is not a sufficiently strong candidate for publication in The EMBO Journal.

As you can see, while reviewer #1 is more positive in their assessment, reviewers #2 and #3 indicate substantive concerns regarding the experimental approach in identification of TSN interactome, and find that substantial additional experiments would be required to validate the dataset and the proposed model of stress granule dynamics and to provide further insights into how TSN regulates SnRK1 function. Based on these evaluations by good experts in the field, I am afraid that we cannot offer further proceedings towards publication in The EMBO Journal.

That being said, I realise that the study will be of interest to other researchers in the field. Therefore I would like to propose a transfer of your manuscript and referee reports to our sister journal Life Science Alliance. I have taken the liberty to discuss your work with Andrea Leibfried, executive editor of Life Science Alliance, and Andrea would like to offer publication of your manuscript after a revision. Andrea indicates that, in addition to a full point-by-point response, the drawbacks of the proteomics pipeline should be openly discussed and the conclusions toned down accordingly, and also answer diligently to all points raised by the reviewers. Further statistical evaluation of the existing proteomic data to some extent and for the IDP analyses would have to be added to address the issues raised by reviewers #2 and #3 in more detail. Andrea would ask reviewers #1 and #3 to re-review such a revised version, explaining the transfer situation and the revision scope that is expected for publication in Life Science Alliance. Andrea would be happy to discuss the process and the scope of the revision at any point - you can contact her at a.leibfried@life-science-alliance.org.

Thank you in any case for the opportunity to consider this manuscript. I am sorry that I could not communicate more positive news, but I nevertheless hope that you will find our reviewers' comments helpful and that you will find the transfer option of interest. Please also note that the transfer link does not have an expiry date, and you can return to this option at a later time point.

With kind regards,

Ieva Gailite, PhD
Editor
The EMBO Journal

Referee #1:

TSN is a Docking Platform for Stress Granule Components and is Essential for SnRK1 Activation. Gutierrez-Beltran..[.].Bozhkov. EMBO J submission 3_20

General Summary and Opinion:

The cytosolic non-membraneous compartments termed stress granules are complexes of proteins and RNAs that become visible by confocal microscopy under a number of extreme stresses in eukaryotic cells, including plants. These macromolecular assemblies are generally associated with the dampening of mRNA translation. They can be found in close proximity to processing bodies that are enriched in the apparatus responsible for mRNA turnover initiated through removal of the protective 5' cap. A number of proteins that form stress granule assemblies are known; a general characteristic of these proteins is regions of high structural disorder. This study addresses the heterogeneity and dynamics of the stress granule compartments enriched with the protein called TSN in *Arabidopsis*.

The authors use a dual tag affinity purification scheme followed by proteomics to identify TSN2 associated proteins under two conditions, non-stress (NS) and a short term high temperature (HS) treatment or salt stress treatment. They validate observations by use of co-localization and co-immunoprecipitation. They identify proteins that are constitutively associated with TSN2 as well as proteins that become associated by HS or salt. They also compare their proteomic data with known stress granule proteomes from mammals, yeast and *Arabidopsis*. The latter is the proteins that co-purify with the stress granule protein Rbp47b of *Arabidopsis* (Kosmacz et al., 2019). The TSN2 NS, TSN2 HS and Rbp47b networks overlap by only 7-21%, with decapping complex proteins missing from the TSN2 co-purified proteins, further emphasizing the heterogeneity of these complexes. In contrast, the proteins that associate with TSN1 and TSN2 were found to be highly similar. Notably, this conclusion would be more robust if the Rbp47b network was defined in the same study. The authors make considerable effort to provide support for TSN2 co-localization with proteins using confocal imaging in transiently transfected *N. benthamiana* protoplasts or intact leaves and for shared TSN1 and TSN2 interactors, using stable transgenics.

Based on the protein interactome and protein co-localization analyses the study concludes that TSN acts as a scaffold critical for stress granule assembly. The highly disordered N-terminal SN repeat portion of the protein is shown to be required for assembly of large complexes with several co-purifying proteins. This part of the study lacks a confocal evaluation of the effect of a *tsn1 tsn2* mutant on the assembly of granules with selected co-purifying proteins. It cannot be excluded that the multiple disordered proteins, including Ubp1 and Rbp47, synergize the assembly of these macromolecular complexes.

The authors conclude that TSN interacts with 11 proteins to form a core complex under non-stress conditions. It is reasonable to consider that there is a "seed" for stress granules, but the assertion that TSN serves as the scaffold will require significant additional experimental analysis. For example, how is the interactome of Rbp47b, which overlaps with TSN2 affected in a *tsn1 tsn2* double mutant? A similar question could be asked for UBP1 proteins. It is suggested that the authors dial back on their conclusions about the importance of TSN1/2 as a seed, as this would not detract from their finding of the core complex.

A second finding is that the TSN interactome includes the energy sensing kinase SnRK1.2. This is fascinating as Kosmacz et al. (2019) reported that Rbp47b co-purifies with the distinct SnRK2.1, involved in ABA response signaling. Here, the authors apply a reasonable strategy to obtain secondary validation of heat-stress enhanced SnRK1.2 association with TSN in stress granule foci/macromolecular assemblies. One of the more convincing pieces of evidence is the observation is inhibited by cycloheximide. This part of the analysis is done with time resolution. The findings would have been even more convincing if the interaction between TSN and SnRK1.2 was

monitored by co-IP over this timecourse. But the comparison of the pT175 in the presence/absence of TSN2 in Fig. 7E and F is convincing. Thus, the data support the conclusion that SnRK1.2 phosphorylation, a signature of its activation, is enhanced following heat stress by its interaction with TSN2. This finding can be followed by future work to better understand the functional significance of this association, what drives kinase activation or limits SnRK1.2 dephosphorylation in these complexes. ABA also includes SnRK1 T175 phosphorylation, encouraging future consideration of whether this in some way driven by interaction with the TSNs.

In summary, this submission provides new information about complex, heterogeneous and dynamic protein assemblies found in the plant cytoplasm that are highly analogous to complexes described for other eukaryotic models. The study provides new insights, but in many places a more cautious interpretation of the data is needed.

Specific Concerns Essential to be addressed

Abstract and Introduction

Lines 40-41. Delete "profound" and "high"; unnecessary hyperbole. There are other places where there is unnecessary use of adjectives and adverbs.

Line 48: constantly or constitutively?

Line 94: Why is Kosmacz et al., 2019 not mentioned? What about information regarding the role of these proteins in rice endosperm development?

Line 129: unclear what is meant by a "form an 'existing dense' interaction network". I think the authors mean that this network is partially assembled. Why is it called "dense".

The introduction does not provide information on the spatial distribution of the proteins and complexes discussed. Are these all in the cytosol? The introduction does not provide any perspective on the two TSNs of Arabidopsis. Is TSN1 functionally equivalent to TSN2, for example?

Results:

Lines 151/155: More details of the growth conditions, plant age, heat stress conditions, and region/cells of the root viewed is needed.

Line 168: Is the same growth/age/stress conditions used for the imaging?

Figure 1: Is it the nucleus the black negative space the cells in following the HS? This panel seems very different.

Line 178: Is there a reference for the statement of well-known SG proteins? Kosmacz et al., 2019?

Line 183: presumptively physiologically relevant?

Line 197-198: comment about lack of DCP and XRN as PB components, but isn't VCS a considered a scaffold of DCPs? This is addressed on line 458 of the discussion, but highlights that protein bodies are also heterogeneous.

Figure 2 legend. Provide citation for the SG proteomes of yeast and humans. Were these defined under stress or non-stress conditions? Make clear if Figure 2 focuses on "stress".

The Rbp47b interactors were defined under heat/ darkness conditions, correct. The text could do a better job explaining that the first analysis is of "stress" interactomes.

Can the authors comment on the observation of different ribosomal protein isoforms in the HS-sensitive and HS-independent interactors, for example RPS6A and RPS6B?

Line 214 and elsewhere: Please use the standard nomenclature for initiation (eIF) and elongation

(eEF) factors to avoid confusion in text, figures and tables. Recommend that proteins such as Rbp47 and Ubp1 be written in capitals based on Arabidopsis convention and for consistency.

Line 213-216: Can the authors provide an interpretation of their findings? Why might the eIF4 subunits be enriched under HS, for example?

Line 218 paragraph: Figure 2D can be more valuable. The network presented is not informative because there are no genes indicated and the relationship between the two networks is obscure. Colors could be used to highlight what is constitutive, such as showing TSN2, Rbp47, Ubp1 and PAB4. A high resolution PDF of these networks should be provided in the supplement.

Line 249-250: The authors concluded that the high disorder characteristic of TNS2 associated proteins supports the conclusion that TSN is a scaffolding factor. An experimental demonstration of required is to reach this conclusion. How can one be certain that Rbp47b or Ubp1 is not what seeds assembly of the complex? The statement from 264-266 is too speculative.

Line 316: CTT can be defined. Related to this, the protein "groups" are not explained in the text for figure 5A and 6A.

Line 353-359 and Figure 6: Was a different root region used for the NS and HS evaluation of foci? Text unclear. Is the HS region more apical? These results are fascinating. Was a time course or concentration gradient of NaCl used? Perhaps the foci for TCTP and SnRK1.2 reflect a more severe stress.

Discussion: The discussion is generally very informative. It does not repeat the results and might also be more cautious with conclusions.

Line 478: The authors should consider the recent findings of Van Leen et al., Nature Plants. doi: 10.1038/s41477-019-0378. Less related, the authors might consider the work of Kawa et al. Plant Physiology. doi: 10.1104/pp.19.00818.
doi: 10.1104/pp.19.00818.

Methods:

The methods appear to lack a description some of the stable Arabidopsis genotypes developed for this study, i.e., as described starting on line 322, the GFP-RH12. It is unclear from the methods when the 35S or a native promoter was used. Details of the constructs and the generation of the plants used should be mentioned in an appropriate location (methods or supplemental table).

Line 607: Add the method used to heat stress N. benthamiana plants or leaves.

Minor Concerns that should be Addressed

Lines 40-41. Delete "profound" and "high"; unnecessary hyperbole. There are other places where there is unnecessary use of adjectives and adverbs.

Line 48: constantly or constitutively?

Datasets:

These are well formatted. It was not possible for me to evaluate the quality of the proteomic analyses from what was provided. Presumably the mass spectrometry data will be cataloged.

Abbreviations: There are a number of places where abbreviations are not spelled out. The

abbreviation MLO is not needed; its unusual and only used once again in the text.

Referee #2:

Comments to the authors

I have read the manuscript with great interest and unfortunately I feel that it is connected to several weaknesses.

Major problems (random order):

1) The authors present careful proteomic investigations of TSN interactors in stressed and non-stressed conditions. However, it is clear that they have not reached saturation in their analysis (eg indicated that different ribosomal proteins are identified at different conditions). This is easily explained by the proteomic approach used. They have used DDA acquisition which means that in a given time only the most abundant ions are taken for fragmentation. Abundance in this sense has a poor connection to the abundance of the peptide in the sample; the peptide sequence is of great importance here as well. Some peptides "fly" better than others in the mass spec and therefore are easier to detect. As only a given number of peptides are selected for identification at each given time, this top list is heavily influenced by the flyability of peptides. Thus a protein that is detected in sample 1 can be at equal concentration (or even higher) in sample 2 without being detected as the protein was kicked out of the top list by the appearance of other peptides that were more abundant. Therefore, the approach used here opens up for a lot of misinterpretation of the data. Especially since that the authors is dependent on "presence" and "absence" in the different samples. This risk is even bigger in a situation where the lists are far from saturation - this random effect has a huge impact. There are better proteomics approaches used that don't select the "abundant" peptides/ions that would have served this study better.

2) The authors make a big thing about finding SnRK1 among their interactors. However, they present no mechanistic model of how SnRK1 contributes to stress signaling more than that it is part of the interesting stress granule. It is just picking the most famous protein out of a quite long list of proteins and we as readers get very little understanding on how this affects downstream processes. In meetings, Snrk1 has been reported to interact with nearly all parts of the plant cell (mitochondria, chloroplast, nuclear, also chromatin) so I'm not so surprised to see it in the stress granule as well. I wanted to know how this stress granule/TSN interaction affects downstream SnRK1 processes.

3) We do not know how many of these interactions are direct. It might be that the authors have found a good way of pulling down SG and loosely connected factors. For some of the candidates, the authors show different forms of microscopical localization studies. I would have wanted biochemical proof from crosslinking etc. That could also have identified more direct interactors and served as an unbiased way to determine direct interactors. Now we rely on the ones the authors have tested (and we do not know if some were tested and show no interaction).

4) The promoter of the TAP tagging construct is not clearly described in the results or the material and methods section. If you check the reference it is clear that the authors have used a very strong constitutive over-expressor construct (promoter is 2 times 35S). This is of course favoring the interpretation of the interaction studies. The author might have filled the cell with TSN and identified interactions that will not happen if the TSN is expressed at normal level.

Minor comments:

The introduction don't clearly distinguish when they are describing plant proteins processes from when they are discussing animal/yeast protein. This makes it confusing eg is single gene knockout of TSN not lethal in plants (line 98). I know it but a less read reader might get either wrong impression.

The connection to IDPs is weak (rise from 93% to nearly 100%) although statistically significant. And not supported by experimental data. It is also not important for the rest of the reasoning so this part can be omitted.

Figure 4: In A the group three is supposed to consist of protein confirmed I figure 4, However most are not mentioned in figure 4? What other reasons justifies this group? If something else it should be mention in legend and the comment in the figure should be corrected.

That "CHX treatment abrogated heat-induced phosphorylation of SnRK1 T175" does not suggest anything more than that protein synthesis is needed for the phosphorylation. Which might indicate that the effect is indirect (require a round of transcription/translation of a new protein), but might as well indicate that a short lived protein is required. Many proteins have natural short half-lives and therefore is the use of CHX and other protein inhibitors hard to interpret. CHX treatments cause changed expression of thousands of genes so there are many other possible interpretations.

Referee #3:

The manuscript by Gutierrez-Beltran et al., focuses on the TSN proteins to investigate Arabidopsis stress granules. The study makes the interesting finding that the phosphorylation of SnRK1 that occurs in response to heat shock, depends upon TSN1 and TSN2 function.

The study begins with the proteomic analysis of proteins co-purifying with TSN2 by using a TAP-tagging approach. Although the tagged protein is shown by microscopy to associate with presumed stress granules, the functionality of the tagged protein is not assessed by complementation of the *tsn1tsn2* double mutant (L155). The biological replicates used here aren't clearly presented - but there is more detail in the methods section (L571). However, there is no statistical analysis of the data - for example to reveal the significance of peptide enrichment by different protein baits and in different conditions. The description of the proteomics results at L173/174 is ambiguous - "with up to 4490 hits". The data is then filtered by comparison to GFP controls and sub-cellular localisation. The bioinformatic analysis includes a study of publicly available protein-protein interaction data (L218), but the source of this is not cited, and the significance of the conclusions here is unclear (Fig 2D). In addition there is an analysis of the proteins for disordered regions. However, with arbitrary cut-offs of weak/moderate/high and apparently 93% of the Arabidopsis proteome containing disordered regions (L240), it is unclear what insight is produced here (Figure 3A-C). Protein interactions are validated using BiFC by possibly over-expressing (the expression levels are not presented) test proteins transiently in a different experimental species - *Nicotiana benthamiana* (which is not mentioned in the Results section). However, BiFC data alone is insufficient not to conclude as stated on L286 that these proteins interacted with TSN2.

Overall, I find the proteomics section relies heavily on previously established concepts and not enough on unbiased analysis of the data produced here.

The authors then identify protein partners of TSN1 and compare heat and salt stress. Interestingly

they detect an association with the kinase SnRNK1. They show that the heat stress-dependent phosphorylation of SnRNK at T175 is dependent on TSN1 and 2. However, the presentation of the data here is problematic. The western blots are separated such that direct comparisons between conditions cannot be made Fig 7C-F.

In the Discussion the authors describe a model L500-508 that they say is based on their results. However, there is no data in the present manuscript that corresponds to the 3 steps they propose. Instead, this largely reflects the output of the stress granule field as a whole over many years.

Overall, I have reservations about the quality of the proteomics analysis and the new insight it reveals. I found the connection to SnRNK1 to be interesting, but it comprises a modest part of the manuscript and the functional significance is not examined. The phenotype of *tsn1tsn2* double mutants with respect to stress is not commented on/cited here.

Other comments

1. Only 1 side of error bars are shown (dynamite plots) both sides should be visible
2. Some Venn diagrams are scaled to the numbers involved, but others are not
3. In Fig 6c the lanes probed with anti-TSN and anti-GFP antibodies do not line up correctly.
4. · Figure S1 + lines 153-155: Colocalization with known SG markers, should be presented to be sure that indeed TSN2-TAPa is redistributed to SGs upon heat stress.
5. · Lines 154-155: SGs can be formed in various stress conditions. Confirmation of heat stress induction by at least showing increased expression of stress markers should be done to be sure that proper stress induction was achieved.
6. · Figure S3: What are the various bands detected in GFP-TAPa input control?
7. Figure S4 and lines 173-175: A large number of background interactors are detected with GFP alone, raising the possibility that the purification is insufficiently stringent.
8. · Figure 4D and lines 284-287: BiFC experiments use a different species and do not include colocalization with known SG marker. From colocalization studies TSN2 is not exclusively present in SGs (there are spots that do not colocalize with UBP1 on Figure 4B). Are interactions between TSN2 and 10 selected proteins occurring in SGs or in the cytosol?
9. · Figure 5C and lines 322-331: colocalization with SG marker would address if observed cytoplasmic foci are SGs or not.
10. · Figure 6D: Colocalization of proteins is missing - but this would help test if they are present in the same foci.
11. · Lines 393-395: Stress induction confirmation should be shown
12. · Materials & Methods: Heat stress and salinity stress induction should be described; procedure for CHX treatment has not been shown in the paper.

Re: Revision plan for the manuscript EMBOJ-2020-105043 “TSN is a Docking Platform for Stress Granule Components and is Essential for SnRK1 Activation”.

Dear Editor, Dear Reviewers,

We highly appreciate constructive comments on our manuscript and are game for addressing them explicitly by performing additional experiments and corresponding amendments in the manuscript. We are confident that by implementing the below-described revision plan we will achieve substantial strengthening of three main novel messages conveyed by our study:

- I. Analysis of TSN interaction networks reveals composition of plant stress granules under various conditions.
- II. TSN is a protein assembly platform of stress granules.
- III. TSN confers activation of SnRK1 under heat stress.

Revision plan for each of the three main messages. Description of additional experiments (and already obtained new results) is highlighted yellow.

- I. Tandem affinity purification coupled to mass spectrometry (TAP-MS) is one of the most advanced methods to characterize protein complexes in plants (e. g Van Leene et al., 2010; Henrichs et al., 2012; Van Leene et al., 2015). For MS, we employed the widely used data-dependent acquisition (DDA) workflow (Burdova et al., 2019; Maffei et al., 2020; Nelson et al., 2020), where the most abundant peptide ion precursors from a survey scan (MS1 spectrum) are isolated individually and fragmented resulting in an MS/MS or MS2 spectrum. Although the DDA-LC-MS workflow may have some flaws, we have been able to identify numerous TSN partners under different conditions. Our LC-MS results have been further validated using a set of approaches, including [1] bioinformatic analysis (Figure 2A, 2B and 2C); [2] *in vivo* colocalization study (Figure 4B and 4C); [3] immunoprecipitation using TSN1 and TSN2 (Figure 5B), RBP47b (Figure 6C) or SnRK1.2 as baits (Figure 7A); [4] bimolecular fluorescence complementation (BiFC) (Figure 4D and 4F), and [5] Förster resonance energy transfer (FRET) (Figure 7B).

However, as pointed out by the reviewers, relative quantification of the identified TSN interacting proteins is missing. To address this important point we plan to use the NSAF

(*normalized spectral abundance factor*) method (Liu et al., 2004; Old et al., 2005), which provides results of the highest reproducibility in comparison with alternative methods (McIlwain et al., 2012; Bubis et al., 2017). This method takes protein length into consideration, since a longer protein will inherently have more observable peptides than a smaller one. Thus, we will calculate the NSAF value for each protein that will serve a proxy for its abundance relative to the entire pool of proteins. The NSAF values for proteins from different conditions will be compared using unpaired two sample *t*-test. Thus, proteins with a *p*-value < 0.05 will be considered as differentially expressed (Neilson et al., 2011).

The reviewers see little sense in including the results of intrinsic disorder (ID) analysis of TSN interaction networks compared to entire Arabidopsis proteome. To make ID analysis more relevant to experimental settings we are going to compare ID status of the members of the interactomes of TSN and GFP (control bait in TAP) using a set of disorder predictors, such as PONDR-VSL2 and PONDR-FIT to name a few.

- II. In a previous study, we observed that, similarly to mammalian systems, Arabidopsis SGs are dynamic structures, since SG components showed a rapid recovery after fluorescence recovery after photobleaching (FRAP) (Gutierrez-Beltran et al., 2015). On the contrary, TSN did not exchange between the cytoplasm and foci, suggesting a role as a scaffolding protein. In addition, TSN was required for the maintenance of the structural integrity of SG protein complexes, indicating its role in the SG maturation (Gutierrez-Beltran et al., 2015). In the present work we have found that TSN interacts with a large number of known SG proteins both prior to and during stress, strengthening its role as a platform for SG assembly.

To reinforce the role of TSN as a SG scaffolding factor, we will, as suggested by the reviewers, compare the interactome of RBP47b, an evolutionarily conserved SG marker protein, in Wild-type (Wt) and *tsn1tsn2* plants under no stress (NS) and heat stress (HS) conditions using *GFP-RBP47* expressing lines available in the lab. These experiments will also provide more adequate dataset for comparison with TSN interactome, as suggested by the reviewer 1.

- III. To further explore the functional implication of the activation of SnRK1 proteins after localization to heat-induced SGs, we have set up a series of additional experiments in collaboration with Dr. Elena Baena-Gonzalez, an expert in SnRK1 signalling (Baena-Gonzalez et al., 2007; Rodrigues et al., 2013; Baena-Gonzalez and Lunn, 2020). Most of our current understanding of plant SnRK1 signalling has been achieved by studying the activation of their direct targets, such as DARK INDUCED6 (DIN6) (Baena-Gonzalez et al., 2007; Dietrich et al., 2011; Ramon et al., 2019). In our study we observed that both SnRK1 proteins are phosphorylated after localization to heat-induced SGs.

To get an explicit proof for the activation of SnRK1 signalling pathway, we have firstly compared the transcriptome profile of SnRK1 activation (provided by Elena Baena-Gonzalez; Baena-Gonzalez et al., 2007) with that associated with SG formation (Gutierrez-Beltran et al., 2015). We have found a significant overlap between the transcriptome changes triggered by SnRK1 and by heat-stress induced formation of SGs, including differential expression of previously characterized SnRK1 reporters (Baena-Gonzalez et al., 2007; Ramon et al., 2019). We have further revealed that the corresponding

transcriptome changes were abrogated in *tsn1tsn2* mutant, indicating that TSN is required for the activation of SnRK1 signaling during HS. The results of this *in silico* approach will be further substantiated by two functional experiments. In one experiment we will study the promoter activation of DIN6 (pDNI6) gene *via* transient expression of LUC-pDIN6 construct [provided by Dr. Filip Rolland (Ramon et al., 2019)] in protoplasts of different backgrounds, including Columbia (Col) Wt, *tsn1tsn2* (knockout line), *SnrK1.1/SnrK1.2* [homozygous for *SnRK1.1* and heterozygote for *SnRK1.2*; provided by Dr. Rolland (Ramon et al., 2019)] and *tsn1tsn2 pTSN2* (*TSN2* complemented line). In another experiment, we will analyse activation of several SnRK1 target genes via qRT-PCR using Arabidopsis seedlings with above-described backgrounds. To confirm the specificity of the SnRK1 signalling activation, both experiments will be performed under different conditions, including NS, HS, HS-CHX (treated with CHX) or NaCl.

To investigate whether phosphorylation of SnRK1 is required for its SG localization, we will analyse subcellular localization of inactive mutants SnRK1.1^{T175A} (phosphorylated T-loop mutant) and SnRK1.1^{K48M} (kinase-dead, ATP binding site mutant), as well as separate domains SnRK1.1^{CD} (N terminal kinase catalytic domain) and SnRK1.1RD (C terminal regulatory domain) under control and HS conditions.

Finally, we will figure out if the presence of TSN is required for the SG localization of SnRK1 using *tsn1tsn2* lines expressing SnRK1.1 or SnRK1.2 fused to a fluorescent protein available in the lab.

Besides major additional experiments described above we are going to make extensive revision of the whole manuscript that will address all other (minor) comments raised by the reviewers (viz. textual comments, data presentation, statistics and additional controls). We estimate the whole work will require approximately 5 months provided we have access to our labs, i.e. no further covid-19 outbreak.

Thank you very much for your consideration.

Yours sincerely,

Peter Bozhkov
On behalf of co-authors

References

- Baena-Gonzalez E, Lunn JE** (2020) SnRK1 and trehalose 6-phosphate - two ancient pathways converge to regulate plant metabolism and growth. *Curr Opin Plant Biol* **55**: 52-59
- Baena-Gonzalez E, Rolland F, Thevelein JM, Sheen J** (2007) A central integrator of transcription networks in plant stress and energy signalling. *Nature* **448**: 938-942
- Bubis JA, Levitsky LI, Ivanov MV, Tarasova IA, Gorshkov MV** (2017) Comparative evaluation of label-free quantification methods for shotgun proteomics. *Rapid Commun Mass Spectrom* **31**: 606-612
- Burdova K, Yang H, Faedda R, Hume S, Chauhan J, Ebner D, Kessler BM, Vendrell I, Drewry DH, Wells CI, Hatch SB, Dianov GL, Buffa FM, D'Angiolella V** (2019) E2F1 proteolysis via SCF-cyclin F underlies synthetic lethality between cyclin F loss and Chk1 inhibition. *EMBO J* **38**: e101443
- Dietrich K, Weltmeier F, Ehlert A, Weiste C, Stahl M, Harter K, Droge-Laser W** (2011) Heterodimers of the Arabidopsis transcription factors bZIP1 and bZIP53 reprogram amino acid metabolism during low energy stress. *Plant Cell* **23**: 381-395
- Gutierrez-Beltran E, Moschou PN, Smertenko AP, Bozhkov PV** (2015) Tudor staphylococcal nuclease links formation of stress granules and processing bodies with mRNA catabolism in Arabidopsis. *Plant Cell* **27**: 926-943
- Henrichs S, Wang B, Fukao Y, Zhu J, Charrier L, Bailly A, Oehring SC, Linnert M, Weiwad M, Endler A, Nanni P, Pollmann S, Mancuso S, Schulz A, Geisler M** (2012) Regulation of ABCB1/PGP1-catalysed auxin transport by linker phosphorylation. *EMBO J* **31**: 2965-2980
- Liu H, Sadygov RG, Yates JR, 3rd** (2004) A model for random sampling and estimation of relative protein abundance in shotgun proteomics. *Anal Chem* **76**: 4193-4201
- Maffei B, Laverriere M, Wu Y, Triboulet S, Perrinet S, Duchateau M, Matondo M, Hollis RL, Gourley C, Rupp J, Keillor JW, Subtil A** (2020) Infection-driven activation of transglutaminase 2 boosts glucose uptake and hexosamine biosynthesis in epithelial cells. *EMBO J* **39**: e102166
- McIlwain S, Mathews M, Bereman MS, Rubel EW, MacCoss MJ, Noble WS** (2012) Estimating relative abundances of proteins from shotgun proteomics data. *BMC Bioinformatics* **13**: 308
- Neilson KA, Mariani M, Haynes PA** (2011) Quantitative proteomic analysis of cold-responsive proteins in rice. *Proteomics* **11**: 1696-1706
- Nelson ME, Parker BL, Burchfield JG, Hoffman NJ, Needham EJ, Cooke KC, Naim T, Sylow L, Ling NX, Francis D, Norris DM, Chaudhuri R, Oakhill JS, Richter EA, Lynch GS, Stockli J, James DE** (2020) Phosphoproteomics reveals conserved exercise-stimulated signaling and AMPK regulation of store-operated calcium entry. *EMBO J* **39**: e104246
- Old WM, Meyer-Arendt K, Aveline-Wolf L, Pierce KG, Mendoza A, Sevinsky JR, Resing KA, Ahn NG** (2005) Comparison of label-free methods for quantifying human proteins by shotgun proteomics. *Mol Cell Proteomics* **4**: 1487-1502
- Ramon M, Dang TVT, Broeckx T, Hulsmans S, Crepin N, Sheen J, Rolland F** (2019) Default Activation and Nuclear Translocation of the Plant Cellular Energy Sensor SnRK1 Regulate Metabolic Stress Responses and Development. *Plant Cell* **31**: 1614-1632
- Rodrigues A, Adamo M, Crozet P, Margalha L, Confraria A, Martinho C, Elias A, Rabissi A, Lumbreras V, Gonzalez-Guzman M, Antoni R, Rodriguez PL, Baena-Gonzalez E** (2013) ABI1 and PP2CA phosphatases are negative regulators of Snf1-related protein kinase1 signaling in Arabidopsis. *Plant Cell* **25**: 3871-3884
- Van Leene J, Eeckhout D, Cannoot B, De Winne N, Persiau G, Van De Slijke E, Vercruysse L, Dedecker M, Verkest A, Vandepoele K, Martens L, Witters E, Gevaert K, De Jaeger G** (2015) An improved toolbox to unravel the plant cellular machinery by tandem affinity purification of Arabidopsis protein complexes. *Nat Protoc* **10**: 169-187
- Van Leene J, Hollunder J, Eeckhout D, Persiau G, Van De Slijke E, Stals H, Van Isterdael G, Verkest A, Neiryneck S, Buffel Y, De Bodt S, Maere S, Laukens K, Pharazyn A, Ferreira PC, Eloy N, Renne C, Meyer C, Faure JD, Steinbrenner J, Beynon J, Larkin JC, Van de Peer Y, Hilson P, Kuiper M, De Veylder L, Van Onckelen H, Inze D, Witters E, De Jaeger G** (2010) Targeted interactomics reveals a complex core cell cycle machinery in Arabidopsis thaliana. *Mol Syst Biol* **6**: 397

Dear Peter,

Thank you for contacting me with a preliminary revision plan for your manuscript. I have now received broadly positive assessments of your revision proposal from reviewers #1 and #2, and they are interested in evaluating the revised manuscript. I have included their comments below.

Based on these evaluations, I would like to invite you to submit a revised manuscript in which you address the comments of all reviewers. Please note that we will ultimately require strong support from the reviewers for publication here.

While we generally allow three months as standard revision time, we will set the revision deadline to six months to allow for a major revision. I should add that it is The EMBO Journal policy to allow only a single major round of revision and that it is therefore important to resolve the main concerns at this stage. As a matter of policy, competing manuscripts published during this period will not negatively impact on our assessment of the conceptual advance presented by your study. However, please contact me as soon as possible upon publication of any related work to discuss how to proceed.

Please feel free to contact me if have any further questions regarding the revision. Thank you for the opportunity to consider your work for publication. I look forward to your revision.

With best regards,

leva

leva Gailite, PhD
Editor
The EMBO Journal

Reviewer #1:

In my opinion the authors should be encouraged to move forward with their plan and then resubmit a revised manuscript.

Reviewer #2:

Comments on the revision plan supplied by the authors:

1) Here the authors will apply a better-suited proteomics technique and the result will be highly dependent on the coverage. If as I suspected that they don't have full coverage using DDA (many proteins in their samples are not detected) then the suggested method will detect fewer bit with

greater accuracy. This may work, but may also result in only a few proteins quantified, and which proteins they detect might be determined by chance. A better method would be to express their bait to lower levels in the plant. If they do that, I think they will get fewer proteins in their precipitations, but these might be more relevant.

2) More analysis of the mutants would benefit the story and place the protein in a biological context, however it will not answer the question on how the protein acts. The authors favor the typical researcher bias best summarized by "my protein is the most important". Even a protein with no regulatory activity can cause dramatic phenotypes when mutated.

3) Comparing to other researcher's data is usually good, but the dataset they want to compare to is not a very good one. It is only two replicates and the samples are taken from overexpression in protoplasts. To get good data, I suggest they compare with experiments done side by side in the same lab and that will most likely give better data. To be sure about the phosphorylation, they need to identify the site and mutant the SNrk1 target site. The experiments suggested will give indications, but not prove direct interaction. CHX treatments give noisy data as the drug causes massive transcriptional and proteomic changes by itself, often with a bigger effect than the one you want to compare with. Here the quality of the result depends on how the researchers are able to compile a convincing list of results that taken together will form a good story. I think this might work. This will of course only confirm the SnrK link, not answer the "landing platform regulatory super hub theory", but might be much more interesting in the end.

So in total - the authors' suggestions are good, but if it is in the end good enough got EMBO journal no-one knows, but I am curious on their future results.

We wish to thank the reviewers for their constructive criticism and useful comments on our manuscript. We have performed a large number of additional experiments and thoroughly revised the manuscript by including new figures and tables, as well as by making extensive textual amendments. The additional experiments resulting in new or revised sets of data are summarized in the table below:

Additional experiments	New or revised datasets
New analysis of raw MS data from TSN-TAPa experiments using MaxQuant iBAQ	Dataset EV1
Isolation and analysis of RBP47 interactomes from WT and tsn1 tsn2 under no stress (NS) and heat stress (HS) conditions	Dataset EV3 and Appendix Fig. S3
Comparison of TSN and RBP47 interactomes isolated in our study	Fig. 2B and Dataset EV2
New evaluation of IDRs and LLPS parameters in TSN2 interactomes by IUPred and PSPpredictor algorithms using GFP interactomes as controls	Fig. 4A, B
Immuno co-localization analysis of TSN2-TAPa and the SG marker eIF4E	Fig. EV1C
Complementation assay of TSN2-TAPa in the tsn1 tsn2 background under long-term HS treatment (analysis of cell death phenotype).	Fig. EV1D
Protein-protein interaction analysis of TSN2 and RBP or TCTP by Co-IP and FRET	Fig. EV4C, D
Co-localization analysis of three novel TSN-interacting proteins with RBP47 and eIF4E, two SG markers, in Arabidopsis .	Fig. EV4B and Appendix Fig. S6
qPCR analysis of the heat stress-induced genes HSP101 and HSF in WT, tsn1 tsn2 and snrk1α1^{-/-} snrk1α2^{+/-} backgrounds	Appendix Fig. S1
New analysis of protein-protein interaction networks within TSN2_NS and TSN2_HS protein pools	Appendix Fig. S2
New set of experiments exploring the mechanistic connection between SnRK1α and TSN in the context of SG assembly, including:	
(i) Protein-protein interaction analysis of TSN and SnRK1α isoforms by Co-IP and FRET	Fig. 7A-C
(ii) Quantification of SnRK1α foci number and size in WT and tsn1 tsn2 backgrounds upon HS	Fig. 7E, F
(iii) Analysis of the SnRK1α mobility in WT and tsn1 tsn2 backgrounds after HS by FRAP	Fig. 7G, H
(iv) Localization study of individual catalytic and regulatory domains of SnRK1α1 in N. benthamiana protoplasts and Arabidopsis plants	Fig 7J; EV5B-E
(v) Analysis of the kinetics of SnRK1α foci formation in WT during HS	Fig. 8C, D
(vi) Quantification of gene expression levels of two target genes of the SnRK1-dependent signalling pathway by qPCR in heat-stressed WT, tsn1 tsn2 , tsn1 tsn2;TSN2 and snrk1α1^{-/-} snrk1α2^{+/-} plants	Fig. 8G

Below is a point-by-point response to the reviewers' comments

Referee 1:

TSN is a Docking Platform for Stress Granule Components and is Essential for SnRK1 Activation. Gutierrez-Beltran..[.].Bozhkov. EMBO J submission 3_20

General Summary and Opinion:

The cytosolic non-membraneous compartments termed stress granules are complexes of proteins and RNAs that become visible by confocal microscopy under a number of extreme stresses in eukaryotic cells, including plants. These macromolecular assemblies are generally associated with the dampening of mRNA translation. They can be found in close proximity to processing bodies that are enriched in the apparatus responsible for mRNA turnover initiated through removal of the protective 5' cap. A number of protein that form stress granule assemblies are known; a general characteristic of these proteins is regions of high structural disorder. This study addresses the heterogeneity and dynamics of the stress granule compartments enriched with the protein called TSN in Arabidopsis.

The authors use a dual tag affinity purification scheme followed by proteomics to identify TSN2 associated proteins under two conditions, non-stress (NS) and a short term high temperature (HS) treatment or salt stress treatment. They validate observations by use of co-localization and co-immunoprecipitation. They identify proteins that are constitutively associated with TSN2 as well as proteins that become associated by HS or salt. They also compare their proteomic data with known stress granule proteomes from mammals, yeast and Arabidopsis. The latter is the proteins that co-purify with the stress granule protein Rbp47b of Arabidopsis (Kosmacz et al., 2019). The TSN2 NS, TSN2 HS and Rbp47b networks overlap by only 7-21%, with decapping complex proteins missing from the TSN2 co-purified proteins, further emphasizing the heterogeneity of these complexes. In contrast, the proteins that associate with TSN1 and TSN2 were found to be highly similar. Notably, this conclusion would be more robust if the Rbp47b network was defined in the same study. The authors make considerable effort to provide support for TSN2 co-localization with proteins using confocal imaging in transiently transfected *N. benthamiana* protoplasts or intact leaves and for shared TSN1 and TSN2 interactors, using stable transgenics.

Response 1:

*Thank you very much for the suggestion. We have now isolated RBP47 interactome from WT and *tsn1 tsn2*, both lines expressing GFP-RBP47 and grown under no stress (NS) and heat stress (HS) conditions. The lists of significantly enriched proteins are provided in a new Dataset EV3, whereas the comparison between GFP-RBP47 and TSN interactomes under NS and HS conditions is presented in Dataset EV2 and Figure 2B.*

Based on the protein interactome and protein co-localization analyses the study concludes that TSN acts as a scaffold critical for stress granule assembly. The highly disordered N-terminal SN repeat portion of the protein is shown to be required for assembly of large complexes with several co-purifying proteins. This part of the study lacks a confocal evaluation of the effect of a *tsn1 tsn2* mutant on the assembly of granules with selected co-purifying proteins. It cannot be excluded that the multiple disordered proteins, including Ubp1 and Rbp47, synergize the assembly of these macromolecular complexes.

Response 2:

*Thank you very much for excellent comment. We have now investigated the effect of TSN deficiency (*tsn1 tsn2* mutant) on the assembly of SGs with selected TSN-interacting proteins by analysing size, number and mobility (FRAP) using both SnRK1 α isoforms. Our new results indicate that TSN is required for both dynamics and assembly of SnRK1 α into SGs (see new Fig. 7D-H). Perturbation of scaffold-partners interaction has been described to have a strong effect on the recruitment of partners to membranellles organelles (Banani, Rice et al., 2016). Therefore, our finding is in accordance with the role of TSN as a scaffold.*

We agree with the reviewer that additional disordered proteins other than TSN could contribute to SG assembly. Indeed, studies in mammalian cells have linked various RNA-binding proteins to SG assembly, including TIA1 (Gilks, Kedersha et al., 2004), HDAC6 (Kwon, Zhang et al., 2007), G3BP1 and G3BP2 (Kedersha, Panas et al., 2016), PRRC2C (Youn, Dunham et al., 2018), CSDE1 (Youn et al., 2018), and UBAP2L (Markmiller, Soltanieh et al., 2018). However, molecular comprehension of the role of these proteins and their plant orthologues in SG assembly remains unknown.

The authors conclude that TSN interacts with 11 proteins to form a core complex under non-stress conditions. It is reasonable to consider that there is a "seed" for stress granules, but the assertion that TSN serves as the scaffold will require significant additional experimental analysis. For example, how is the interactome of Rbp47b, which overlaps with TSN2 affected in a *tsn1 tsn2* double mutant? A similar question could be asked for UBP1 proteins. It is suggested that the authors dial back on their conclusions about the importance of TSN1/2 as a seed, as this would not detract from their finding of the core complex.

Response 3:

*Excellent comment! We have now isolated and compared RBP47 interactomes from WT vs. *tsn1 tsn2* plants under NS and HS conditions. TSN deficiency led to severe perturbation in the RBP47 interactome under both conditions (see new Fig. 2). Results of these new experiments together with our previously published FRAP data showing stable association of TSN with SGs (Gutierrez-Beltran, Moschou et al., 2015) provide strong evidence for scaffolding role of this protein in SGs.*

A second finding is that the TSN interactome includes the energy sensing kinase SnRK1.2. This is fascinating as Kosmacz et al. (2019) reported that Rbp47b co-purifies with the distinct SnRK2.1, involved in ABA response signaling. Here, the authors apply a reasonable strategy to obtain secondary validation of heat-stress enhanced SnRK1.2 association with TSN in stress granule foci/macromolecular assemblies. One of the more convincing pieces of evidence is the observation is inhibited by cycloheximide. This part of the analysis is done with time resolution. The findings would have been even more convincing if the interaction between TSN and SnRK1.2 was monitored by co-IP over this timecourse. But the comparison of the pT175 in the presence/absence of TSN2 in Fig. 7E and F is convincing. Thus, the data support the conclusion that SnRK1.2 phosphorylation, a signature of its activation, is enhanced following heat stress by its interaction with TSN2. This finding can be followed by future work to better understand the functional significance of this association, what drives kinase activation or limits SnRK1.2 dephosphorylation in these complexes. ABA also includes SnRK1 T175 phosphorylation, encouraging future consideration of whether this in some way driven by interaction with the TSNs.

Response 4:

We agree with the reviewer that SnRK1-related part of the manuscript is exciting. We have carried out new experiments presented in new Figures 7, 8, and EV5 to deepen this part even further. The data of the revised manuscript demonstrate that:

- (i) TSN-SnRK1 interaction pre-exists prior to stress and is not influenced by HS, which agrees with our proteomic data (Fig. 7B; see also Fig. 3C; Dataset EV1).*
- (ii) TSN interacts in vivo with both SnRK1 α isoforms, as shown by co-IP (Fig. 7A), BiFC (Fig. 3D, 4D, Appendix Fig. S4) or FRET (Fig. 7C).*
- (iii) TSN takes part in the assembly of SnRK1 α isoforms in SGs and is required for full mobility of SnRK1 $\alpha 1$ (Fig. 7E – H, EV5A).*
- (iv) Catalytic and regulatory domains of SnRK1 $\alpha 1$ exhibit differential behaviour in SGs, pointing to that they may have different roles in targeting SnRK1 $\alpha 1$ to the heat SGs (Fig. 7I, J and Fig. EV5B-E).*
- (v) SnRK1 α phosphorylation correlates with the recruitment of both SnRK1 α isoforms to SGs (Fig. 8A, C, and D).*

- (vi) *SG targeting and phosphorylation of SnRK1 α relay to the downstream signalling events, triggering expression of ASN2 and ASN6 (Fig. 8G).*
- (vii) *Finally, TSN and SGs confer heat-induced activation of SnRK1 pathway (Fig. 8A, B, E-G).*

In summary, this submission provides new information about complex, heterogeneous and dynamic protein assemblies found in the plant cytoplasm that are highly analogous to complexes described for other eukaryotic models. The study provides new insights, but in many places a more cautious interpretation of the data is needed.

Specific Concerns Essential to be addressed Abstract and Introduction

Lines 40-41. Delete "profound" and "high"; unnecessary hyperbole. There are other places where there is unnecessary use of adjectives and adverbs.

Response 5:

We agree with the reviewer and removed all unnecessary hyperboles, adjectives and adverbs.

Line 48: constantly or constitutively?

Response 6:

This sentence has been removed from the abstract.

Line 94: Why is Kosmacz et al., 2019 not mentioned? What about information regarding the role of these proteins in rice endosperm development?

Response 7:

We agree, both references, Kosmacz et al., (2019) and Chou H.L et al., (2017) are now included to the introduction; see lines 82, 86 and 98 in the revised manuscript.

Line 129: unclear what is meant by a "form an 'existing dense' interaction network". I think the authors mean that this network is partially assembled. Why is it called "dense".

Response 8:

The sentence was deleted.

The introduction does not provide information on the spatial distribution of the proteins and complexes discussed. Are these all in the cytosol? The introduction does not provide any perspective on the two TSNs of Arabidopsis. Is TSN1 functionally equivalent to TSN2, for example?

Response 9:

For Arabidopsis TSN homologues, see lines 99-101. We also refer to our review Gutierrez-Beltran et al. 2016 where we dwell on intracellular localization of TSN in various lineages (lines 97-99).

Results:

Lines 151/155: More details of the growth conditions, plant age, heat stress conditions, and region/cells of the root viewed is needed.

Response 10:

All figure legends are now supplemented with the requested details. They can also be found in the corresponding sections of Materials and methods.

Line 168: Is the same growth/age/stress conditions used for the imaging?

Response 11:

All figure legends are now supplemented with the requested details. They can also be found in the corresponding sections of Materials and methods.

Figure 1: Is it the nucleus the black negative space the cells in following the HS? This panel seems very different.

Response 12:

Thanks for spotting this out. We have now replaced this panel, so that all panels in a new Fig. 1B show similar optical plane.

Line 178: Is there a reference for the statement of well-known SG proteins? Kosmacz et al., 2019?

Response 13:

We have now re-analyzed the proteomics data using MaxQuant intensity-based absolute quantification (iBAQ), which yields summed intensity values of the identified peptides divided by the number of theoretical peptides (Esgleas, Falk et al., 2020, Tyanova, Temu et al., 2016). As a consequence, we amended the text in several places, including the one you mention.

Line 183: presumptively physiologically relevant?

Response 14:

Good point, we now added "presumptively" (line 177 in the revised manuscript).

Line 197-198: comment about lack of DCP and XRN as PB components, but isn't VCS a considered a scaffold of DCPs? This is addressed on line 458 of the discussion, but highlights that protein bodies are also heterogeneous.

Response 15:

This part has been largely re-written and we took away the comparison between SGs and PBs, as it appeared out of scope in the revised story.

Figure 2 legend. Provide citation for the SG proteomes of yeast and humans. Were these defined under stress or non-stress conditions? Make clear if Figure 2 focuses on "stress".

Response 16:

This is new Figure 1D, E, F. The mammalian and yeast data are from Jain et al. (2016), the reference now included to the figure legend.

The Rbp47b interactors were defined under heat/ darkness conditions, correct. The text could do a better job explaining that the first analysis is of "stress" interactomes.

Response 17:

This is new Figure 1E whose legend includes reference to Kosmacz et al. (2019); the conditions for RBP47-SG proteome isolation are also described in the text; lines 198-200.

Can the authors comment on the observation of different ribosomal protein isoforms in the HS-sensitive and HS-independent interactors, for example RPS6A and RPS6B?

Response 18:

We have re-analysed MS data using MaxQuant IBAQ, and the appearance of different ribosomal protein isoforms among HS-sensitive and HS-independent interactors is no longer a case.

Line 214 and elsewhere: Please use the standard nomenclature for initiation (eIF) and elongation (eEF) factors to avoid confusion in text, figures and tables. Recommend that proteins such as Rbp47 and Ubp1 be written in capitals based on Arabidopsis convention and for consistency.

Response 19:

We used standard nomenclature and capitalized protein names throughout revised manuscript.

Line 213-216: Can the authors provide an interpretation of their findings? Why might the eIF4 subunits be enriched under HS, for example?

Response 20:

It has been repeatedly demonstrated that mammalian SGs contain both eIF2 and eIF4 subunits representing sites at which stalled initiation complexes accumulate (Kimboll et al. 2003). MaxQuant IBAQ did not reveal enrichment for either eIF2 or eIF4 in heat-induced TSN interactomes, albeit components of both initiation complexes have been enriched in the TSN1_NS and TSN2_NS interactomes (Figure 5A).

Line 218 paragraph: Figure 2D can be more valuable. The network presented is not informative because there are no genes indicated and the relationship between the two networks is obscure. Colors could be used to highlight what is constitutive, such as showing TSN2, Rbp47, Ubp1 and PAB4. A high resolution PDF of these networks should be provided in the supplement.

Response 21:

Absolutely, we have now enlarged this figure and highlighted TSN proteins, PAB4, and PUB8 (see new Appendix Figure 2).

Line 249-250: The authors concluded that the high disorder characteristic of TSN2 associated proteins supports the conclusion that TSN is a scaffolding factor. An experimental demonstration of required is to reach this conclusion. How can one be certain that Rbp47b or Ubp1 is not what seeds assembly of the complex? The statement from 264-266 is too speculative.

Response 22:

Thank you very much for lifting up this important issue. The revised manuscript provides a major evidence for TSN serving a scaffold role through the analysis of RBP47 interactome in WT and TSN-deficient plants (see new Figure 2 and our Response 3 above). The results of IDR and LLPS analyses are rather explanatory for why TSN represents efficient scaffold. Since MS/MS data have been re-analyzed using MaxQuant IBAQ, we had to redo ID and LLPS analyses using new protein sets. This time we have chosen GFP-TAPa protein interactor pools as controls and used IUPred and PSPpredictor algorithms (Erdos & Dosztanyi, 2020, Sun., Li. et al., 2019) for evaluating ID and LLPS parameters, respectively, of both TSN2_NS and TSN2_HS protein pools. The analysis revealed a significant enrichment in both pools of ID-containing proteins and proteins prone to phase separation (see new Fig. 4A and B). To strengthen bioinformatics results with functional data, we have included panel D to a new Figure 4 showing that it's a highly disordered N-terminal part of TSN composed of tandem repeat of four SN domains which interacts with SG components under heat stress. Yet, we agree with the reviewer that other proteins, including RBP47 or UBPI, could recruit their IDRs to contribute to SG assembly. In mammalian cells, several SG components such as TIA1 (Gilks et al., 2004), HDAC6 (Kwon et al., 2007), G3BP1 and G3BP2 (Kedersha et al., 2016), PRRC2C (Youn et al., 2018), CSDE1 (Youn et al., 2018), and UBAP2L (Markmiller et al., 2018) were shown to be essential for SG assembly (see also our Response 3 above).

Line 316: CTT can be defined. Related to this, the protein "groups" are not explained in the text for figure 5A and 6A.

Response 23:

CTT and other protein abbreviations have now been spelled out. The protein groups 1, 2 and 3 shown in Fig. 5A are defined on the Fig. 5A itself (see boxed text), as well as in the figure legend. Fig. 6A has been modified after MaxQuant IBAQ re-analysis and does not any longer include group classification.

Line 353-359 and Figure 6: Was a different root region used for the NS and HS evaluation of foci? Text unclear. Is the HS region more apical? These results are fascinating. Was a time course or concentration gradient of NaCl used? Perhaps the foci for TCTP and SnRK1.2 reflect a more severe stress.

Response 24:

We appreciate this comment and made the corresponding part of the manuscript clearer. The panel C of Figure 6 shows root elongation zones of 5-day-old Arabidopsis GFP-tagged lines. To have uniform optical plane on all images, we have now replaced some images for new ones. Also, the text in the results section (see lines 378-386) and figure legends has been modified for better explanation of this experiment. To induce formation of SGs under salt treatment, we incubated 5-day-old seedling on MS plates containing 200 mM of NaCl, as previously described (Hamada, Yako et al., 2018). While RBP47 and UBPI granules become visible at 40 min, SnRK1 or TCTP foci are still absent at 60 min after onset of salt stress. However, one cannot exclude that TCTP or SnRK1 foci might appear in response to a more severe salt stress.

Discussion:

The discussion is generally very informative. It does not repeat the results and might also be more cautious with conclusions.

Response 25:

The discussion has been largely re-written to match new results and to convey novel findings in more coherent and concise way.

Line 478: The authors should consider the recent findings of Van Leen et al., Nature Plants. doi: 10.1038/s41477-019-0378. Less related, the authors might consider the work of Kawa et al. Plant Physiology. doi: 10.1104/pp.19.00818. doi: 10.1104/pp.19.00818.

Response 26:

Thanks, van Leen et al., reference is indeed highly relevant and now appears on lines 576, 580, and 581 of the revised manuscript.

Methods:

The methods appear to lack a description some of the stable Arabidopsis genotypes developed for this study, i.e., as described starting on line 322, the GFP-RH12. It is unclear from the methods when the 35S or a native promoter was used. Details of the constructs and the generation of the plants used should be mentioned in an appropriate location (methods or supplemental table).

Response 27:

Explicit details of all the constructs used in the present study are now summarized in Appendix Table S2. Furthermore, the constructs used in different experiments are provided in the legends of the corresponding figures.

Line 607: Add the method used to heat stress N. benth plants or leaves.

Response 28:

Done, see lines 900-901 in the revised manuscript.

Minor Concerns that should be Addressed

Lines 40-41. Delete "profound" and "high"; unnecessary hyperbole. There are other places where there is unnecessary use of adjectives and adverbs.

Response 29:

We agree with the reviewer and removed all unnecessary hyperboles, adjectives and adverbs.

Line 48: constantly or constitutively?

Response 30:

This sentence has been removed from the abstract.

1. Datasets:

These are well formatted. It was not possible for me to evaluate the quality of the proteomic analyses from what was provided. Presumably the mass spectrometry data will be cataloged.

Response 31:

The MS data have been now re-analysed using MaxQuant iBAQ algorithm. The raw proteomics data have been submitted to the JPOST repository.

For TSN interactome, temporary URL:

<https://repository.jpostdb.org/preview/1859198337605a27b62caaf>. Access key: 9765. For RBP47 interactome, temporary URL: <https://repository.jpostdb.org/preview/1808935749605a2892311b6>.

Access key: 7277

Abbreviations: There are a number of places where abbreviations are not spelled out. The abbreviation MLO is not needed; its unusual and only used once again in the text.

Response 32:

Abbreviation MLO is removed; all abbreviations are spelled out.

Referee 2:

1) The authors present careful proteomic investigations of TSN interactors in stressed and non-stressed conditions. However, It is clear that they have not reached saturation in their analysis (eg indicated that different ribosomal protein are identified at different conditions). This is easily explained by the proteomic approach used. They have used DDA acquisition which means that in a given time only the most abundant ions are taken for fragmentation. Abundance in this sense have poor connection to the abundance of the peptide in the sample the peptide sequence is of great importance here as well. Some peptides "fly" better than others in the mass spec and therefore are easier to detect. As only a given number of peptides are selected for identification at each given time this top list heavily influenced for the flyability of peptides. Thus a protein that is detected in sample 1 can be at equal concentration (or even higher) in sample 2 without being detected as the protein was kicked out of the top list by the appearance of other peptides the were more abundant. Therefore, the approach used here opens up for a lot miss-interpretation of the data. Especially since that authors is dependent on "presence" and "absence" in the different samples. This risk is even bigger in a situation where the lists are far from saturation - this random effect have a huge impact. There are better proteomics approaches used that don't select the "abundant peptides/ions that would have served this study better.

Response 1:

For TSN interactomes, we originally employed the widely used data-dependent acquisition (DDA) workflow (Burdova et al., 2019; Maffei et al., 2020; Nelson et al., 2020), where the most abundant peptide ion precursors from a survey scan (MS1 spectrum) are isolated individually and fragmented resulting in an MS/MS or MS2 spectrum. Although the DDA-LC-MS workflow may have some flaws, we have been able to identify numerous TSN partners under different conditions. However, as very correctly pointed out by the reviewer, relative quantification of the identified TSN interacting proteins is missing.

To address this important point, we have re-analyzed all MS data throughout the entire manuscript to determine the relative abundance of proteins using MaxQuant intensity-based absolute quantification (iBAQ). The latter is the result of the summed intensity values of the identified peptides divided by the number of theoretical peptides (Esgleas et al., 2020, Tyanova et al., 2016). To compare protein abundances derived from all dataset, the iBAQ protein intensities were normalized by the total sum of all protein intensities, followed by a \log_2 transformation. In order to identify specific interactors of TSN, we filtered the results using a two-step procedure. First, we selected specifically proteins enriched in TSN dataset compared to those from GFP immunoprecipitation (relative change > 2; P value < 0.05). Second, proteins were filtered based on subcellular localization according to The Arabidopsis Subcellular Database SUBA, version 4 (Hooper, Castleden et al., 2017), excluding proteins localized to chloroplasts. MaxQuant iBAQ algorithm is one of the most efficient tools available for label-free quantification of proteins (Arora, Abel et al., 2020, Banzhaf, Yau et al., 2020, Esgleas et al., 2020).

2) The authors make a big thing about finding SnRK1 among their interactors. However, they present no mechanistic model of how SnRK1 contributes to stress signaling more than that it is part of the interesting Stress granule. It is just picking the most famous protein out of a quite long list of proteins and we as readers get very little understanding on how this affect downstream processes. In meetings, Snrk1 have been reported to interact with nearly all parts of the plant cell (mitochondria, chloroplast, nuclear, also chromatin) so Im not so surprised to see it in the stess granule as well. I wanted to know how this stress granule/TSN interaction affect downstream SnRK1 processes.

Response 2:

We appreciate the reviewer' criticism which provoked us into further exploring the mechanistic connection between SnRK1 and TSN in the context of SG assembly. We have carried out additional experiments presented in new Figures 7, 8, and EV5 to deepen this part even further. The data of the revised manuscript demonstrate that:

- (i) TSN-SnRK1 interaction pre-exists prior to stress and is not influenced by HS, which agrees with our proteomic data (Fig. 7B; see also Fig. 3C; Dataset EV1).*
- (ii) TSN interacts in vivo with both SnRK1 α isoforms, as shown by co-IP (Fig. 7A), BiFC (Fig. 3D, 4D, Appendix Fig. S4) or FRET (Fig. 7C).*
- (iii) TSN takes part in the assembly of SnRK1 α isoforms in SGs and is required for full mobility of SnRK1 $\alpha 1$ (Fig. 7E – H, EV5A).*
- (iv) Catalytic and regulatory domains of SnRK1 $\alpha 1$ exhibit differential behaviour in SGs, pointing to that they may have different roles in targeting SnRK1 $\alpha 1$ to the heat SGs (Fig. 7I, J and Fig. EV5B-E).*
- (v) SnRK1 α phosphorylation correlates with the recruitment of both SnRK1 α isoforms to SGs (Fig. 8A, C, and D).*
- (vi) SG targeting and phosphorylation of SnRK1 α relay to the downstream signalling events, triggering expression of ASN2 and ASN6 (Fig. 8G).*
- (vii) Finally, TSN and SGs confer heat-induced activation of SnRK1 pathway (Fig. 8A, B, E-G).*

3) We do not know how many of these interactions that are direct. It might be the authors have found a good way of pullnig down SG and loosely connected factors. For some of the candidates, the authors show different forms of microscopical localization studies. I would have wanted biochemical proof from crosslinking etc. That could also have identified more direct interactor and served as an unbiased way to determine direct interactors. Now we rely on the ones the authors have tested (and we do not know is some were tested and show no interaction).

Response 3:

We agree with the reviewer that the hierarchy of molecular interactions within the TSN interactome remains elusive. We are currently pursuing isolation of TSN-bound protein-RNA complexes for cryo-EM analysis, but it will take time before these data will be available for publication. To strengthen the current manuscript, we have during revision focused on two lines of experiments.

First, we wanted to validate scaffolding role of TSN. It has been proposed that components of cytoplasmic foci, such as SGs, can be classified in two groups, scaffolds and clients (Banani et al., 2016). Scaffolds are defined as stably associated components required for assembly, integrity and compositional specificity of SGs (Banani et al., 2016, Ditlev, Case et al., 2018, Xing, Muhlrud et al., 2020). We have now isolated and compared RBP47 interactomes from WT vs. *tsn1 tsn2* plants under NS and HS conditions. TSN deficiency led to severe perturbation in the RBP47 interactome under both conditions (see new Fig. 2). Results of these new experiments together with our previously published FRAP data showing stable association of TSN with SGs (Gutierrez-Beltran et al., 2015) provide strong evidence for scaffolding role of this protein in SGs.

Second, the TSN-protein interactions have been further validated using a broad arsenal of tools, including:

- (i) BiFC (Fig. 3D, 4D and Appendix Figure S4),
- (ii) Reciprocal Co-IP, with the results displayed in Fig. 6B (RBP47 as a bait), 7A (the two *SnRK1 α* isoforms as baits), and Fig. EV4D (TCTP and an uncharacterized RNA-binding protein (RBP) as baits),
- (iii) Co-IP of RH12 using both *TSN1* and *TSN2* as baits (Fig. 5B),
- (iv) FRET (Fig. 7C and EV4C).

In addition, co-localization between TSN or SG marker proteins and TSN-interacting proteins has been shown for a large number of proteins, either using protoplasts (Fig. 3B, 3C, 7J, EV3 and EV5B) or transgenic plants [Fig. EV1C (*eIF4E*), Fig. EV4B (TCTP and RBP) and Appendix Figure S6 (RH12)].

4) The promoter of The TAP tagging construct is not clearly described in the results or the material and methods section. If you check the reference it is clear that the authors have used a very strong constitutive over-expressor construct (promoter is 2 times 35S). This is of course flavoring the interpretation of the interaction studies. The author might have filled the cell with TSN and identified interactions that will not happened if the TSN is expressed at normal level.

Response 4:

The double 35S promoter of the TAP construct is denoted in Fig. 1A. Tandem affinity purification coupled to mass spectrometry (TAP-MS) is one of the most advanced and widely used methods to characterize protein complexes in plants (Fernandez-Calvo, Chini et al., 2011, Henrichs, Wang et al., 2012, Van Leene, Eeckhout et al., 2015, Van Leene, Hollunder et al., 2010). Notably, TAP tagged proteins have been extensively used for physiological experiments (Lee, Kim et al., 2017, Shi, Shen et al., 2016, Yu, Rubio et al., 2008). We would like to thank the reviewer for pointing to potential caveat of using strong promoters and have therefore included new data showing that, similar to its native counterpart, TAP-tagged *TSN2* is localized in SGs (Fig. EV1C) and is fully functional in suppressing cell death (Fig. EV1D).

Minor comments:

The introduction don't clearly distinguish when they are describing plant proteins processes from when they are discussing animal/yeast protein. This makes it confusing eg is single gene knockout of TSN not lethal in plants (line 98). I know it but a less read reader might get either wrong impression.

Response 5:

The introduction has been carefully revised taking into consideration your comment.

The connection to IDPs is weak (rise from 93% to nearly 100%) although statistically significant. And not supported by experimental data. It is also not important for the rest of the reasoning so this part can be omitted.

Response 6:

We reasoned that the results of IDR and LLPS analyses would help to understand why TSN represents efficient scaffold for SG components. Since MS data have been re-analyzed using MaxQuant IBAQ, we had to redo ID and LLPS analyses using new protein sets. This time we have chosen GFP-TAP α protein interactor pools as controls and used IUPred and PSPredictor algorithms (Erdos & Dosztanyi, 2020, Sun. et al., 2019) for evaluating ID and LLPS parameters, respectively, of both TSN2_NS and TSN2_HS protein pools. The analysis revealed a significant enrichment in both pools of ID-containing proteins and proteins prone to phase separation (see new Fig. 4A and B). To support bioinformatics results with functional data, we have included panel D to a new Figure 4 showing that it's a highly disordered N-terminal part of TSN composed of tandem repeat of four SN domains which interacts with SG components under heat stress.

Figure 4: In A the group three is supposed to consist of protein confirmed I figure 4, However most are not mentioned in figure 4? What other reasons justifies this group? If something else it should be mention in legend and the comment in the figure should be corrected.

Response 7:

We believe that the reviewer refers to Figure 5A (group 3). To avoid misunderstanding, we have defined the group 3 both in Fig. 5A itself (boxed legend) and figure legend as "novel plant SG components validated in Fig. 3, EV3 and EV4", i.e. TSN1- and TSN2-interacting proteins re-distributing to cytoplasmic foci upon heat stress.

That "CHX treatment abrogated heat-induced phosphorylation of SnRK1 T175" does not suggest anything more than that protein synthesis is needed for the phosphorylation. Which might indicate that the effect is indirect (require a round of transcription/translation of a new protein), but might as well indicate that a short lived protein is required. Many proteins have natural short half-lives and therefore is the use of CHX and other protein inhibitors hard to interpret. CHX treatments cause changed expression of thousands of genes so there are many other possible interpretations.

Response 8:

This is a good point. Indeed, CHX inhibits protein synthesis (translation elongation), thus stabilizing polysomes on translating mRNAs (Ohn & Anderson, 2010). Since messenger ribonucleoproteins (mRNPs) within SGs are in a dynamic equilibrium with polysomes, CHX inhibits SG assembly (Kedersha & Anderson, 2007), and has therefore been extensively used to study SG assembly in various model organisms, ranging from yeast and plants to mammals (Boeynaems, Bogaert et al., 2017, Chantarachot et al., 2020, Gutierrez-Beltran et al., 2015, Jain, Wheeler et al., 2016, Kosmacz, Gorka et al., 2019, Namkoong, Ho et al., 2018, Wheeler, Matheny et al., 2016). We have carried out new experiments which reinforce our previous finding that inhibition of the heat-induced SnRK1 activation after a CHX treatment is dependent on the formation of SGs (and the presence of TSN). The results of new experiments demonstrate that:

- (i) CHX inhibits assembly of SnRK1 in the cytoplasmic, RBP47-positive, foci following heat stress (Fig. 7J)*
- (ii) CHX inhibits stress signalling downstream of SnRK1 α phosphorylation in WT plants, in a manner similar to TSN- or SnRK1 α -deficiency (Fig. 8G).*
- (iii) TSN level remains constant during CHX treatment (Fig. 8B), so inhibitory effect of CHX on SnRK1 α phosphorylation is not via interference with the TSN content in the cells.*

Referee 3:

The manuscript by Gutierrez-Beltran et al., focuses on the TSN proteins to investigate Arabidopsis stress granules. The study makes the interesting finding that the phosphorylation of SnRK1 that occurs in response to heat shock, depends upon TSN1 and TSN2 function. The study begins with the proteomic analysis of proteins co-purifying with TSN2 by using a TAP-tagging approach. Although the tagged protein is shown by microscopy to associate with presumed stress granules, the functionality of the tagged protein is not assessed by complementation of the *tsn1tsn2* double mutant (L155).

Response 1:

Good comment and our apologies for not including complementation data in the original submission. The complementation analysis is shown in a new Fig. EV1D; we also include new data showing that TAP-tagged TSN2 is localized in SGs (Fig. EV1C).

The biological replicates used here aren't clearly presented - but there is more detail in the methods section (L571). However, there is no statistical analysis of the data - for example to reveal the significance of peptide enrichment by different protein baits and in different conditions. The description of the proteomics results at L173/174 is ambiguous - "with up to 4490 hits".

Response 2:

Indeed, original manuscript lacked relative quantification of the TSN-interacting proteins. To address this important point, we have re-analyzed all MS data throughout the entire manuscript to determine the relative abundance of proteins using MaxQuant intensity-based absolute quantification (iBAQ). The latter is the result of the summed intensity values of the identified peptides divided by the number of theoretical peptides (Esgleas et al., 2020, Tyanova et al., 2016). To compare protein abundances derived from all dataset, the iBAQ protein intensities were normalized by the total sum of all protein intensities, followed by a log₂ transformation. In order to identify specific interactors of TSN, we filtered the results using a two-step procedure. First, we selected specifically proteins enriched in TSN dataset compared to those from GFP immunoprecipitation (relative change > 2; P value < 0.05). Second, proteins were filtered based on subcellular localization according to The Arabidopsis Subcellular Database SUBA, version 4 (Hooper et al., 2017), excluding proteins localized to chloroplasts. MaxQuant iBAQ algorithm is one of the most efficient tools available for label-free quantification of proteins (Arora et al., 2020, Banzhaf et al., 2020, Esgleas et al., 2020).

The data is then filtered by comparison to GFP controls and sub-cellular localisation. The bioinformatic analysis includes a study of publicly available protein-protein interaction data (L218), but the source of this is not cited, and the significance of the conclusions here is unclear (Fig 2D).

Response 3:

Thank you for bringing up this point. The information missing in the original manuscript is now added to Materials and methods, section "Bioinformatics". Further, original Fig. 2D has now been improved to be easier understood by a reader (see Appendix Fig. S2); see also explanatory text on lines 219-223.

In addition there is an analysis of the proteins for disordered regions. However, with arbitrary cut-offs of weak/moderate/high and apparently 93% of the Arabidopsis proteome containing disordered regions (L240), it is unclear what insight is produced here (Figure 3A-C).

Response 4:

We reasoned that the results of IDR and LLPS analyses would help to understand why TSN represents efficient scaffold for SG components. Since MS data have been re-analyzed using MaxQuant IBAQ, we had to redo ID and LLPS analyses using new protein sets. This time we have chosen GFP-TAPα protein interactor pools as controls and used IUPred and PSPpredictor algorithms (Erdos & Dosztanyi, 2020, Sun. et al., 2019) for evaluating ID and LLPS parameters, respectively, of both TSN2_NS and TSN2_HS protein pools. The analysis revealed a significant enrichment in both pools of ID-containing proteins and proteins prone to phase separation (see new Fig. 4A and B). To support bioinformatics results with functional data, we have included panel D to a new Figure 4 showing that it's a highly disordered N-terminal part of TSN composed of tandem repeat of four SN domains which interacts with SG components under heat stress.

Protein interactions are validated using BiFC by possibly over-expressing (the expression levels are not presented) test proteins transiently in a different experimental species - *Nicotiana benthamiana* (which is not mentioned in the Results section). However, BiFC data alone is insufficient not to conclude as stated on L286 that these proteins interacted with TSN2.

Response 5:

We agree with the reviewer that BiFC data alone is insufficient to conclude that two proteins interact. Therefore, in addition to BiFC (Fig. 3D, 4D and Appendix Figure S4) protein-protein interaction have been further validated using other methods:

- (i) Reciprocal Co-IP, with the results displayed in Fig. 6B (RBP47 as a bait), 7A (the two SnRK1 α isoforms as baits), and Fig. EV4D [TCTP and an uncharacterized RNA-binding protein (RBP) as baits],*
- (ii) Co-IP of RH12 using both TSN1 and TSN2 as baits (Fig. 5B),*
- (iii) FRET (Fig. 7C and EV4C).*

In addition, co-localization between TSN or SG marker proteins and TSN-interacting proteins has been shown for a large number of proteins, either using protoplasts (Fig. 3B, 3C, 7J, EV3 and EV5B) or transgenic plants [Fig. EV1C (eIF4E), Fig. EV4B (TCTP and RBP) and Appendix Figure S6 (RH12)]. Experimental conditions for all protein-protein interaction and co-localization analyses are now explicitly described in the figure legends and Materials and methods.

Overall, I find the proteomics section relies heavily on previously established concepts and not enough on unbiased analysis of the data produced here.

Response 6:

Please see response 2 above.

The authors then identify protein partners of TSN1 and compare heat and salt stress. Interestingly they detect an association with the kinase SnRNK1. They show that the heat stress-dependent phosphorylation of SnRNK at T175 is dependent on TSN1 and 2. However, the presentation of the data here is problematic. The western blots are separated such that direct comparisons between conditions cannot be made Fig 7C-F.

Response 7:

*We kindly provide the reviewer with the immunoblot images of protein samples collected at 0 min and 60 min post-heat stress from experiments presented in the original Fig. 7C-F (Fig. 8A, B, E and F in the revised manuscript) and now loaded on the same gel. The chart at the bottom shows SnRK1 activity, expressed as the ratio of phosphorylated to total SnRK1 protein calculated by taking a ratio of integrated band intensity (for both isoforms) normalized to 0 min. The blot shows that the two SnRK1 α isoforms were rapidly activated by stress in WT and complementation line (*tsn1 tsn2; TSN2*) plants, the effect being abrogated by TSN deficiency (*tsn1 tsn2*) or CHX treatment.*

In the Discussion the authors describe a model L500-508 that they say is based on their results. However, there is no data in the present manuscript that corresponds to the 3 steps they propose. Instead, this largely reflects the output of the stress granule field as a whole over many years.

Response 8:

We agree with the reviewer and have removed the model from the revised manuscript.

Overall, I have reservations about the quality of the proteomics analysis and the new insight it reveals.

Response 9:

The proteomics analyses have been redone. Please see response 2 above.

I found the connection to SnRNK1 to be interesting, but it comprises a modest part of the manuscript and the functional significance is not examined.

Response 10:

We appreciate the reviewer's criticism which provoked us into further exploring the mechanistic connection between SnRK1 and TSN in the context of SG assembly. We have carried out additional experiments presented in new Figures 7, 8, and EV5 to deepen this part. The data of the revised manuscript demonstrate that:

- (i) TSN-SnRK1 interaction pre-exists prior to stress and is not influenced by HS, which agrees with our proteomic data (Fig. 7B; see also Fig. 3C; Dataset EV1).*
- (ii) TSN interacts in vivo with both SnRK1 α isoforms, as shown by co-IP (Fig. 7A), BiFC (Fig. 3D, 4D, Appendix Fig. S4) or FRET (Fig. 7C).*
- (iii) TSN takes part in the assembly of SnRK1 α isoforms in SGs and is required for full mobility of SnRK1 α 1 (Fig. 7E – H, EV5A).*
- (iv) Catalytic and regulatory domains of SnRK1 α 1 exhibit differential behaviour in SGs, pointing to that they may have different roles in targeting SnRK1 α 1 to the heat SGs (Fig. 7I, J and Fig. EV5B-E).*
- (v) SnRK1 α phosphorylation correlates with the recruitment of both SnRK1 α isoforms to SGs (Fig. 8A, C, and D).*
- (vi) SG targeting and phosphorylation of SnRK1 α relay to the downstream signalling events, triggering expression of ASN2 and ASN6 (Fig. 8G).*
- (vii) Finally, TSN and SGs confer heat-induced activation of SnRK1 pathway (Fig. 8A, B, E-G).*

The phenotype of *tsn1 tsn2* double mutants with respect to stress is not commented on/cited here.

Response 11: The phenotype of tsn1 tsn2 double knock-out is now described on lines 151-153 in the revised manuscript. Furthermore, root cell-death phenotype of tsn1 tsn2 is shown in a new Figure EV1D.

Other comments

1. Only 1 side of error bars are shown (dynamite plots) both sides should be visible

Response 12: All charts have been modified to show both upper and lower whiskers.

2. Some Venn diagrams are scaled to the numbers involved, but others are not

Response 13: Venn diagrams included in new Figures 1D, 1E, 5A, EV2A and EV2B have now been scaled. Scaling the diagram shown in Figure 6A is problematic due to a low number of proteins in the TSN2_NaCl protein pool.

3. In Fig 6c the lanes probed with anti-TSN and anti-GFP antibodies do not line up correctly.

Response 14: The figure has been now corrected.

4. Figure S1 + lines 153-155: Colocalization with known SG markers, should be presented to be sure that indeed TSN2-TAPa is redistributed to SGs upon heat stress.

Response 15: Thanks for pointing this out. Co-localization analysis of TSN2-TAP with the SG marker eIF4E has now been added to the revised manuscript (see Fig. EV1C).

5. Lines 154-155: SGs can be formed in various stress conditions. Confirmation of heat stress induction by at least showing increased expression of stress markers should be done to be sure that proper stress induction was achieved.

*Response 16: Although formation of microscopically visible SGs is considered as a stress marker itself (Mahboubi & Stochaj, 2017), we have now added new data showing enhanced expression of two heat stress marker genes [HSP101 and HSF; (Pecinka, Dinh et al., 2010)] in WT, *tsn1 tsn2* and *snrk1α1^{-/-} snrk1α2^{+/-}* backgrounds (see Appendix Figure S2).*

6. Figure S3: What are the various bands detected in GFP-TAPa input control?

Response 17: Now Figure EV1E; these are unspecific bands in the input fraction from GFP-TAPa expressing plants detected by anti-Myc.

7. Figure S4 and lines 173-175: A large number of background interactors are detected with GFP alone, raising the possibility that the purification is insufficiently stringent.

Response 18: The experimental settings were carefully adjusted to get the most optimal conditions for TAPa. To decrease the number of false positives, the TAPa system has two steps of purification (see Fig. EV1A). Finally, the following observations make us to believe that our TSN-TAPa analytical pipeline was properly performed:

- (i) *similarity of the obtained protein pools with the previously characterized mammalian and yeast SG proteomes (Fig. 1D-F), including SG remodelers (Fig 1E), and*
- (ii) *an increased ID content and propensity to phase separation of the TSN interactomes compared to control represented by GFP interactomes (see new Fig. 4A, B).*

8. Figure 4D and lines 284-287: BiFC experiments use a different species and do not include colocalization with known SG marker. From colocalization studies TSN2 is not exclusively present in SGs (there are spots that do not colocalize with UBP1 on Figure 4B).

Response 19: Although we have previously shown that a small proportion of TSN protein is co-localized with DCP1 (Gutierrez-Beltran et al., 2015), a marker for processing bodies, our proteomics datasets lack core PB components such as decapping enzymes (DCPs), 5'-to-3' exoribonucleases (XRN) or Sm-like proteins (LSM), indicating that TSN is a robust SG marker in plants. Importantly, SGs were defined as non-homogeneous foci containing substructures (Cirillo, Cieren et al., 2020, Wheeler et al., 2016), whose number, signal intensity or dynamics can change depending on the protein used for detection (Marmor-Kollet, Siany et al., 2020, Protter & Parker, 2016). This explains why rs/rp coefficient varies depending of the interactor analysed (Fig. 3B).

Are interactions between TSN2 and 10 selected proteins occurring in SGs or in the cytosol?

Response 20: These interactions take place in the cytosol under no stress conditions (Appendix Figure S4), and in SGs when cells are exposed to stress (Fig. 3D).

9. Figure 5C and lines 322-331: colocalization with SG marker would address if observed cytoplasmic foci are SGs or not.

Response 21: We appreciate this comment. The results of the co-localization analysis of RH12 and the SG marker RBP47 have been included to the revised manuscript (see Appendix Figure S6).

10. Figure 6D: Colocalization of proteins is missing - but this would help test if they are present in the same foci.

Response 22: Co-localization analyses of TCTP with eIF4E and of SnRK1 α with RBP47 have been included to the revised manuscript; see Figures EV4B and 7J respectively. eIF4E and RBP47 are two commonly used SG markers (Chantarachot et al., 2020, Kosmacz et al., 2019, Yan, Yan et al., 2014)

11. Lines 393-395: Stress induction confirmation should be shown

Response 23: qPCR analysis of the two heat-stress marker genes, HSP101 and HSF in WT, tsn1 tsn2 and snrk1 α 1^{-/-} snrk1 α 2^{+/-} genotypes is now provided in Appendix Figure S1.

12. · Materials & Methods: Heat stress and salinity stress induction should be described; procedure for CHX treatment has not been shown in the paper.

Response 24: The corresponding details have now been added to the Materials and methods, lines 614-617, 626-628, and 760-762.

References

- Arora D, Abel NB, Liu C, Van Damme P, Yperman K, Eeckhout D, Vu LD, Wang J, Tornkvist A, Impens F, Korbei B, Van Leene J, Goossens A, De Jaeger G, Ott T, Moschou PN, Van Damme D (2020) Establishment of Proximity-Dependent Biotinylation Approaches in Different Plant Model Systems. *The Plant cell* 32: 3388-3407
- Banani SF, Rice AM, Peeples WB, Lin Y, Jain S, Parker R, Rosen MK (2016) Compositional Control of Phase-Separated Cellular Bodies. *Cell* 166: 651-663
- Banzhaf M, Yau HC, Verheul J, Lodge A, Kritikos G, Mateus A, Cordier B, Hov AK, Stein F, Wartel M, Pazos M, Solovyova AS, Breukink E, van Teeffelen S, Savitski MM, den Blaauwen T, Typas A, Vollmer W (2020) Outer membrane lipoprotein Nlpl scaffolds peptidoglycan hydrolases within multi-enzyme complexes in Escherichia coli. *The EMBO journal* 39: e102246
- Boeynaems S, Bogaert E, Kovacs D, Konijnenberg A, Timmerman E, Volkov A, Guharoy M, De Decker M, Jaspers T, Ryan VH, Janke AM, Baatsen P, Vercruyse T, Kolaitis RM, Daelemans D, Taylor JP, Kedersha N, Anderson P, Impens F, Sobott F et al. (2017) Phase Separation of C9orf72 Dipeptide Repeats Perturbs Stress Granule Dynamics. *Molecular cell* 65: 1044-1055 e5
- Chantarachot T, Bailey-Serres J (2018) Polysomes, Stress Granules, and Processing Bodies: A Dynamic Triumvirate Controlling Cytoplasmic mRNA Fate and Function. *Plant physiology* 176: 254-269
- Chantarachot T, Sorenson RS, Hummel M, Ke H, Kettenburg AT, Chen D, Aiyetiwa K, Dehesh K, Eulgem T, Sieburth LE, Bailey-Serres J (2020) DHH1/DDX6-like RNA helicases maintain ephemeral half-lives of stress-response mRNAs. *Nat Plants* 6: 675-685
- Cirillo L, Cieren A, Barbieri S, Khong A, Schwager F, Parker R, Gotta M (2020) UBAP2L Forms Distinct Cores that Act in Nucleating Stress Granules Upstream of G3BP1. *Current biology : CB* 30: 698-707 e6
- Ditlev JA, Case LB, Rosen MK (2018) Who's In and Who's Out-Compositional Control of Biomolecular Condensates. *J Mol Biol* 430: 4666-4684

Erdos G, Dosztanyi Z (2020) Analyzing Protein Disorder with IUPred2A. *Curr Protoc Bioinformatics* 70: e99

Esgleas M, Falk S, Forne I, Thiry M, Najas S, Zhang S, Mas-Sanchez A, Geerlof A, Niessing D, Wang Z, Imhof A, Gotz M (2020) Trnp1 organizes diverse nuclear membrane-less compartments in neural stem cells. *The EMBO journal* 39: e103373

Fernandez-Calvo P, Chini A, Fernandez-Barbero G, Chico JM, Gimenez-Ibanez S, Geerinck J, Eeckhout D, Schweizer F, Godoy M, Franco-Zorrilla JM, Pauwels L, Witters E, Puga MI, Paz-Ares J, Goossens A, Reymond P, De Jaeger G, Solano R (2011) The Arabidopsis bHLH transcription factors MYC3 and MYC4 are targets of JAZ repressors and act additively with MYC2 in the activation of jasmonate responses. *Plant Cell* 23: 701-15

Gilks N, Kedersha N, Ayodele M, Shen L, Stoecklin G, Dember LM, Anderson P (2004) Stress granule assembly is mediated by prion-like aggregation of TIA-1. *Molecular biology of the cell* 15: 5383-98

Gutierrez-Beltran E, Moschou PN, Smertenko AP, Bozhkov PV (2015) Tudor Staphylococcal Nuclease Links Formation of Stress Granules and Processing Bodies with mRNA Catabolism in Arabidopsis. *The Plant cell*

Hamada T, Yako M, Minegishi M, Sato M, Kamei Y, Yanagawa Y, Toyooka K, Watanabe Y, Hara-Nishimura I (2018) Stress granule formation is induced by a threshold temperature rather than a temperature difference in Arabidopsis. *Journal of cell science* 131

Henrichs S, Wang B, Fukao Y, Zhu J, Charrier L, Bailly A, Oehring SC, Linnert M, Weiwad M, Endler A, Nanni P, Pollmann S, Mancuso S, Schulz A, Geisler M (2012) Regulation of ABCB1/PGP1-catalysed auxin transport by linker phosphorylation. *EMBO J* 31: 2965-80

Hooper CM, Castleden IR, Tanz SK, Aryamanesh N, Millar AH (2017) SUBA4: the interactive data analysis centre for Arabidopsis subcellular protein locations. *Nucleic acids research* 45: D1064-D1074

Jain S, Wheeler JR, Walters RW, Agrawal A, Barsic A, Parker R (2016) ATPase-Modulated Stress Granules Contain a Diverse Proteome and Substructure. *Cell* 164: 487-98

Kedersha N, Anderson P (2007) Mammalian stress granules and processing bodies. *Methods in enzymology* 431: 61-81

Kedersha N, Panas MD, Achorn CA, Lyons S, Tisdale S, Hickman T, Thomas M, Lieberman J, McInerney GM, Ivanov P, Anderson P (2016) G3BP-Caprin1-USP10 complexes mediate stress granule condensation and associate with 40S subunits. *The Journal of cell biology* 212: 845-60

Kosmacz M, Gorka M, Schmidt S, Luzarowski M, Moreno JC, Szlachetko J, Leniak E, Sokolowska EM, Sofroni K, Schnittger A, Skiryicz A (2019) Protein and metabolite composition of Arabidopsis stress granules. *New Phytol* 222: 1420-1433

Kwon S, Zhang Y, Matthias P (2007) The deacetylase HDAC6 is a novel critical component of stress granules involved in the stress response. *Genes & development* 21: 3381-94

Lee BD, Kim MR, Kang MY, Cha JY, Han SH, Nawkar GM, Sakuraba Y, Lee SY, Imaizumi T, McClung CR, Kim WY, Paek NC (2017) The F-box protein FKF1 inhibits dimerization of COP1 in the control of photoperiodic flowering. *Nature communications* 8: 2259

Mahboubi H, Stochaj U (2017) Cytoplasmic stress granules: Dynamic modulators of cell signaling and disease. *Biochimica et biophysica acta* 1863: 884-895

Markmiller S, Soltanieh S, Server KL, Mak R, Jin W, Fang MY, Luo EC, Krach F, Yang D, Sen A, Fulzele A, Wozniak JM, Gonzalez DJ, Kankel MW, Gao FB, Bennett EJ, Lecuyer E, Yeo GW (2018) Context-Dependent and Disease-Specific Diversity in Protein Interactions within Stress Granules. *Cell* 172: 590-604 e13

Marmor-Kollet H, Siany A, Kedersha N, Knafo N, Rivkin N, Danino YM, Moens TG, Olender T, Sheban D, Cohen N, Dadosh T, Addadi Y, Ravid R, Eitan C, Toth Cohen B, Hofmann S, Riggs CL, Advani VM, Higginbottom A, Cooper-Knock J et al. (2020) Spatiotemporal Proteomic Analysis of Stress Granule Disassembly Using APEX Reveals Regulation by SUMOylation and Links to ALS Pathogenesis. *Molecular cell* 80: 876-891 e6

Namkoong S, Ho A, Woo YM, Kwak H, Lee JH (2018) Systematic Characterization of Stress-Induced RNA Granulation. *Molecular cell* 70: 175-187 e8

Ohn T, Anderson P (2010) The role of posttranslational modifications in the assembly of stress granules. *Wiley interdisciplinary reviews RNA* 1: 486-93

Pecinka A, Dinh HQ, Baubec T, Rosa M, Lettner N, Mittelsten Scheid O (2010) Epigenetic regulation of repetitive elements is attenuated by prolonged heat stress in Arabidopsis. *The Plant cell* 22: 3118-29

Protter DS, Parker R (2016) Principles and Properties of Stress Granules. *Trends Cell Biol* 26: 668-79

Shi H, Shen X, Liu R, Xue C, Wei N, Deng XW, Zhong S (2016) The Red Light Receptor Phytochrome B Directly Enhances Substrate-E3 Ligase Interactions to Attenuate Ethylene Responses. *Dev Cell* 39: 597-610

Sun. T, Li. Q, Xu. Y, Zhang. Z, Lai. L, Pei. J (2019) Prediction of liquid-liquid phase separation proteins using machine learning. *bioRxiv*

Tyanova S, Temu T, Cox J (2016) The MaxQuant computational platform for mass spectrometry-based shotgun proteomics. *Nature protocols* 11: 2301-2319

Van Leene J, Eeckhout D, Cannoot B, De Winne N, Persiau G, Van De Slijke E, Vercruyse L, Dedecker M, Verkest A, Vandepoele K, Martens L, Witters E, Gevaert K, De Jaeger G (2015) An improved toolbox to unravel the plant cellular machinery by tandem affinity purification of Arabidopsis protein complexes. *Nat Protoc* 10: 169-87

Van Leene J, Hollunder J, Eeckhout D, Persiau G, Van De Slijke E, Stals H, Van Isterdael G, Verkest A, Neiryneck S, Buffel Y, De Bodt S, Maere S, Laukens K, Pharazyn A, Ferreira PC, Eloy N, Renne C, Meyer C, Faure JD, Steinbrener J et al. (2010) Targeted interactomics reveals a complex core cell cycle machinery in Arabidopsis thaliana. *Mol Syst Biol* 6: 397

Wheeler JR, Matheny T, Jain S, Abrisch R, Parker R (2016) Distinct stages in stress granule assembly and disassembly. *Elife* 5

Xing W, Muhlrad D, Parker R, Rosen MK (2020) A quantitative inventory of yeast P body proteins reveals principles of composition and specificity. *Elife* 9

Yan C, Yan Z, Wang Y, Yan X, Han Y (2014) Tudor-SN, a component of stress granules, regulates growth under salt stress by modulating GA2ox3 mRNA levels in Arabidopsis. *Journal of experimental botany*

Youn JY, Dunham WH, Hong SJ, Knight JDR, Bashkurov M, Chen GI, Bagci H, Rathod B, MacLeod G, Eng SWM, Angers S, Morris Q, Fabian M, Cote JF, Gingras AC (2018) High-Density Proximity Mapping Reveals the Subcellular Organization of mRNA-Associated Granules and Bodies. *Molecular cell* 69: 517-532 e11

Yu JW, Rubio V, Lee NY, Bai S, Lee SY, Kim SS, Liu L, Zhang Y, Irigoyen ML, Sullivan JA, Zhang Y, Lee I, Xie Q, Paek NC, Deng XW (2008) COP1 and ELF3 control circadian function and photoperiodic flowering by regulating GI stability. *Molecular cell* 32: 617-30

Dear Emilio,

Thank you for submitting a revised version of your manuscript. I sincerely apologise for the unusually protracted review process due to delayed submission of referee reports. Your study has now been seen by one of the original reviewers, who finds that their main concerns have been addressed and now recommends publication of the manuscript after a minor revision. Additionally, I would like to invite you to address the remaining editorial issues before I can extend the official acceptance of the manuscript:

1. Our data editor has done their pre-publication check on your manuscript. I have attached the file here. Please take a look at the word file and the comments in the Figure Legends section and respond to the issues.
2. Please rename Data Submission section into Data Availability. Please also make sure that the datasets are publicly available upon acceptance.
3. Figure panels 2A, 2B and EV5A have not been mentioned in the text.
4. Thank you for providing synopsis text and image for your manuscript. Unfortunately when resized to the required dimension of 550 x 300-600 pixels the text becomes difficult to read (please see the attached image). Please modify the image to increase readability.

Please let me know if you have any further questions regarding any of these points. You can use the link below to upload the revised files.

With best regards,

leva

leva Gailite, PhD
Scientific Editor
The EMBO Journal
Meyerohofstrasse 1
D-69117 Heidelberg
Tel: +4962218891309
i.gailite@embojournal.org

Referee #2:

I have read the manuscript and the added experimental data improves the paper however the critique about in principle cherry picking data from the proteomics datasets is still present although now the conclusions are better supported by independent experiments.

Also, the hyperbolic way of expression has been toned down a bit there are still remnants of this in the manuscript.

However, the findings are interesting and timely and will attract readers and citations so with some changes I am in favor for publishing this manuscript in EMBO journal.

Changes I would like to see prior acceptance/publication.

- 1) Clearly indicate the weakness of performing IP with a massively overexpressed bait protein. The interactors identified are at best the factors that can interact with the bait. For some of the confirming assay the authors also use overexpressed baits or heterologous systems and then this still only answers if the two protein can interact not if they interact in plants.
- 2) In the abstract the authors indicate that they have showed that TNS is an intrinsically disordered protein but that is refer from sequence and not shown in this manuscript. They have run a software tool developed by others to get this prediction.
- 3) to test the activity of the SnRK1 system the authors have checked few downstream targets of SNRK1 but failed to test the most commonly used ones ASN1 and DIN1, Why using non standard markers ? If the standard markers don't give the expected results the reader need to know.
- 4) In the Venn diagrams Figure 1 and 5 the author highlights an interesting group of genes and then subdivide that group according to another criteria. However, in both cases is the biggest group (16 and 95 proteins in 1F and 57 in figure 5A) not listed. This gives the impression that a majority of the protein in the Venn defined group has this function that is the basis for the sub-listing. Include for both figures one more group ("the rest" or "junk" or "we don't know that these are" group).

We wish to thank the reviewer for their constructive criticism and further useful comments on our revised manuscript.

Below is a point-by-point response to the reviewers' comments

Referee #2:

I have read the manuscript and the added experimental data improves the paper however the critique about in principle cherry picking data from the proteomics datasets is still present although now the conclusions are better supported by independent experiments. Also, the hyperbolic way of expression has been toned down a bit there are still remnants of this in the manuscript.

We have now taken away the whole sentence (lines 294-296 in the previous version of the manuscript) and deleted or toned down all the adjectives throughout the manuscript that might sound as hyperbolic.

However, the findings are interesting and timely and will attract readers and citations so with some changes I am in favor for publishing this manuscript in EMBO journal. Changes I would like to see prior acceptance/publication.

1) Clearly indicate the weakness of performing IP with a massively overexpressed bait protein. The interactors identified are at best the factors that can interact with the bait. For some of the confirming assay the authors also use overexpressed baits or heterologous systems and then this still only answers if the two protein can interact not if they interact in plants.

We fully agree with the reviewer and added a new sentence (lines 531-534 in the revised manuscript) pointing to a caveat of using strong promoters for TAP.

2) In the abstract the authors indicate that they have showed that TNS is an intrinsically disordered protein but that is refer from sequence and not shown in this manuscript. They have run a software tool developed by others to get this prediction.

A good point; the abstract is modified accordingly.

3) to test the activity of the SnRK1 system the authors have checked few downstream targets of SNRK1 but failed to test the most commonly used ones ASN1 and DIN1, Why using non standard markers? If the standard markers don't give the expected results the reader need to know.

Thank you for bringing up this important point. We have realized that we have made a mistake in the gene nomenclature. ASN2 and ASN6 genes used in our study actually correspond to DARK INDUCIBLE 2 (DIN2; At3g60140) and DIN6 [also named as ASPARAGINE SYNTHASE1 (ASN1); At3g47340] respectively. In fact, the qPCR primers shown in Appendix Table S1 align to DIN2 and DIN6. The mistake has been corrected in the new version of the manuscript.

To confirm that both TSN interaction and SG localization are required for the activation of SnRK1 signalling, we initially performed qPCR analyses for three genes: DIN6 and DORMANCY ASSOCIATED GENE 2 (DRM2/AXP; At2g33830), two commonly used target

genes of the SnRK1-dependent signalling pathway (Belda-Palazon et al, 2020; Crozet et al, 2016; Ramon et al, 2019; Rodrigues et al, 2013; Zhang et al, 2009), as well as DIN2, another marker gene of the same pathway selected from Baena-Gonzalez et al. 2007 (see Supplementary Table S3 in their work). However, to avoid overloading of the manuscript, the DRM2 results (the plot shown below) were finally omitted.

4) In the Venn diagrams Figure 1 and 5 the author highlights an interesting group of genes and then subdivide that group according to another criteria. However, in both cases is the biggest group (16 and 95 proteins in 1F and 57 in figure 5A) not listed. This gives the impression that a majority of the protein in the Venn defined group has this function that is the basis for the sub-listing. Include for both figures one more group ("the rest" or "junk" or "we don't know that these are" group).

Thank you for spotting this out. We now provide a clear explanation for where are the larger groups of the TSN-interacting protein in the legends for Figures 1E and 5A (see lines 1189-1191 and 1243-1245 in the revised manuscript).

REFERENCES

- Baena-Gonzalez E, Rolland F, Thevelein JM, Sheen J (2007) A central integrator of transcription networks in plant stress and energy signalling. *Nature* 448: 938-942
- Belda-Palazon B, Adamo M, Valerio C, Ferreira LJ, Confraria A, Reis-Barata D, Rodrigues A, Crozet P, Margalha L, Butowt R, Fernandes N, Elias CA, Orosa B, Tomanov K, Teige M, Bachmair A, Sadanandom A et al (2016) SUMOylation represses SnRK1 signaling in Arabidopsis. *The Plant journal : for cell and molecular biology* 85: 120-133
- Ramon M, Dang TVT, Broeckx T, Hulsmans S, Crepin N, Sheen J, Rolland F (2019) Default Activation and Nuclear Translocation of the Plant Cellular Energy Sensor SnRK1 Regulate Metabolic Stress Responses and Development. *The Plant cell* 31: 1614-1632
- Rodrigues A, Adamo M, Crozet P, Margalha L, Confraria A, Martinho C, Elias A, Rabissi A, Lumberras V, Gonzalez-Guzman M et al (2013) ABI1 and PP2CA phosphatases are negative regulators of Snf1-related protein kinase1 signaling in Arabidopsis. *The Plant cell* 25: 3871-3884
- Zhang Y, Primavesi LF, Jhurreea D, Andralojc PJ, Mitchell RA, Powers SJ, Schlupepmann H, Delatte T, Wingler A, Paul MJ (2009) Inhibition of SNF1-related protein kinase1 activity and regulation of metabolic pathways by trehalose-6-phosphate. *Plant physiology* 149: 1860-1871

Thank you again for submitting the final revised version of your manuscript. I am pleased to inform you that we have now accepted it for publication in The EMBO Journal.

Your article will be processed for publication in The EMBO Journal by EMBO Press and Wiley, who will contact you with further information regarding production/publication procedures and license requirements.

Corresponding Author Name: Emilio Gutierrez-Beltran, Peter Bozhkov

Journal Submitted to: EMBO J

Manuscript Number: EMBOJ-2020-105043